# BRD9 determines the cell fate of hematopoietic stem cells by regulating chromatin state

Muran Xiao[1,2,25], Shinji Kondo[3,4,25], Masaki Nomura[1,5,25], Shinichiro Kato[6,7,8], Koutarou Nishimura[1], Weijia Zang[1,9], Yifan Zhang[1,9], Tomohiro Akashi[8,10], Aaron Viny[11], Tsukasa Shigehiro[12], Tomokatsu Ikawa[12], Hiromi Yamazaki[1], Miki Fukumoto[1], Atsushi Tanaka[1,13], Yasutaka Hayashi[1], Yui Koike[1], Yumi Aoyama[1,9], Hiromi Ito[1], Hiroyoshi Nishikawa[6,7,8,14], Toshio Kitamura[1,2], Akinori Kanai[15,16], Akihiko Yokoyama[17], Tohru Fujiwara[18,19], Susumu Goyama[20], Hideki Noguchi[3,4], Stanley C. Lee[21,22], Atsushi Toyoda[4,23], Kunihiko Hinohara[6,7,8], Omar Abdel-Wahab[24] & Daichi Inoue[1,9] ✉

ATP-dependent chromatin remodeling SWI/SNF complexes exist in three subcomplexes: canonical BAF (cBAF), polybromo BAF (PBAF), and a newly described non-canonical BAF (ncBAF). While cBAF and PBAF regulate fates of multiple cell types, roles for ncBAF in hematopoietic stem cells (HSCs) have not been investigated. Motivated by recent discovery of disrupted expression of BRD9, an essential component of ncBAF, in multiple cancers, including clonal hematopoietic disorders, we evaluate here the role of BRD9 in normal and malignant HSCs. BRD9 loss enhances chromatin accessibility, promoting myeloid lineage skewing while impairing B cell development. BRD9 significantly colocalizes with CTCF, whose chromatin recruitment is augmented by BRD9 loss, leading to altered chromatin state and expression of myeloid-related genes within intact topologically associating domains. These data uncover ncBAF as critical for cell fate specification in HSCs via three-dimensional regulation of gene expression and illuminate roles for ncBAF in normal and malignant hematopoiesis.

SWI/SNF complexes are ATP-dependent chromatin remodeler complexes that play critical roles in gene regulation by modulating chromatin architecture and DNA accessibility[1–4]. SWI/SNF can be grouped into three subcomplexes of differing sizes: canonical BAF (cBAF), polybromo BAF (PBAF), and the newly identified non-canonical BAF (ncBAF, also known as GBAF)[5–9]. Distinct from cBAF or PBAF, ncBAF lacks the core BAF subunits ARID1/2, BAF47, and BAF57, but includes the unique subunits GLTSCR1/1L and BRD9, a bromodomain-containing protein[5,8,10].

The observation that more than 20% of human malignancies have genetic alterations in SWI/SNF subunits, with specific subunits mutated in distinct malignancies[7,11], suggests subunit- and complex-specific functions[12–15]. Although ncBAF-specific components (BRD9, GLTSCR1, and GLTSCR1L) are rarely mutated in cancer, we recently identified that BRD9 is substantially downregulated due to mRNA degradation from mis-splicing by mutations in the spliceosomal protein, SF3B1[16], which occur in ~30% of myelodysplastic syndrome (MDS) patients[17,18]. These data suggest that ncBAF may play a role in maintaining normal

hematopoiesis. However, in contrast with the pan-BAF component, BRG, the ATPase subunit of BAF chromatin remodeling complex, BAF155, and BAF53a, and the cBAF or PBAF-specific components ARID1A, ARID2, and BAF45a[19–23], functional consequences of ncBAF loss in normal and malignant hematopoiesis in vivo are currently unknown.

Here, using a newly generated *Brd9* conditional knockout (KO) murine model, we identify that BRD9-depleted HSCs alter their cell fate both quantitatively and qualitatively in a cell-autonomous fashion undergoing myeloid skewing while impairing B cell lineage development and self-renewal capacity. BRD9 loss increases CTCF-binding and the formation of chromatin loops thereby inducing differential gene expression with topologically associating domains (TADs) and A/B compartments intact. Consistent with this, we find that BRD9 is essential for the development and maintenance of acute myeloid leukemia (AML) as its loss remarkably promotes myeloid differentiation with open chromatin at CTCF sites. These data thereby elucidate a mechanistic basis for BRD9/ncBAF in hematopoiesis, delineating a role for pathologic perturbation in hematologic malignancies.

## Results

### BRD9 is required for normal hematopoietic differentiation and stemness

To examine the function of BRD9 in HSCs in vitro, we depleted BRD9 using shRNA in human cord blood (CB)-derived CD34+ cells. Two independent *BRD9*-targeting shRNAs inhibited cell growth (Fig. 1a and Supplementary Fig. 1a). The effect on proliferation was more robust when sgRNA targeting the bromodomain was transduced in Cas9-expressing K562 cell line, compared to sgRNAs targeting the domain of unknown function (DUF) (Fig. 1b and Supplementary Fig. 1b). Both in murine and human HSCs, BRD9-depleted cells gave rise to significantly fewer colony-forming units (CFUs) in all lineages when the cells were plated in methylcellulose with cytokines (Fig. 1c and Supplementary Fig. 1c–g). In line with this, BRD9-depleted CD34+ CB cells exhibited greater loss of CD34+ hematopoietic stem and progenitor cells (HSPCs) and CD34+CD38− more enriched hematopoietic stem cells (HSCs) when cultured in a condition to allow HSC expansion for ten days (Fig. 1d). Additionally, we observed an increase in myelomonocytic differentiation in BRD9-depleted cells when cultured with myeloid expansion supplement compared to control, as assessed by both flow cytometric markers (CD33+: myeloid-lineage; CD14+: monocytic-lineage) and morphology (Fig. 1d and Supplementary Fig. 1h, i). The promotion of differentiation toward neutrophils was reproduced in HL60 cells treated with 1 μM all-trans retinoic acid (ATRA) for six days in terms of morphology and CD11b intensity (Fig. 1e). Moreover, BRD9 depletion in K562 cells disturbed erythroid lineage differentiation in all of the single clones, evaluated by CD71/CD235a (Fig. 1f and Supplementary Fig. 1j).

Next, we examined the in vivo effects of BRD9 loss on hematopoietic differentiation in a bone marrow (BM) transplant model. To test this, we retrovirally transduced shRNA against *Brd9* into Lin−Kit+ murine HSPCs and transplanted transduced cells into lethally-irradiated wildtype recipients. Notably, GFP+ shRNA-expressing cells in recipients exhibited significantly fewer B220+ cells and more mature CD11b+Gr1+ neutrophils (Fig. 1g). Moreover, shBrd9-transduced HSPCs reduced colony output, even in *Tet2*-null conditions, which are known to enhance the self-renewal capacity of HSPCs (Supplementary Fig. 1k–m). Next, to further determine whether the effect on the linage specification was bromodomain-dependent or not, we performed a similar transplant using retroviruses expressing GFP alone (empty vector), wildtype (WT) BRD9, and the bromodomain disrupted mutants with deletion of the bromodomain (dBD) or N216A[24]. As shown in Fig. 1g, a similar differential disturbance was observed, suggesting the importance of the BRD9-bromodomain in normal differentiation, while ectopic expression of WT had no effect. To connect

the results with *Brd9* mRNA level in each hematopoietic population, we FACS-sorted murine HSPCs and more mature fractions and performed quantitative RT-PCR of *Brd9* mRNA and found that *Brd9* mRNA was relatively abundant in mouse B cell lineage, less expressed in myeloid progenitors, and mature myeloid cells (Fig. 1h and Supplementary Data 1). Taken together, these data suggest that BRD9 depletion enhances myeloid differentiation while impairing B cell development, along with reduced stemness.

### Generation and characterization of *Brd9* knockout murine model

Given the recurrent downregulation of *BRD9* expression due to disrupted mRNA splicing of *BRD9* in MDS patients[16–18], we hypothesized that BRD9 may play an important role in the hematopoietic system in vivo. To test this hypothesis, we generated a mouse model permitting time- and tissue-specific deletion of *Brd9* (Fig. 2a and Supplementary Fig. 2a). This was accomplished by generating *Mx1*-Cre;*Brd9*fl/fl mice, as *Mx1*-Cre is a well-established system allowing for conditional, time-controlled, and efficient deletion of genes in postnatal hematopoietic cells[25–28]. Conditional Cre-mediated excision of exon 4-6, which encode the bromodomain of BRD9, efficiently downregulated *Brd9* mRNA in whole bone marrow (BM) cells with the expected DNA recombination (Fig. 2b), which was further supported by RNA-seq analysis of HSPCs (Supplementary Fig. 2b). We successfully confirmed undetectable full-length BRD9 protein without any truncated BRD9 (Fig. 2c and Supplementary Fig. 2c).

To determine the role of endogenous *Brd9* in vivo, we administered polyinosinic-polycytidylic acid (pIpC) to primary *Mx1*-Cre;*Brd9*fl/fl mice 12 weeks after birth to induce *Brd9* deletion. Peripheral blood and spleen analyses 4 months later revealed cytopenia, macrocytic anemia, striking B cell reduction, and myeloid-lineage skewing (Fig. 2d and Supplementary Fig. 2d, e). We also found that hematopoietic-specific BRD9 deletion in 24-week-old mice (12-week post-pIpC) reduced proliferation as well as the clonogenic capacity in vitro colony formation assay (Fig. 2e and Supplementary Fig. 2f, g). Evidence of reduced early B cell differentiation was further bolstered by in vitro colony assays in IL-7-enriched methylcellulose (M3630), where *Brd9* homozygous KO HSPCs severely failed to generate PreB lymphoid progenitor colonies (Fig. 2e). The effect on preB cell colonies was supported by the observation that the well-studied the bromodomain inhibitor (BI-7273[29]) reduced the pre-B colonies and pre-B cell proportion in a dose-dependent manner, indicating a functional role of the bromodomain in B-cell development (Fig. 2f and Supplementary Fig. 2h).

We next sought to delineate the blocked stage of the B-cell maturation. PreB (Lineage (CD3/Ter119/CD11b/Gr1)− IgM−CD19+CD43−) and Immature B (Lineage−IgM+CD19+CD43−) cells are strikingly reduced in *Brd9* KO mice, while PreProB (Lin−IgM−CD19−CD43+) and ProB (Lin−IgM−CD19+CD43+) stages were comparable between control and KO mice (Fig. 2g and Supplementary Fig. 2i, j). The cell-cycling population (S phase) was found to be decreased at the stage of ProB cells or later (Supplementary Fig. 2k). Although BRD9 deletion does not impact BM cell numbers (Supplementary Fig. 3a), morphologic abnormalities with BM dysplasia was evident (Supplementary Fig. 3b). BRD9 deletion also resulted in increased LSKs (Lin−Kit+Scal+ cells) in the active phase of the cell cycle (Fig. 2h and Supplementary Fig. 3c) and with elevated levels of gH2AX (a marker of DNA damage) in HSPCs (Fig. 2i and Supplementary Fig. 3d), reduced mitochondria membrane potential (Supplementary Fig. 3e), and the tendency of delayed stress erythropoiesis when treated with phenylhydrazine (PHZ), a drug that induces rapid intravascular hemolysis (Supplementary Fig. 3f)[30]. Most of these are key features of aged or MDS-like hematopoiesis[31–37]. Indeed, by reanalyzing human RNA-seq profiles of young and aged lineage−CD34+CD38− cells (n = 10 per group)[38], we found *BRD9* mRNA was significantly suppressed in aged human HSCs (Fig. 2j). Taken together, these data indicate that bromodomain-containing BRD9 is

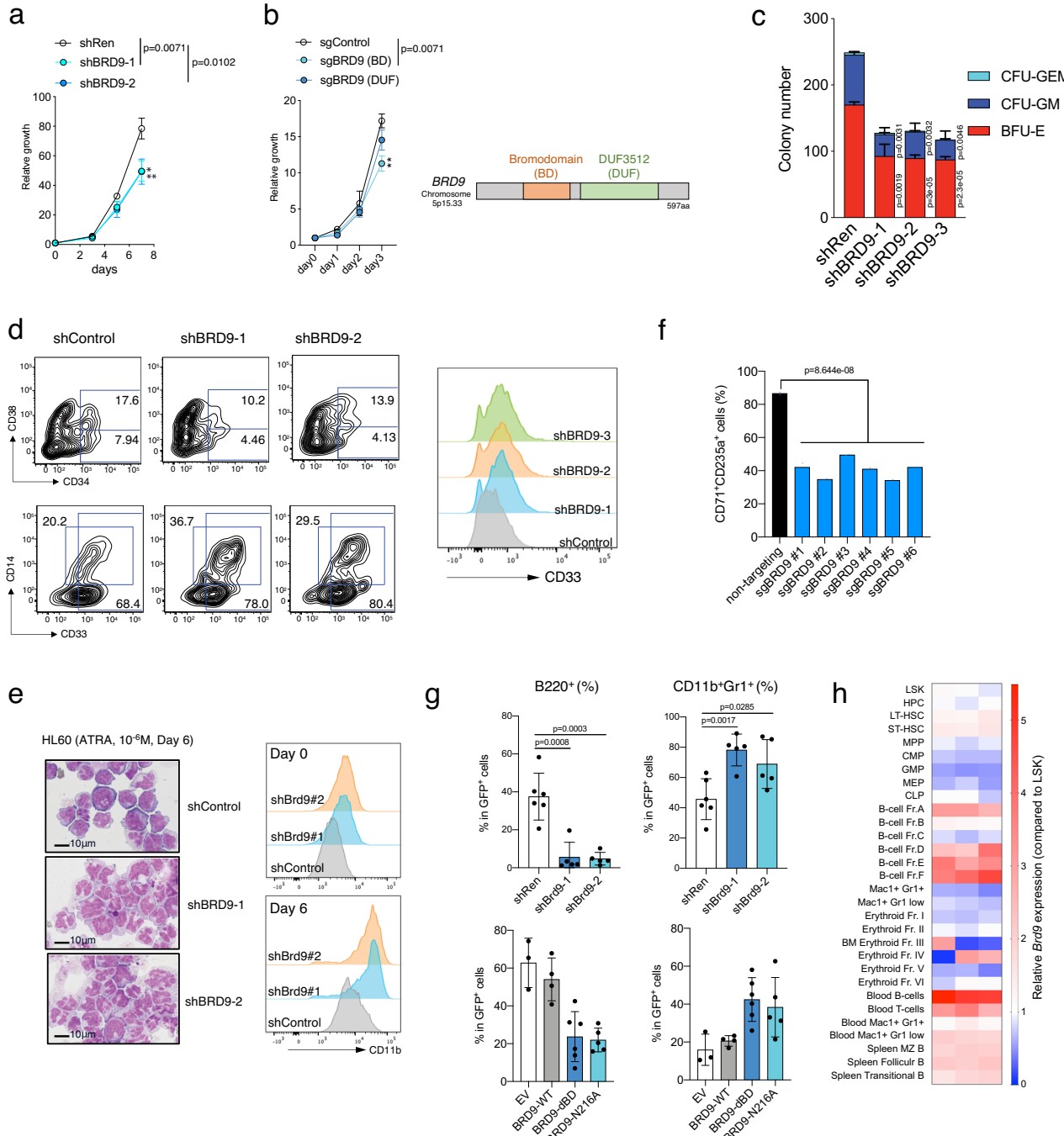

**Fig. 1 | BRD9 is required for normal differentiation and stemness of HSCs. a** The number of viable human CD34+ cord blood cells over the course of 7 days, beginning 3-days post-transduction of shRNAs. *n* = 3 independent experiments; error bars, means ± s.e.m. **b** The number of K562 cells with sgControl, sgBRD9 targeting bromodomain (BD) or domain of unknown function (DUF), over the course of 3 days. The domain structure of wild-type BRD9 is indicated (right). *n* = 3 independent experiments; error bars, means ± s.e.m. **c** Colony formation of human CD34+ cord blood transduced with a scramble (control) or BRD9 targeting shRNA. Cells were sorted on the basis of GFP positivity 3 days after lentiviral transduction and colonies were scored 10 days after plating in methylcellulose. *n* = 3 independent experiments; mean and s.e.m are plotted. **d** Expression of the surface markers for HSCs (CD34 and CD38) and for myeloid differentiation (CD11b and CD14), as assessed by flow cytometry 10 days after plating under the condition with the supplement for CD34 expansion and myeloid differentiation. The proportion of CD34+, CD34+CD38−, CD33+, and CD14+ are indicated. Representative histograms of

CD33 staining in shControl and shBRD9 HSCs are shown on the right.
**e** Representative cytosin images (left) and histograms of CD11b staining (right) in HL60 cells with shControl and shBRD9 after all-trans retinoic acid (ATRA) treatment (10−6M) for 6 days (*n* = 3 independent experiments). Bar: 10 μm. **f** The proportion of CD71+CD235a+ cells in K562 cells with *BRD9* KO after hemin treatment for 3 days. Two sgRNAs and three independent single cell clones per sgRNA were used. **g** Frequency of B220+ (left) and CD11b+Gr1+ (right) cells in GFP+ donor-derived peripheral blood cells in the transplant model of normal BM cells transduced with shRen (shControl, *n* = 6 independent samples), shBrd9 (*n* = 6 independent samples), empty vector (EV, *n* = 3 independent samples), BRD9-WT (*n* = 4 independent samples), dBD (*n* = 6 independent samples), and N216A mutants (*n* = 5 independent samples). Error bars, means ± s.e.m. **h** Heatmap of *Brd9* mRNA expression evaluated quantitative RT-PCR in each hematopoietic stem and progenitor cells (HSPCs) and mature cells. The relative expression levels against that of LSK are shown. Two-tailed Student's *t* test. Source data are provided as a Source Data file.

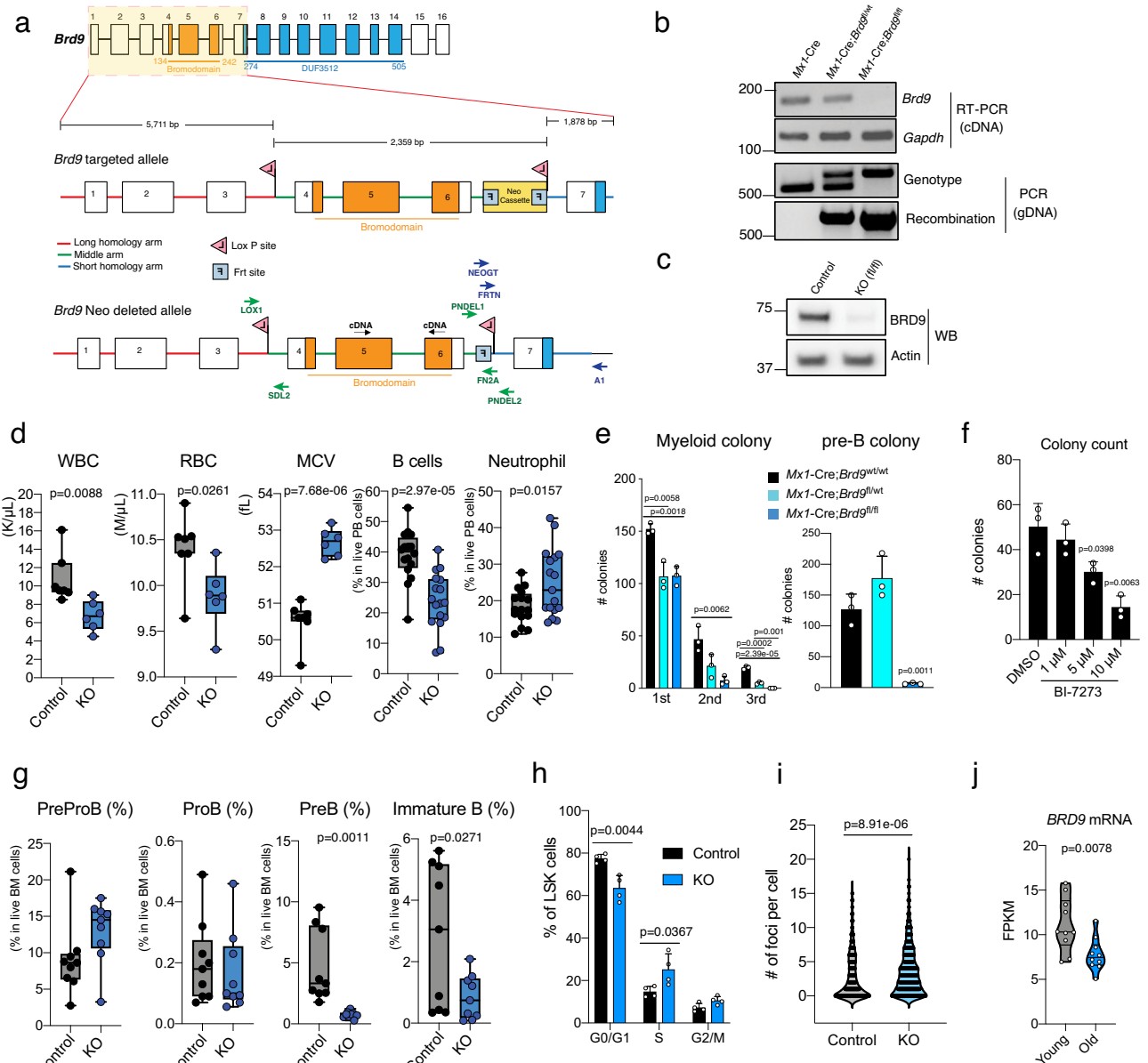

**Fig. 2 | Generation and characterization of *Brd9* knockout murine model.**
**a** Target scheme for *Brd9* conditional knockout (cKO) mice. *Brd9* targeted alle and *Brd9* Neo Cassette deleted allele are indicated. Primers were designed for confirming LoxP insertion, Neo Cassette deletion, and genotyping. **b** Successful deletion and recombination of *Brd9* in the cKO bone marrow (BM) 4 weeks post-pIpC injections. PCR by using cDNA and genomic DNA from pIpC-treated mice bone marrow cells to validate *Brd9* exon 4–6 deletion (Representative images of *n* = 3 independent experiments). Primers PNDEL1/PNEDL2 and LOX1/PNDEL2 were used for genotype and recombination, respectively. **c** Western blot for murine BRD9 in whole BM cells derived from pIpCed *Mx1*-Cre;*Brd9*^WT/WT (Control) mice and *Mx1*-Cre;*Brd9*^fl/fl mice cKO (fl/fl) mice (Representative images of *n* = 3 independent experiments). **d** *Brd9* cKO mice developed a cytopenia phenotype with macrocytosis 3 months post-pIpC. Whole blood counts of white blood cells (WBCs), red blood cells (RBCs), and mean corpuscular volume (MCV) and frequency of B cells and neutrophils in *Mx1*-Cre;*Brd9*^WT/WT (Control) mice and *Mx1*-Cre;*Brd9*^fl/fl mice cKO (KO) mice. *n* = 6 each in WBC, RBC, and MCV; *n* = 16 (Control) and *n* = 17 (KO) in B cells and Neutrophils, both males and females. For box and whiskers plots throughout, bar indicates median, box edges first and third quartile values, and whisker edges minimum and maximum values. *p* value relative to control by a two-sided *t*-test. **e** Colony formation of whole BM cells. Colonies were scored 10 days

after plating in methylcellulose in M3434 and M3630, for myeloid colonies and pre-B colonies, respectively. *n* = 3 independent experiments; mean and s.e.m are plotted. Statistical significance was calculated with two-tailed unpaired *t*-test. **f** Colony formation of wildtype whole BM cells treated with DMSO and BI-7273 (BRD9 bromodomain inhibitor) for 24 h, respectively. Colonies were scored 10 days after plating in methylcellulose M3630 for Pre-B colonies. *n* = 3 independent experiments; mean and s.e.m are plotted. *p* value relative to control by a two-sided *t*-test. **g** Frequency of PreProB, ProB, PreB, and Immature B cells in BM of 24-week-old *Mx1*-Cre;*Brd9*^WT/WT (Control) mice and *Mx1*-Cre;*Brd9*^fl/fl mice cKO (KO) mice. *n* = 9 each, both males and females. For box and whiskers plots, bar indicates median, box edges first and third quartile values, and whisker edges minimum and maximum values. *p* value relative to control by a two-sided *t*-test. **h** Percentages of bromodeoxyuridine (BrdU)⁺ (S), DAPI⁺BrdU⁻ (G2/M), DAPI-BrdU- (G0/G1) LSK cells in the BM of primary 12-week-old *Brd9* KO and control mice. *n* = 4 independent experiments, males; mean and s.e.m are plotted. *p* value relative to control by a two-sided *t*-test. **i** The number of foci in gH2AX (a marker of DNA damage)-staining of *Brd9* KO and control Lin⁻Kit⁺ cells. *p* values were calculated relative to the control group by two-sided *t*-tests and are indicated in the figures. **j** *BRD9* mRNA levels in young and old human Lin⁻CD34⁺CD38⁻ cells (*n* = 10 per age group). *p* value relative to control by a two-sided *t*-test. Source data are provided as a Source Data file.

necessary for normal and healthy hematopoietic development and differentiation.

## BRD9 loss transcriptionally alters the cell fate specification of HSCs in vivo

Given the broad, multi-lineage disturbance of BRD9 loss in the hematopoietic system, we hypothesized that the defects originate from HSPCs. We first examined the various HSPC compartments in *Brd9* KO mice using cell surface markers into several HSCs and multipotent progenitors (MPPs). Moreover, MPPs, which all have short-term stem cell activity, can be further subdivided into megakaryocytic and erythroid-biased MPP2 (CD135⁻CD150⁺CD48⁺ LSK), myeloid-biased MPP3 (CD135⁻CD150⁻CD48⁺ LSK), and lymphoid-biased MPP4 (CD135⁺ LSK)[39–41]. Of note, the number of LSK and myeloid-biased MPP3 was significantly increased upon *Brd9* KO (Fig. 3a and Supplementary Fig. 4a). However, despite expanded LSK fraction, we observed a profound reduction of immunophenotypic HSCs, termed long-term HSCs (LT-HSC, CD150⁺CD48⁻ LSK) (Fig. 3a). In the more lineage-committed fraction, CMPs were increased while MEPs and CLPs (Lin⁻Sca1ⁱⁿᵗc-KitⁱⁿᵗCD127⁺CD135⁺) tended to be reduced (Fig. 3a and Supplementary Fig. 4a). Based on the expression levels of c-Kit and Ly6G, the frequency of subpopulation II-IV, which comprises promyelocytes, myelocytes, and metamyelocytes, respectively, was significantly increased in the *Brd9* KO group, suggesting the expansion of myeloid-lineage committed cells (Supplementary Fig. 4b)[42].

We, therefore, sought to evaluate the altered transcriptional program in HSPCs. RNA-seq analysis of FACS-purified BM Lin⁻c-Kit⁺ cells of 4-month-old *Brd9*ᶠˡ/ᶠˡ and *Mx1*-Cre;*Brd9*ᶠˡ/ᶠˡ mice was performed eight weeks after pIpC injection in biological triplicate for each model. Transcriptomes of Lin⁻c-Kit⁺ cells from each model are substantially different from the others (Fig. 3b, Supplementary Fig. 4c, d and Supplementary Data 2). Of note, the number of the genes upregulated in KO group was larger than those downregulated (Fig. 3c, Supplementary Fig. 4e and Supplementary Data 2) (744 genes vs. 277 genes, applying adjusted $p < 0.1$ and $|\log_2 FC| > \log_2(1.5)$). Gene ontology and gene set enrichment analysis (GSEA) of KO versus control RNA-seq results revealed the positive enrichment for myeloid differentiation, GMP signature, myeloid leukocyte mediated immunity, and the negative enrichment for MYC targets, heme biosynthetic process, mitochondrial oxidative phosphorylation, and eukaryotic translation (Fig. 3d, Supplementary Fig. 4f, g and Supplementary Data 3). All of these are compatible with our phenotypic observations in Fig. 2. Next, to decipher the effect of *Brd9* deletion on lineage commitment identity and priming, we performed single-cell RNA-seq (scRNA-seq) targeting 10,000 Lin⁻c-Kit⁺ cells from control *Brd9*ᶠˡ/ᶠˡ and *Mx1*-Cre;*Brd9*ᶠˡ/ᶠˡ mice, at 8 weeks post-deletion. Uniform Manifold Approximation and Projection (UMAP) analysis integrating both models revealed the expected stem and progenitor clusters, including LT-HSC, MPP, and erythroid, myeloid, and lymphoid progenitors described previously in a series of scRNA-seq studies[43–46]. Interestingly, in *Brd9* KO mice, the frequency of transcriptionally-defined LT-HSCs and MEPs was reduced, while the MPP3 population was increased (Fig. 3e and Supplementary Fig. 4h). Such myeloid-lineage skewing was supported by data demonstrating more and less cell-cycling features in MPP3 and MEP, respectively (Supplementary Fig. 4i). Indeed, in maturation PC analysis[47], increased and decreased expression of early myeloid and erythroid commitment signatures were observed, respectively (Fig. 3f and Supplementary Fig. 5a). Consistent with Fig. 3a, scRNA-seq data indicated reduced and increased proportion of LT-HSC and myeloid-committed MPP3, respectively (Supplementary Fig. 4h), although MPP4 frequency was unchanged.

We next sought to evaluate further the individual gene expression essential for the lineage specification in each HSPC population at the single-cell level. Concordant with the severe B-cell phenotype, expression of *Il7r* and *Dntt*, essential for immunoglobulin

recombination and rearrangement during lymphocyte development[48,49] was strikingly down-regulated in estimated MPP4 fraction by scRNA-seq (Fig. 3g and Supplementary Data 4) as well as FACS-sorted MPP4 (Supplementary Fig. 5b). In line with these findings, we observed negative enrichment of transcriptional factor (TF) motifs essential for normal B cell development, including SPIB, PAX5, EBF1, and IKZF1 in MPP4, in stark contrast to a positive enrichment of TF motifs, such as CEBP families and SPI1, in myeloid-biased *Brd9* KO progenitors (Fig. 3h and Supplementary Data 4). Similarly, erythroid lineage TFs were negatively enriched in KO progenitors (Supplementary Fig. 5c and Supplementary Data 4)[50,51]. Altogether, these data suggest that BRD9 plays a pivotal role in dictating cell identity and lineage commitment in HSCs.

To correlate gene expression changes of *Brd9* KO HSPCs with chromatin accessibility and enhancers, we undertook comprehensive genome-wide mapping by performing ChIP-seq for H3K27ac in purified Lin⁻c-Kit⁺ cells, which marks active enhancers[52], classified into super enhancers (SEs) and typical enhancers (TEs) (Fig. 3i and Supplementary Fig. 5d, e, g). Among the increased enhancers in KO cells, some SEs were associated with well-known myeloid-lineage-associated genes such as Pu.1, Myb, and Etv6[53–55]. Since SEs identified in various cell types are located at chromatin-accessible regions[56,57], we next performed ATAC-seq (assay for transposase-accessible chromatin through sequencing) in the same Lin⁻c-Kit⁺ fraction. As expected, the ATAC signal was remarkably higher in KO samples at SEs than at TEs (Fig. 3j and Supplementary Fig. 5f). Interestingly, the ATAC signal of SEs/TEs and enhancer length were significantly promoted in *Brd9* KO compared with controls in biologically replicated experiments (Fig. 3j, Supplementary Fig. 5e, f and Supplementary Data 5). Next, HOMER motif enrichment analysis at regions with increased H3K27ac was performed to identify the TFs related to the changes in dynamic enhancer signaling. Of note, we found the gain of accessibility for genes with CTCF/CTCFL, CEBP, and PU.1 (ETS) motifs (Fig. 3k) in BRD9-depleted cells. In contrast, the motifs of YY1, essential for all stages of B-cell differentiation[58], were enriched in control (Fig. 3k). These data suggest that BRD9 plays a pivotal role in regulating chromatin structure and that its loss dysregulates normal differentiation of HSCs.

## BRD9 loss disturbs the cell fate of HSCs in a cell-autonomous fashion

To examine if the effects of BRD9 loss result from cell-autonomous effects in HSCs and if the effects are dose-dependent on BRD9, we next performed BM transplant of control *Mx1*-Cre, *Mx1*-Cre;*Brd9*ᶠˡ/ʷᵗ, *Mx1*-Cre;*Brd9*ᶠˡ/ᶠˡ into lethally irradiated CD45.1 recipient mice (Fig. 4a). This approach enables evaluation of hematopoietic reconstitution starting from whole bone marrow cells. After confirming successful engraftment four weeks after transplant, we treated recipient mice with pIpC. As observed in primary mice, four months after pIpC homozygous KO recipients exhibited significant leukopenia caused by B cell depletion, macrocytosis of erythrocytes, and striking myeloid lineage skewing compared with the control group (Fig. 4b and Supplementary Fig. 6a). Notably, these events occurred in a BRD9 dose-dependent manner, suggesting the direct role of *Brd9* depletion (Fig. 4c). The *Mx1*-Cre;*Brd9*ᶠˡ/ᶠˡ recipient mice developed MDS-like phenotype with morphological abnormalities and hastened death compared to controls (Fig. 4d and Supplementary Fig. 6b). The mice in the KO group exhibited consistent KO efficiency (Supplementary Fig. 6c), striking anemia (Supplementary Fig. 6d), and myeloid lineage skewing with dysplastic features in BM and spleen (Supplementary Fig. 6d–f) without leukemic transformation. In line with the phenotype shown in Figs. 2 and 3, a significant expansion of GMP and myeloid-biased MPP3, reduction of lymphoid-biased MPP4 (Supplementary Fig. 6g), and the differentiation block at the level of metamyelocytes (group IV) (Supplementary Fig. 6h) was observed. Moreover, histological analysis of

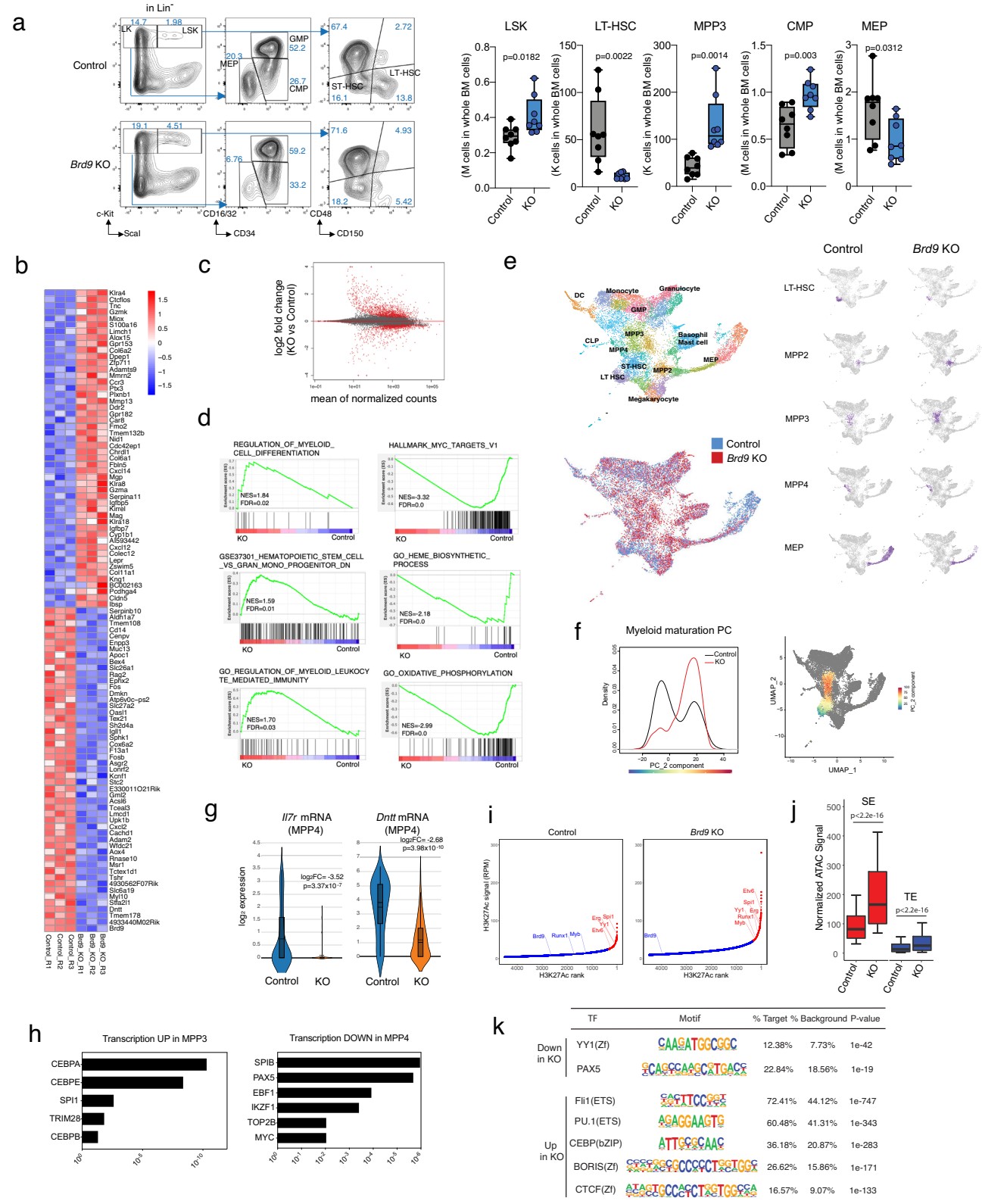

the spleen exhibited destruction of normal spleen architecture with the infiltration of CD11b[+] cells, occupying a large proportion of splenic cells (Supplementary Fig. 6i). The primary mice also developed significant myelodysplasia (observation period <12 months); however, the transplanted mice died earlier, suggesting that prolonged stress condition under a transplant setting may enhance the phenotype of Brd9 depletion.

Although detailed analysis of primary KO mice demonstrated impairment of the HSPC program and lineage commitment, analyses of primary models cannot evaluate cell-autonomous effects of *Brd9*-depleted HSCs compared to normal HSCs. We, therefore, examined in vivo self-renewal of control *Mx1*-Cre, *Mx1*-Cre;*Brd9*[fl/wt], *Mx1*-Cre;*Brd9*[fl/fl] mice model in a competitive transplantation experiment (Fig. 4e). Equal numbers of CD45.2[+] BM cells from each model were mixed with

**Fig. 3 | BRD9 loss transcriptionally alters the cell fate specification of HSCs in vivo. a** FACS analysis and gating strategy of BM cells from representative primary mice for evaluating the ratio of stem and progenitor fraction. Box-and-whisker plots of numbers of LSK, LT-HSCs, MPP3, CMPs, and MEPs in BM of primary 6-month-old $Brd9^{fl/fl}$ (Control) and $Mx1$-Cre;$Brd9^{fl/fl}$ (KO). $n = 8$ independent samples, both males and females; error bars, means ± s.e.m. For box and whiskers plots, bar indicates median, box edges first and third quartile values, and whisker edges minimum and maximum values. $p$ value relative to control by a two-sided $t$-test. **b** Heatmap showing the top 50 up- or down-regulated genes evaluated by RNA-seq in BRD9 KO HSPCs. **c** MA plot showing differentially expressed genes (adjusted $p$ value < 0.1). $p$ values were obtained by applying Wald test to absolute values of $\log_2$FC and adjusted using the Benjamini-Hochberg correction. **d** GSEA enrichment plot for dysregulated genes in RNA-seq of KO vs. Control. **e** Identification of different hematopoietic clusters in Control and KO Lin⁻cKit⁺ cells base on UMAP analysis from single-cell RNA sequencing (scRNA-seq). The estimated fractions of stem, progenitor, and mature cells are labeled and highlighted. **f** Cellular densities along Myeloid maturation PC and the scaled values of the maturation PC visualized on UMAP space. **g** The violin plots of $Il7r$ and $Dntt$ mRNA expression in the indicated cluster, where 158 and 148 cells belong to Control and KO, respectively. Box plot and kernel density plot of $\log_2$ expression values are shown. $p$ values were obtained by negative binomial test and adjusted using the Benjamini-Hochberg correction. In the box-and-whisker plots, the 0th, 25th, 50th, 75th and 100th percentiles and mean (dashed lines) are shown. **h** Transcriptional factor motif analysis of genes upregulated in MPP3 (left) and downregulated in MPP4 (right) of KO mice. $p$ values are indicated and generated via Enricher. **i** Enhancers ranked by H3K27ac ChIP-seq signals in HSPCs of Control and KO mice. SEs (red) and TEs (blue) were identified using HOMER software. **j** Normalized ATAC-seq signal (RPGC) at SE and TE (detected with HOMER fdr <0.001) in HSPCs. One biologically independent sample in each group was used. Results of additional biologically independent samples are shown in Supplementary Fig. 5f. The $p$ values were obtained by two-sided Wilcoxon rank sum test. In the box-and-whisker plots, the 10th, 25th, 50th, 75th and 90th percentiles are shown. **k** Transcriptional factor motif analysis of typical enhancers (TE) by comparing TE peaks between Control and KO HSPCs. Source data are provided as a Source Data file.

CD45.1⁺ wild-type BM cells and injected into lethally irradiated CD45.1 recipient mice. After DNA recombination by pIpC, we followed the donor- and competitor-derived chimerism in PB every month and euthanized all of the recipients five months later to examine the chimerism in BM and spleen. In stark contrast to the profound loss of hematopoiesis in $Mx1$-Cre;$Brd9^{fl/fl}$ group in PB and spleen, we detected comparable chimerism in the BM (Fig. 4f–h). Notably, the competitive disadvantage in PB and spleen was primarily attributed to the near-complete loss of $Brd9$ KO cells in B220⁺ and CD3⁺ fractions (Fig. 4g, h). On the other hand, the chimerism in CD11b⁺Gr1⁺ and CD11b⁺Gr1⁻ fraction was preserved even in the homozygous KO model. In each HSPC population of BM cells, we observed modest expansion of $Brd9$ KO chimerism (%), especially in LSK, MPP (CD150⁻CD48⁺ LSK), CMP, and GMP, and indeed the significantly larger number of the donor-derived LSK and MPP in the $Brd9$ KO group (Fig. 4f and Supplementary Fig. 6j). Of interest, the proportion of LT-HSC was not affected (Fig. 4f) despite profound reduction in the primary model (Fig. 3), suggesting more dormant LT-HSC can reconstitute the HSPC fraction in the transplant setting. Again, when restricted to CD45.2⁺ cells, the donor-derived cells exhibited remarkable myeloid lineage skewing while impairing B cells (Supplementary Fig. 6k). Although the transplant model provided information on the impact of Brd9 depletion on hematopoietic reconstitution by HSCs, some of the results were discrepant with those of primary mice, such as progenitor distribution and CD3⁺ fractions (Fig. 3 and Supplementary Fig. 2). Nevertheless, the cell-autonomous effects of $Brd9$ loss in the development of myeloid and B cells were consistent.

## Genomic distribution of BRD9 in HSPCs

Although several reports demonstrated that BRD9 and ncBAF have characteristic genomic distribution such as CTCF and BRD4 sites[8,9,16,59,60], this has not been evaluated in primary mouse tissue or HSPCs. We therefore performed ChIP-seq in Lin⁻c-Kit⁺ cells of control and KO mice, for BRD9, pan-BAF subunit BRG1, CTCF, BRD4, and the histone marks H3K4me1, H3K4me3, and H3K27ac. Consistent with previous reports in cell lines, BRD9 peaks comprised subsets of BRG1 and CTCF peaks in control hematopoietic precursors and were located mainly at promoters (Fig. 5a and Supplementary Fig. 7a–c). Regardless of the presence of BRG1, BRD9 peaks were characterized by the enrichment at CTCF colocalized sites and at active promoters (H3K27ac and H3K4me3) rather than enhancers (H3K27ac and H3K4me1)[8] (Fig. 5b and Supplementary Fig. 7d, e). These results were in contrast to those of BRD9⁻BRG1⁺ peaks corresponding to BAFs other than ncBAF. Indeed, BRD9⁺BRG1⁺ and BRD9⁺ peaks highly overlapped with CTCF peaks (57.6% and 55.2%, respectively), contrary to that of BRD9⁻BRG1⁺ peaks (17.6%) (Supplementary Fig. 7c). Given the similar genomic distribution of BRD9⁺BRG1⁺ and BRD9⁺ peaks, BRD9

localization is determined mainly via ncBAF and distinct from that of other BAFs.

## BRD9 loss enhances CTCF localization resulting in active transcription

Next, we examined genome-wide correlations between BRD9, pan-BAF BRG1, CTCF, BRD4, and histone mark ChIP-seq signals (Fig. 5c). Considering no correlation between BRG1 and CTCF, the association of CTCF and ncBAF was largely dependent on BRD9, which was confirmed by the immunoprecipitation studies revealing direct interaction between BRD9 and CTCF (Supplementary Fig. 7f), as previously reported[61]. In addition to the well-defined BRD9-BRG1 and BRD9-CTCF correlation[8,16], an association of BRD9 with BRD4, H3K4me3, and H3K27ac was observed. Thus, we next evaluated whether BRG1, CTCF, and BRD4 peaks are dependent on BRD9 or whether these peaks are enhanced or diminished by BRD9 loss genome-wide. Of note, CTCF signals increased upon BRD9 depletion while BRD9 loss has a lesser impact on BRG1 and BRD4 binding (Fig. 5d, e, Supplementary Fig. 7g and Supplementary Data 6). BRD9 depletion did not alter the mRNA or protein expression of $Ctcf$/CTCF or cohesin subunit ($Stag1$, $Stag2$, $Smc3$, $Smc1a$, and $Rad21$) mRNA (Supplementary Fig. 7h, i), excluding the possibility that CTCF enrichment after BRD9 loss is derived from transcriptional regulation. To further characterize the CTCF peaks enhanced upon BRD9 depletion, we next classified 51,135 CTCF peaks detected in control or KO samples into three groups based on the behavior of CTCF peaks upon BRD9 loss: Up, 21,593 sites; Neutral, 27,772 sites; Down, 1770 sites (Fig. 5f). Intriguingly, CTCF enrichment after BRD9 loss occurred at relatively modest CTCF peaks in control cells compared to CTCF Neutral group. Given that the greater proportion of promoter-located CTCF peaks was observed in Up/Neutral groups (Fig. 5f), we therefore focused on the association of promoter-located CTCF peaks (Up, 2837 peaks; Neutral, 3280 peaks; Down, 105 peaks) with transcriptional regulation. Notably, promoter peaks in Up group were associated with enhanced transcription (mean $\log_2$FC = 0.03648238, $p = 1.238e{-}08$), in contrast to Neutral/Down promoter peaks (mean $\log_2$FC = 0.01032778, $p = 0.06398$ and mean $\log_2$FC = 0.01327493, $p = 0.7486$, respectively) (Fig. 5g, Supplementary Fig. 7j and Supplementary Data 7). We found that enhanced CTCF peaks are enriched in promoter regions compared to weakened CTCF peaks on BRD9 loss ($p$ value < 2.2e{-}16) with robust change of observed proportion of the peaks against the expected ones in promoters ($\log_2$Ratio = 3.566) (Supplementary Fig. 7k). Indeed, such CTCF peaks in Up group showed enhanced active chromatin marks, H3K4me3 and H3K27ac signals ($p < 2.2e{-}16$ and $p < 2.2e{-}16$, respectively) (Fig. 5h). Moreover, we observed that the gene names given to peaks in Up group were significantly enriched for biological processes involved in neutrophil activation, myeloid differentiation, cell cycle progression,

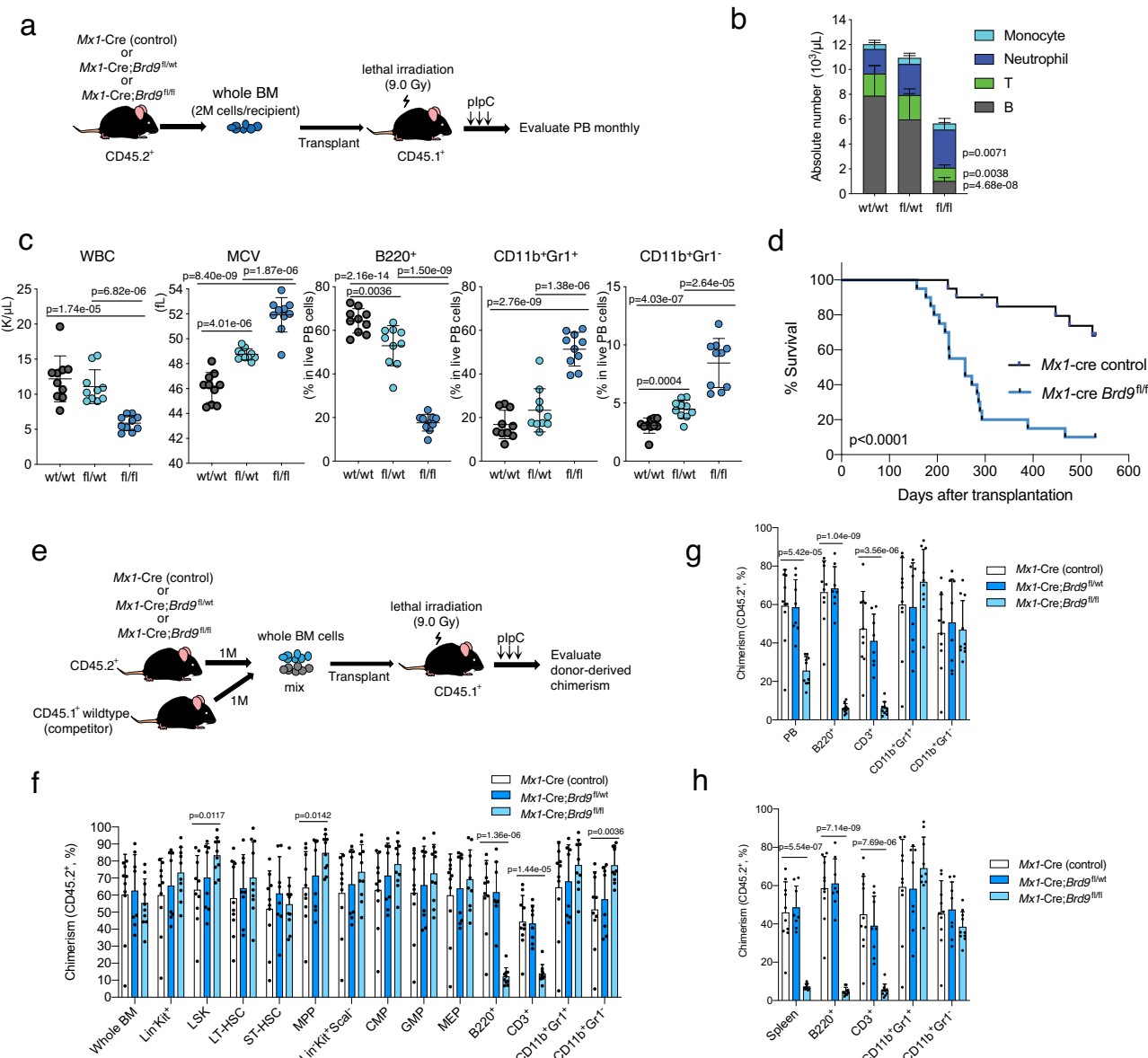

**Fig. 4 | BRD9 loss disturbs the cell fate of HSCs in a cell-autonomous fashion.** **a** Schema of non-competitive BM transplantation assays. **b** Absolute number of B220+, CD3+, CD11b+Gr1+, CD11b+Gr1- cells in the PB of each group. *n* = 10 independent samples; mean and s.e.m are plotted. *p* value relative to control by a two-sided *t*-test. **c** Plots of WBC, MCV, and the proportion of B220+, CD11b+Gr1+, and CD11b+Gr1- cells in PB of recipient mice. *n* = 10 independent samples; error bars, means ± s.e.m. Statistical significance was calculated with two-tailed unpaired *t*-test. **d** Kaplan-Meier survival of each recipient group. *p* value was calculated by log-rank test. **e** Schema of competitive BM transplantation assays. Percent of donor-derived (CD45.2+) cells of each indicated population in BM (**f**), PB (**g**), and spleen (**h**) following 5 months of transplantation. *n* = 10 independent samples in *Mx1-cre control* and *Mx1-cre;Brd9fl/fl* group, *n* = 8 independent samples in *Mx1-cre;Brd9fl/wt* group (both males and females). Mean and s.e.m are plotted. *p* values were calculated by two-sided *t*-test. Source data are provided as a Source Data file.

and DNA damage response (Fig. 5i and Supplementary Data 8), all of which are compatible with the HSPC's phenotype observed in Figs. 2 and 3. Even with short-term (four days) depletion of BRD9 in K562 cells with dBRD9[62], we observed differential gene expression (Supplementary Fig. 8a, b) and confirmed the increment of CTCF peaks (Supplementary Fig. 8c) and significant CTCF-motif enrichment in the opened chromatin sites (Supplementary Fig. 8d), indicating that the alteration of chromatin states and CTCF localization occur as a primary or direct event after BRD9 depletion. Finally, we extended our knowledge to the relationship among BRD9-CTCF-YY1 since YY1 is a chromatin structural regulator[63] and the chromatin accessibility around YY1 binding motifs decreased after BRD9 KO (Fig. 3k). By integrating our data with anti-YY1 ChIP-seq in Lin- wildtype BM cells[64], we identified that YY1 peaks significantly overlapped with BRD9 peaks, not with CTCF peaks and

that CTCF-upregulated sites by BRD9 KO do not harbor YY1 binding (Supplementary Fig. 8e, f). Altogether, these findings indicate that the CTCF enrichment after BRD9 depletion occurs selectively and enhanced CTCF peaks at gene promoters results in active transcription for the observed phenotype.

## BRD9 loss did not change the formation of TADs or compartmentalization

In addition to the essential role of CTCF in maintaining TADs, several lines of evidence suggest that CTCF plays a critical role in chromatin looping regulation as well[65–68]. Therefore, we next investigated how CTCF-bound TADs and chromatin-chromatin interactions were dysregulated after BRD9 depletion by performing HiC analysis in BM Lin-c-Kit+ cells from control and *Brd9* KO mice. While chromatin loops

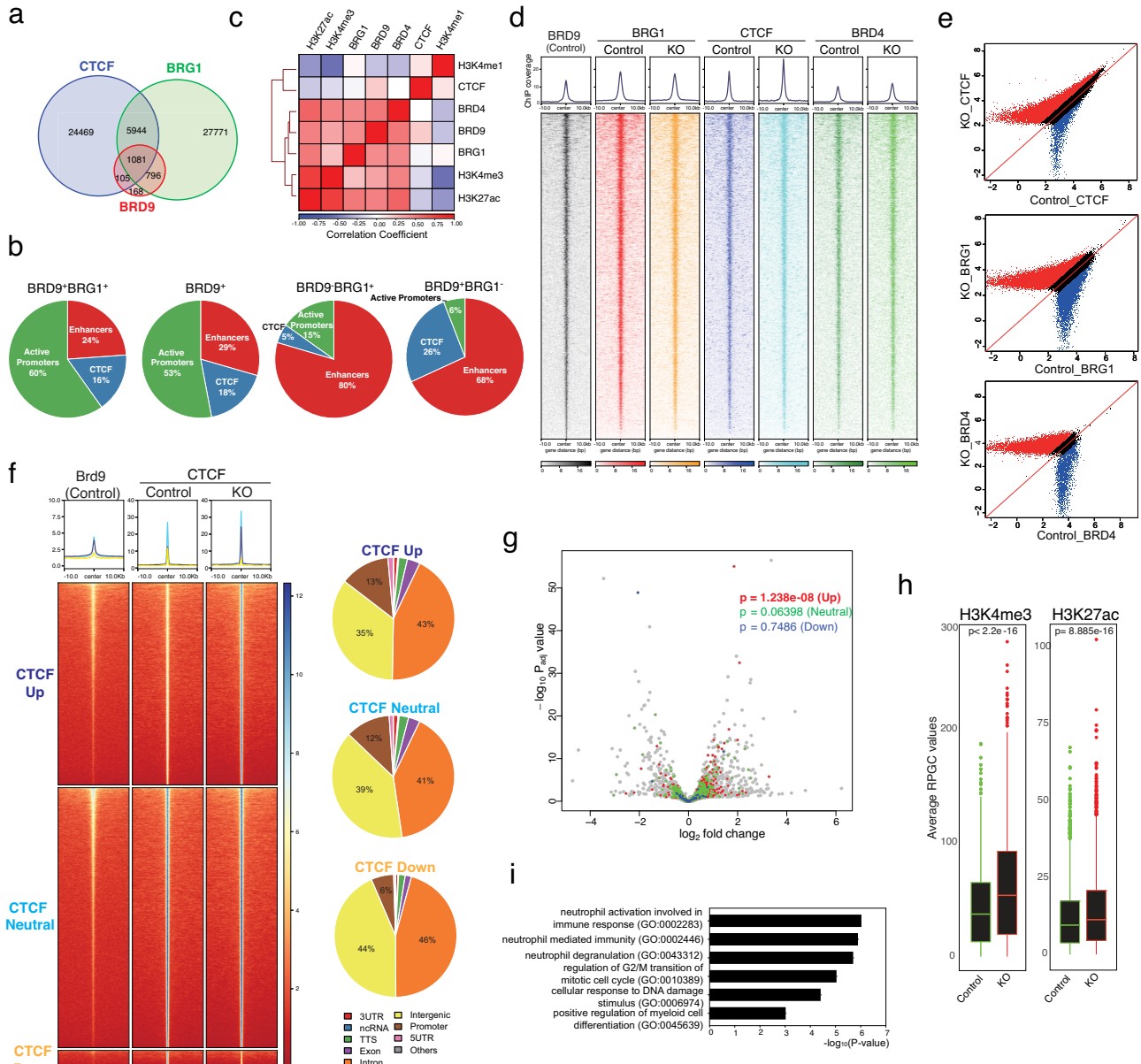

**Fig. 5 | BRD9 loss enhances CTCF localization resulting in active transcription.** **a** Venn diagram of peaks from BRD9, BRG1, and CTCF evaluated by ChIP-seq experiments. **b** The proportion of BRD9⁺BRG1⁺(ncBAF), BRD9⁺ (ncBAF + ncBAF-independent BRD9 targets), BRD9⁻BRG1⁺ (BAFs other than ncBAF) and BRD9⁺BRG1⁻ (ncBAF-independent BRD9 targets) peaks at CTCF, active promoters and enhancers. **c** Heatmap representing the correlations between average ChIP-seq reads (log$_2$[reads per genome coverage (RPGC)]) over a merged set of BRD9, BRG1, CTCF, BRD4, H3K4me1, H3K4me3, and H3K27ac peaks. **d** The average ChIP enrichment profile of BRD9, BRG1, CTCF, and BRD4. Heatmap illustrating the ChIP-seq signal 10 kb up and downstream at BRD9 peaks ($q < 0.05$). **e** Scatterplot of log$_2$ average RPGC values of CTCF, BRG1, and BRD4 in control vs. KO condition (Red and Blue: ratio of average RPGC values > 1.5, Black: otherwise). **f** The average ChIP enrichment profile of BRD9 and CTCF (top) over the peak regions in three groups: CTCF Up (dark blue, fold change >1.5), CTCF Neutral (light blue, 1.5 ≥ fold change ≥ 1/1.5), and CTCF Down (yellow, fold change <1/1.5) in KO compared to Control. Heatmap illustrating the ChIP-seq signal 10 kb up and downstream. The genomic distribution of each group is shown on the right. **g** Volcano plot of DEGs in *Brd9* KO HSPCs. Genes whose promoter-TSS sites locate on CTCF peaks in CTCF Up (red), CTCF Neutral (green), CTCF Down (blue) are highlighted and *p* values of two-sided *t*-test were indicated. **h** Correlation between the active histone marks (H3K4me3, left; H3K27ac, right) and the promoter-located CTCF peaks of CTCF Up group in Control and KO conditions. One biologically independent sample in each group was used. *p* values of two-sided *t*-test were indicated. In the box-and-whisker plots, the lower and upper hinges correspond to the 25th and 75th percentiles. The whiskers extend from the hinges to the values no further than 1.5 x the inter-quartile ranges. Data points beyond the whiskers are shown as dot plots. **i** GO terms enrichment of the promoter-located CTCF peaks of CTCF Up group. *p* values are generated via Enricher. Source data are provided as a Source Data file.

are cell type- and developmental stage-specific[67,69], the global arrangement of active (A) and inactive (B) compartments and TAD formation are highly conserved between distinct cells and even between humans and mice[65]. We therefore sought to evaluate whether BRD9 loss was involved in maintaining large-scale chromatin structures. We created Omni-C HiC libraries in three biological replicates for

each Control and KO cells and merged the dereplicated read-pairs of all three biological replicates into a single dataset in each Control and KO cells after evaluating the reproducibility between the biological replicates (Supplementary Figs. 9 and 10, "Methods" in the detail). First, we investigated active (A) and inactive (B) compartments defined by eigenvector. The eigenvector computed by juicer[67], e.g., the first

eigenvector (principal component) of the correlation matrix of normalized chromatin contacts (binned at 10 kb resolution), clusters the binned genomic regions according to the enrichment profiles of chromatin contacts of a genomic region with surrounding regions, and splits genomic regions into A and B compartments in terms of chromatin contacts. The global arrangement of A/B compartments was unchanged in KO cells (Fig. 6a), where active (A) compartments coincided remarkably with transcribed regions mapped by H3K27ac peaks and supported by RNA-seq (Supplementary Fig. 11a). Next, we analyzed TAD formation based on insulation scores. The insulation scores, i.e., the aggregates of normalized contact frequencies in a sliding window of certain size (we used 1 Mb) along the HiC matrix diagonal, measure degrees of enrichment of chromatin contacts in a given genomic region[70]. Minima and peaks of the insulation score profile denote areas of high and low insulation, e.g., boundaries and centers of TADs, respectively. We detected 3789 and 3397 TADs in control and KO cells, respectively (Supplementary Fig. 11b); 3042 TADs were common, and 747 and 355 TADs were control- and KO-specific, respectively. Of note, based on the high correlation of genome-wide eigenvector values (Pearson $R = 0.99$) (Fig. 6b) and insulation scores (Spearman $\rho = 0.98$) (Fig. 6c), we demonstrated that overall genome-wide A/B compartments and TADs structures remained unchanged even in BRD9-depleted HSPCs (Fig. 6d).

## BRD9 loss alters gene expression via chromatin loop and accessibility

In contrast to the conservation of TADs in different cell types, chromatin loop formations are far more dynamic and regulated by CTCF and other proteins, including cohesin and YY1[63]. We therefore examined the effect of BRD9 loss on CTCF-bound chromatin-chromatin interactions by using Juicer[67], and detected 2239 and 4666 chromatin loops in control and KO cells, respectively (denoted as Control loops and KO loops) (Fig. 6e). For each loop identified, Juicer reported a pair of highly interacting genomic regions of 5 kb, 10 kb, or 25 kb in size (loop anchors) and a normalized contact frequency (Knight-Ruiz balanced observed/expected interaction frequencies termed "loop intensities")[67,71]. Both anchors of the 1767 (78.1%) Control and 3836 (82.2%) KO loops overlapped with CTCF enrichment peaks (FDR < 10$^{-4}$), and 772 (34.4%) Control and 1551 (33.2%) KO loops had one or more pairs of CTCF binding motifs convergently oriented in the anchors[67]. Intriguingly, loop intensities of both Control and KO loops were mostly elevated in KO cells relative to Control (Fig. 6f and Supplementary Fig. 11c). For further analyses, we generated a default set of 5027 non-redundant loops by removing the overlapped loops in Control shown in Fig. 6e. Given the drastic increase of loops in KO cells (Supplementary Fig. 11d, 2785 KO-specific vs. 361 Control-specific) co-occurring with the genome-wide intensification of CTCF enrichment in KO cells, we next evaluated whether loop intensities are augmented with the increasing CTCF enrichment in loop anchors (measured by normalized numbers of CTCF ChIP read pairs mapped to the loop anchors) in KO cells relative to Control (KO/Control). Specifically, we observed that fold changes of loop intensities (KO/Control) raised with increasing CTCF enrichment in four groups of loops ranked from low to high in fold changes of CTCF enrichment (Fig. 6g). We observed a significantly increased transcription of genes whose promoters (±1 kb of transcription start sites) overlap anchors of loops (loop-anchored genes), and concomitance of raised contact frequencies around TSSs with upregulation of genes (Fig. 6h, i). For example, at the loop-anchored BRD9-binding gene loci, like *Igfbp7* (Fig. 6h) and *Hgfac* (Supplementary Fig. 11e), loop intensities were raised in KO cells, mRNA transcription was upregulated in KO cells, and both CTCF and H3K27ac enrichment was significant in loop anchors, which is compatible with the upregulation of genes concomitant with intensification of contact frequencies caused by increased CTCF enrichment in KO cells. The degree of upregulation of loop-anchored genes

positively correlates with the change of loop intensities from Control to KO cells (Fig. 6i), consistent with the previous report[69]. We identified significant upregulation of loop-anchored genes compared to non-loop-anchored genes (rank 0 in Fig. 6i), located outside of loops (of similar expression levels in Control and KO cells) (Fig. 6i), which demonstrates the apparent influence of the loop formation itself on gene activation in addition to the positive correlation between degree of upregulation of loop-anchored genes and the change of loop intensities from Control to KO cells (Fig. 6i). These positive correlations were also observed between the enhancer-promoter loops (EP loops) and expressed genes anchored by the EP loops (Supplementary Fig. 11f). The numbers of EP loops are 934 and 1847 in Control and KO loops, respectively. Such a positive correlation between loop intensity and CTCF enrichment, and between the expression level of loop-anchored genes and loop intensity is maintained for another set of 5024 non-redundant loops by removing 1333 overlapped loops in KO cells as shown in Supplementary Fig. 11g–i). The influence of chromatin loops on gene expression was also evaluated by comparing the loop-anchored genes and genes located outside the loops (loop-outside genes). Of 3633 loop-anchored and 5361 loop-outside genes, the number of up/down-regulated genes ($q < 0.05$) was 221/115 and 165/171, respectively (chi-square $p < 2.5e-07$). Intriguingly, the genes outside the loops were significantly downregulated than the loop-anchored genes (Supplementary Fig. 11i, Mann–Whitney $p < 0.000774$). These data demonstrated that BRD9 depletion promotes CTCF-bound chromatin loop intensities while compartments and TADs remain unchanged and that such enhanced loops positively or negatively dysregulate mRNA expression, altering the lineage commitment program.

## BRD9 loss diminishes leukemic potential in vivo via chromatin regulation

A previous report demonstrated that acute myeloid leukemia (AML) cell lines require BRD9 to sustain MYC transcription for leukemia maintenance and proliferation[24]. However, the in vivo effects of BRD9 on malignant hematopoiesis have not been evaluated to date. Thus, we next examined the impact of *Brd9* KO in the *MLL-AF9*-induced AML maintenance as well as initiation. To this end, we designed two transplant models: (1) transduction of *MLL-AF9* cDNA into control (*Brd9*$^{fl/fl}$) or BRD9 null (*Mx1-*Cre;*Brd9*$^{fl/fl}$) lineage (B220/CD3/Ter119/CD11b/Gr1)$^-$ cells followed by transplant into sublethally irradiated wildtype recipients and (2) transplant of *MLL-AF9* leukemia cells in the background of *Mx1-*Cre;*Brd9*$^{fl/fl}$ followed by PBS or pIpC injection to deplete BRD9 after the disease development (Fig. 7a). In the first strategy, we observed remarkably reduced AML burden as assessed by the frequency of GFP$^+$ AML cells and by white blood cell (WBC) counts and smaller size of spleen in the *Brd9* KO background compared to control (Fig. 7b and Supplementary Fig. 12a). As a result, the *Mx1-*Cre;*Brd9*$^{fl/fl}$ group had significantly prolonged survival (Fig. 7c). Histological and morphological analysis of BM and spleen cells in control mice were marked by hypercellularity, destruction of normal spleen architecture, and frequent immature blasts (Fig. 7d). However, this was not seen in the *Brd9* KO group (Fig. 7d). Although *MLL-AF9* cDNA transduced HSPCs have been shown to enhance self-renewal capacity in vitro, immortalization in the presence of IL-3/IL-6/SCF/EPO or with IL-7 was not observed in the *Brd9* KO background, while EPO-induced BFU-E colonies were maintained in the KO group compared to the control (Fig. 7e and Supplementary Fig. 12b). In line with these results, although almost all LK (Lin$^-$c-Kit$^+$Sca1$^-$) cells were derived from granulocyte-monocyte progenitor (GMPs; CD34$^+$FcγR$^+$LK) in *Brd9*$^{fl/fl}$ control group while a balanced distribution of LKs into megakaryocyte-erythrocyte progenitor (MEPs; CD34$^-$FcγR$^-$LK), common myeloid progenitors (CMPs; CD34$^+$FcγR$^-$LK), and GMPs occurred in *Mx1-*Cre;*Brd9*$^{fl/fl}$ group (Fig. 7f).

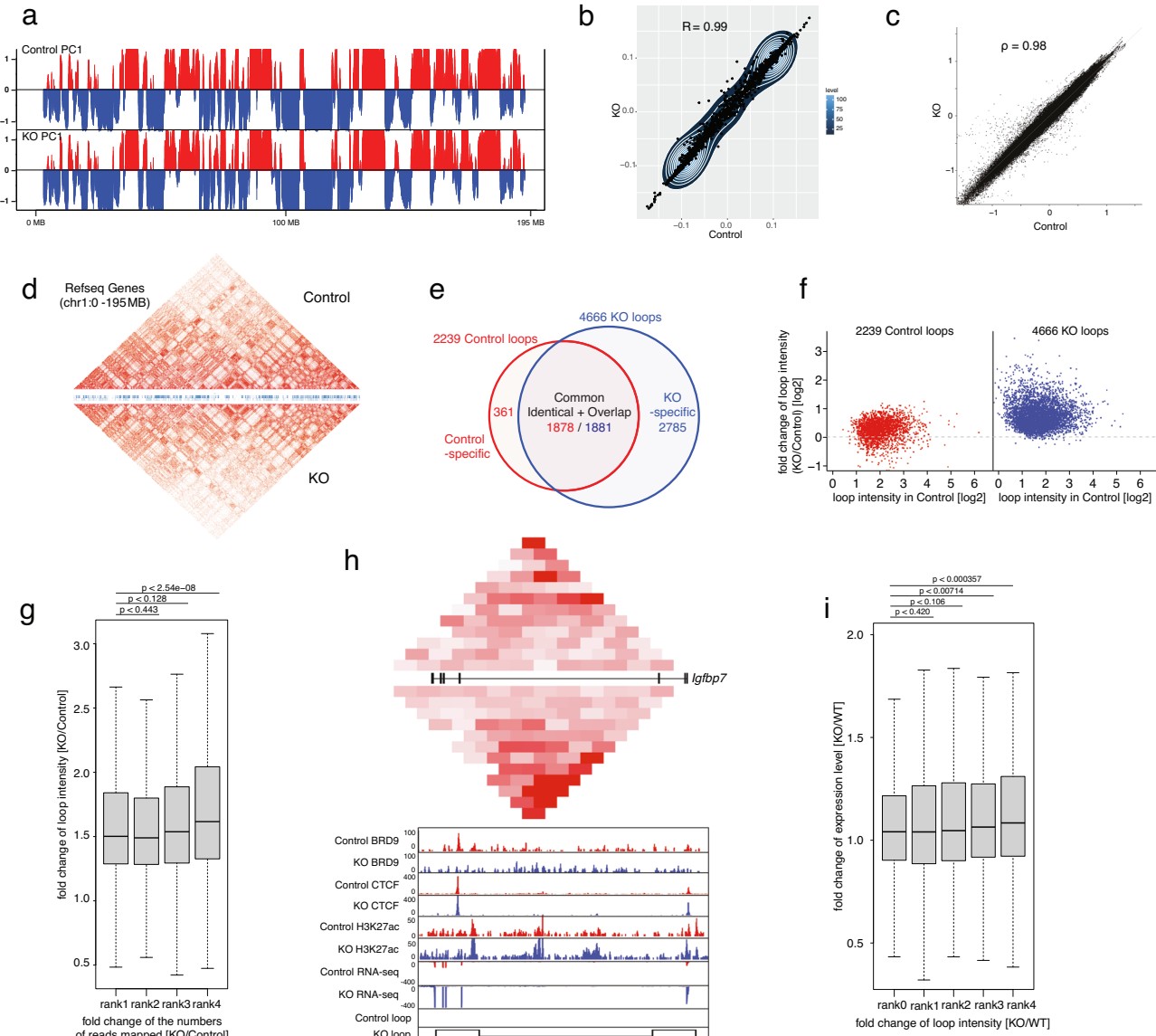

**Fig. 6 | The roles of BRD9/CTCF in chromatin three-dimensional organization.**
**a** The global arrangement of active (red) and inactive (blue) compartments on chr1 in Control (top panel) and KO (second panel) cells. **b** Genome-wide comparison of the eigenvector values between Control (*x* axis) and KO (*y* axis) cells. **c** Genome-wide comparison of the insulation scores between Control (*x* axis) and KO (*y* axis) cells. **d** Comparison of TADs between Control and KO cells. A heatmap of the Knight-Ruiz balanced observed/expected interaction frequencies (loop intensities) on chr1 as (**a**) is compared between Control (upper portion) and KO (lower portion) cells. **e** Numbers of the chromatin loops detected in Control and KO cells. The numbers of common (identical and overlapped i.e., both anchors overlap), Control-specific and KO-specific loops are shown. **f** Fold change of loop intensities from Control to KO cells of chromatin loops detected in Control (left, red) and KO (right, blue) cells. *X* and *y* axes represent the loop intensities detected in Control and the fold change of loop intensities (KO/Control), respectively. **g** Relationship between loop intensity and CTCF enrichment in loop anchors. A boxplot of fold changes of loop intensities in KO relative to Control cells (KO/Control) for four groups of loops showing distinct levels of fold change of CTCT enrichment in KO relative to Control cells (KO/Control) is shown. The 5027 loops in the default set were ranked by the fold change of CTCT enrichment from low to high in rank 1 to rank 4 (each ~1257 loops). Hereinafter, the exact *p* values computed by two-sided Mann–Whitney tests are presented, and in each boxplot, bar indicates median, box edges first and third quartile values, and whisker edges minimum and maximum values. **h** Comparison of the profiles of contact frequencies, BRD9, CTCF, H3K27ac,

and transcription around *Igfbp7* locus between Control and KO samples. The combined mapping data presents HiC interaction frequencies, ChIP-seq profiles of BRD9, CTCF, and H3K27ac, and RNA expression on *Igfbp7* locus in both Control and KO. The balanced HiC two-dimensional contact matrix is compared between Control (above) and KO sample (below) in top panel. The exon and intron structure of *Igfbp7* is given in between. The color intensity presents represents level of interaction frequency computed at 5 kb resolution. ChIP-seq and RNA-seq profiles are presented below. The positions of loop anchors are shown at the bottom. The maximum *y*-axis value of ChIP-seq or RNA-seq signal is set as indicated.
**i** Relationship between gene expression and loop intensity. A boxplot of fold changes of the expression level of one group of non-loop anchored genes and four groups of loop-anchored genes showing distinct levels of fold changes of loop intensities in KO relative to Control cells (KO/Control) is shown. In total, 1998 genes expressed (FPKM ≥ 0.5) in Control or KO cells were anchored by 1681 loops. A total of 2359 combinations of genes and loops were ranked by the fold change of loop intensities from low to high in rank 1 to rank 4 (each ~590 combinations). Rank 0 represents a group of 2402 non loop anchored genes, located outside of loops (of similar expression levels in Control and KO cells) The *p* values of rank 1 to 4 were computed against rank 0. The results here are based on biological triplicate of HiC, biological duplicate of BRD9, CTCF, and H3K27ac ChIP, and biological triplicate of RNA-seq datasets for each Control and KO sample. Source data are provided as a Source Data file.

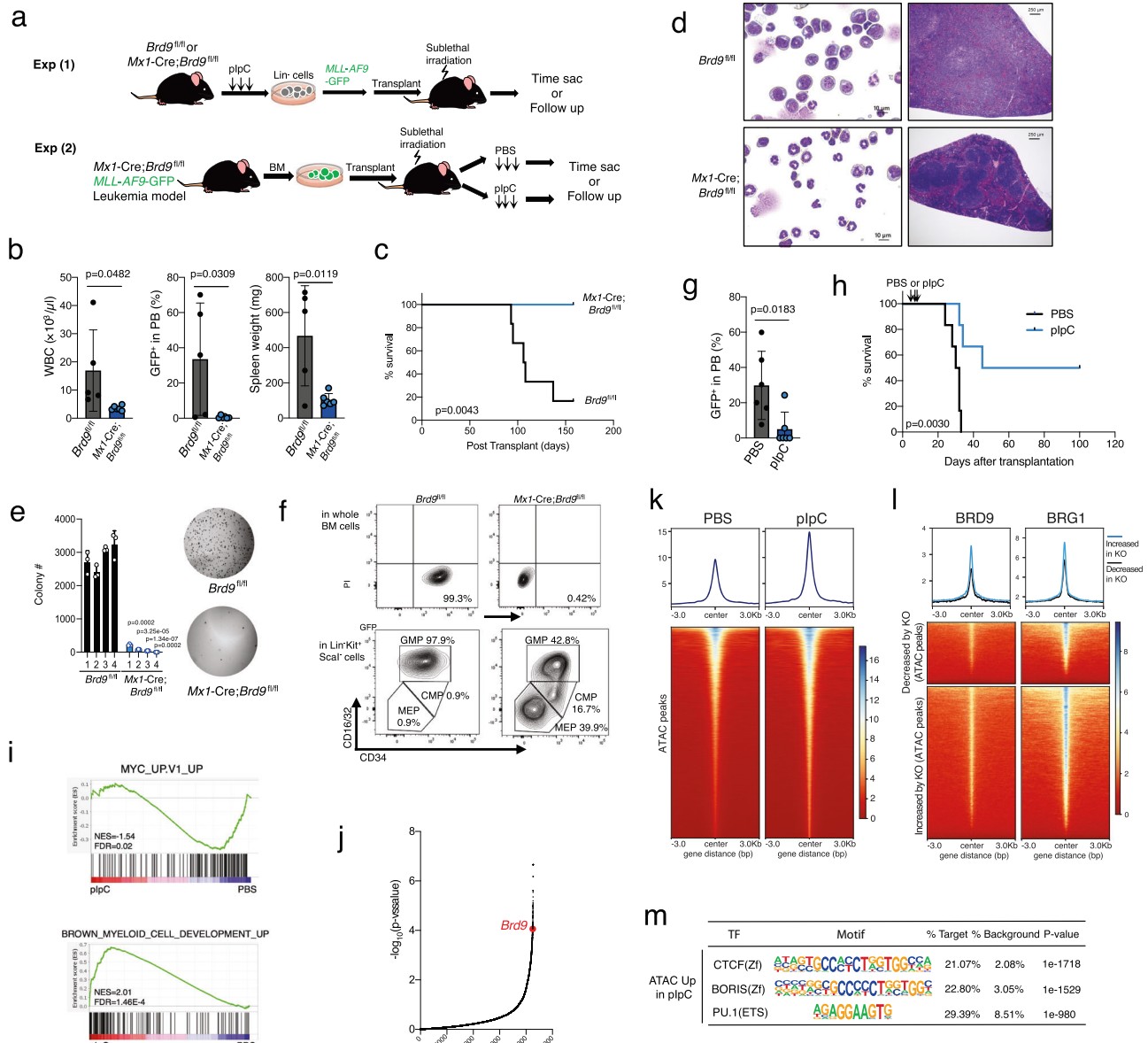

**Fig. 7 | BRD9 loss inhibited leukemia development and maintenance in vivo.**
**a** Schema of two different transplant models for the development (Experiment 1) and the maintenance (Experiment 2) of *MLL-AF9*-induced AML. **b** WBC count, GFP⁺ rate in the PB, and spleen weight of *Brd9*ᶠˡ/ᶠˡ (*n* = 5 independent samples, males) and *Mx1*-Cre;*Brd9*ᶠˡ/ᶠˡ (*n* = 6 independent samples, males) recipient mice in Experiment 1. Error bars, means ± s.e.m. *p* value relative to control by a two-sided *t*-test. **c** Kaplan–Meier survival of each recipient group. *p* value was calculated by log-rank test. **d** Representative images of the BM cytospin (left) and spleen (right). Bar: 10 μm (left), 250 μm (right). **e** The number of colonies serially replated in methylcellulose M3434 (left). *n* = 3 independent experiments; mean and s.e.m are plotted. *p* value relative to control by a two-sided *t*-test. Representative images (right) of the second colony. **f** Representative FACS plots of data of GFP⁺ cells and of GMP, CMP, and MEP in Lin⁻-Kit⁺Sca1⁻ cells. **g** GFP positive rate in the PB live cells 14 days after pIpC injection. *n* = 6 independent samples, males; Error bars, means ± s.e.m. *p* value relative to control by a two-sided *t*-test. **h** Kaplan–Meier survival of each recipient group in Experiment 2 (PBS vs. pIpC). *p* value was calculated by log-rank test. **i** GSEA enrichment plot for MYC and myeloid development-associated genes in RNA-seq of pIpC-treated group vs. PBS-treated group. **j** Rank plot for the −log₁₀ (*p* value) associated with each sgRNA in the screen using the murine AML model and negative-enrichment whole-genome CRISPR–Cas9 pooled lentiviral screen[73] (*p* values were calculated using a Wilcoxon matched-pairs signed rank test). **k** The average ATAC peaks enrichment profile in PBS or pIpC-treated *MLL-AF9* GFP⁺ BM cells. Heatmap illustrating the ChIP-seq signal 3 kb up and downstream. **l** The average ChIP enrichment profile of BRD9 and BRG1 in the cluster of decreased by KO (top) and increased by KO (bottom) shown in Supplementary Fig. 4f. **m** Transcriptional factor motif analysis of increased ATAC peaks in pIpC-treated KO group. Source data are provided as a Source Data file.

The above data indicated that *Brd9* is indispensable for *MLL-AF9* leukemia initiation. We therefore next investigated the effects of *Brd9* depletion on leukemia maintenance. Genomic deletion of exons 4-6 of *Brd9* significantly reduced the GFP⁺ rate and prolonged survival compared to the PBS-treated group (Fig. 7g, h). Although these results were not consistent amongst all pIpC-treated mice, we found that *Brd9* was not efficiently depleted in the GFP⁺ BM cells of mice that died within 50 days after transplant (Supplementary Fig. 12c), indicating escape of

*Brd9* deletion in these cases. Consistent with these data, Gene Set Enrichment Analysis (GSEA) of c-Kit⁺ BM cells demonstrated enhanced myeloid-lineage development program and reduced expression of leukemia-associated pathways involved in MYC activation[24] and ribosomal biogenesis[72] with *Brd9* deletion (Fig. 7i and Supplementary Fig. 12d) as in Fig. 3d and Supplementary Fig. 4f. Moreover, we newly performed RNA-seq analysis using dBRD9-treated MOLM-13 cells (50 nM for four days) and found that ribosome biogenesis and

neutrophil pathways were most significantly down- and up-regulated, respectively (Supplementary Fig. 12e). Indeed, the dependency of AML cells' survival on BRD9 has been reproduced by our reanalyzing of the results of previously performed whole-genome CRISPR dropout screening (Fig. 7j)[73]. These findings suggest that, in addition to the profound effects on normal hematopoiesis, BRD9 has a pivotal role in AML development and maintenance in vivo.

To understand the impact of BRD9 loss on chromatin accessibility in primary AML cells, we performed ATAC-seq 10 days after pIpC injection in *MLL-AF9* transformed bone marrow cells from *Mx1-cre Brd9*fl/fl mice. This revealed that BRD9 loss led to an overall increase in euchromatin genome-wide (50,237 peaks examined, Fig. 7k). Moreover, by clustering ATAC-seq peaks (Supplementary Fig. 12f), the previously reported BRD9/BRG1-binding ncBAF sites in RN2 cells (mouse *MLL-AF9/Nras*G12D AML cells)[24] were associated with open chromatin after BRD9 depletion (Fig. 7l). This is supported by the fact that BRD9/BRG1-binding to chromatin in RN2 cells overlapped with open chromatin sites following BRD9 depletion (Supplementary Fig. 12g). Of note, these open chromatin regions significantly correlated with the motifs of CTCF, CTCFL (BORIS), and PU.1 binding (Fig. 7m), reproducing the findings in normal hematopoiesis.

## Discussion

Although roles for ncBAF have been investigated in the regulation of mouse[9] and human[74] pluripotent stem cells, the function of this complex has not been thoroughly previously investigated in hematopoiesis. Here we report that the BRD9 subunit of the ncBAF chromatin remodeling complex is required for the normal properties and differentiation of HSCs. We directly targeted the BRD9 bromodomain in vivo and examined the roles of BRD9 on normal HSCs and identified that BRD9 loss promoted myeloid-lineage commitment and mimicked aged HSCs and an MDS-like phenotype. Moreover, loss of BRD9 impaired HSC stemness, remarkably inhibited B cell development, and promoted myeloid differentiation. At the same time, targeting BRD9 in AML inhibited its development and maintenance by immediately enhancing myeloid differentiation and impairing Myc pathway and ribosome biogenesis (Fig. 7 and Supplementary Fig. 12). Related to the effects on lymphoid development, the unaffected CLP and MPP4 commitment does not directly cause the increase of myeloid cells in vivo, and the observed myeloid skewing is not generated at the expense of lymphoid lineage commitment. BRD9 loss negatively impacts on cell-cycling expansion stages of lymphoid lineage, such as preB to proB transition (Fig. 2g and Supplementary Fig. 2k). Moreover, given that the ectopic expression of BRD9 did not increase mature B cells (Fig. 1g), it seems that the endogenous level of BRD9 is enough for B-cell development or maturation and that the B-cell phenotype is observed only when BRD9 is depleted or functionally impaired.

Despite the robust reduction of LT-HSC in the primary KO model, we observed comparable chimerism of LT-HSC and HSPCs in the competitive transplant setting (Fig. 4). We speculate that the lower number of "phenotypic" LT-HSCs is due to the decrease of cell-cycling LT-HSCs in the *Brd9* KO primary mice and that the more dormant LT-HSC can reconstitute the HSPC fraction in the transplant model under stress conditions. Another possibility is that *Brd9* KO HSCs modify the BM microenvironment to overcome their disadvantage against the normal HSCs. Indeed, we recently reported that MDS cells impair osteolineage differentiation of MSCs via MDS-derived extracellular vesicles (EVs), leading to less supportive BM niche function for normal hematopoiesis and to the relative dominance of MDS cells in the BM[75]. Our observation in the *Brd9* KO model will promote future studies to evaluate further why and how less proliferative MDS cells (e.g., spliceosomal disruption of BRD9[16]) become dominant over normal hematopoiesis.

In spite of apparently discrepant results where BRD9 loss induced MDS and inhibited AML, we observed shared effects of *Brd9* KO on

chromatin in normal and malignant hematopoietic cells, including enhancing chromatin accessibility at ncBAF/BRD9 and CTCF loci. Our data suggest that these events occur as a primary or direct events shortly after BRD9 depletion (Fig. 1d, e and Supplementary Figs. 8c and 12e). Our mechanistic studies uncovered the roles of BRD9/CTCF in vivo and established that BRD9 loss disturbed the TF-mediated differentiation programs. Taken together, these results underscore that CTCF and chromatin organization mediates the cell fate of HSPCs.

In line with previous findings using embryonic stem cells and transformed cell lines[8,9], we demonstrated that ncBAF complexes uniquely localize to CTCF sites and promoters and that BRD9 peaks were correlated with CTCF, BRG1, and BRD4 in primary HSPCs. Such colocalization with CTCF is restricted to BRD9-containing ncBAF, not to cBAF and PBAF[8]. Given that PBAF or cBAF are involved in promoting terminal myeloid differentiation[23,76], it is tempting to speculate that ncBAF competes with other BAFs at pan-BAF ATPase BRG1 sites or independently constructs CTCF-associated loops. Although a study showed that CTCF binding is not affected by chemical BRD9 inhibition and ncBAF is not required for establishing CTCF sites in prostate cancer cell lines[59], the precise effects of BRD9/ncBAF on CTCF localization have been poorly understood and might be context-dependent. Unlike the modest changes of BRG1 and BRD4, CTCF peaks were robustly enhanced in KO condition (Fig. 5e), suggesting that BRD9 negatively affects CTCF chromatin localization at certain loci in HSPCs. It is also important to note that such increment of CTCF peaks in KO cells was mainly observed at sites with modest CTCF localization in control cells (Fig. 5f). Indeed, the DNA binding affinity of one TF can be affected by another factor binding nearby[77]. Thus, one potential explanation for these findings is that CTCF–DNA binding might be different depending on the distance between chromatin binding sites of CTCF and BRD9/ncBAF. Another possibility is that BRD9 physically regulates CTCF's target search before CTCF tightly binds to sites essential for chromatin loop formation[78]. In both cases, how BRD9 recognizes CTCF and DNA, for example, via the interaction between BRD9 bromodomain and acetylated lysine of histones or CTCF, will need to be addressed in future studies. Although our phenotypic analysis suggested the effect of BRD9 on proliferation, differentiation, and CTCF localization occur in a bromodomain- (Figs. 1b, g and 2f) and ncBAF- (Fig. 5b and Supplementary Fig. 7c) dependent manner, further evaluation will be necessary.

Integrating ChIP-, ATAC-, and RNA-seq analyses with HiC, we found that BRD9 deletion impacted genome-wide chromatin loop profiles without profound effects on larger scale compartment A/B or TAD formation. Loop formations are far more dynamic and regulated during cell differentiation not only by CTCF and cohesin, but by other factors, such as YY1[63]. CTCF-upregulated sites by BRD9 KO do not harbor YY1 binding (Supplementary Fig. 8e, f), suggesting essential roles of CTCF in the differentiation program of HSCs. It has been shown that the deletion of cohesin binding component STAG2, which is frequently mutated in MDS, increases HSC self-renewal and reduces differentiation capacity[47,79]. A recent study reported that loss of ZNF143, a TF regulating the binding of CTCF, significantly decreased the intensities of the chromatin interactions, but did not affect the A/B compartmentalization or formation of TAD[71]. Intriguingly, in *Brd9* KO conditions, the number of chromatin loops was drastically increased, and contact frequencies of the loops present in control cells were mostly raised. BRD9 deletion also caused genome-wide CTCF enrichment, particularly in the anchors of loops, which may have led in part to the increased number of loops and their contact frequencies since elevated contact frequencies were accompanied by greater CTCF signals in the loop anchors.

Although the increment of CTCF peaks at promoter regions was associated with increased mRNA expression in KO cells, how chromatin loops ultimately impact transcriptional regulation is a matter of ongoing investigation[80]. While artificial looping of an enhancer to

promoter of the β-globin locus could trigger transcription of β-globin gene[81], stable and ubiquitous loops are present at the *HoxD* locus in tissues where *HoxD* genes are not expressed[82,83]. Moreover, elimination of loop domains by cohesin degradation results only in modest transcriptional consequences[80]. Indeed, STAG1 can maintain TAD boundary integrity in STAG2-depleted conditions, while STAG2 is required for intra-TAD interactions for the transcription of specific lineage genes[47]. In our model, increased chromatin loops did not always trigger transcription. Further studies are needed to clarify how BRD9/ ncBAF selectively regulates loop-anchored or non-loop-anchored genes in a lineage-dependent manner.

In summary, our data illustrate a key role for BRD9 in the lineage commitment of HSCs and in the fine-tuning of CTCF-mediated chromatin state. Our studies provide compelling evidence that BRD9 determines the cell fate of HSCs in a cell-autonomous fashion. This further expands the function of ncBAF complexes in chromatin 3D regulation as well as in normal and malignant stem cell biology, offers the therapeutic opportunity by regulating differentiation, and contributes to our understanding of the pathogenesis of cancers where *BRD9* expression is post-transcriptionally disrupted.

## Methods

### Ethical approval
All animal procedures were completed in accordance with the Guidelines for the Care and Use of Laboratory Animals and were approved by the Institutional Animal Care and Use Committees at MSKCC and FBRI. All mouse experiments were performed in accordance with a protocol approved by the MSKCC (11-12-029) and FBRI (18-06) Institutional Animal Care and Use Committee. All animals were housed at Memorial Sloan Kettering Cancer Center (MSKCC) and at Foundation for Biomedical Research and Innovation (FBRI, Japan) using a 12 light/12 dark cycle and with ambient temperature maintained at $72\,°F \pm 2\,°F$ (~21.5 °C ± 1 °C) with 30–70% humidity. *Mx1*-cre mice were obtained from The Jackson Laboratory[33].

### Generation of *Brd9* conditional knockout (cKO) mice
A 9.95 kb region used to construct the targeting vector was first subcloned from a positively identified C57BL/6 BAC clone (RP23-209L12) using a homologous recombination-based technique. The region was designed such that the long homology arm (LA) extends ~5.7 kb 5′ to the distal LoxP site. The LoxP cassette is 322 bp upstream of exon 4. A Neo cassette is positioned 219 bp downstream exon 6. The targeted region is ~2.4 kb including exons 4-6 corresponding its bromodomain (Fig. 2a). The short homology arm (SA) extends ~1.9 kb 3′ to the Neo cassette. The final targeting vector was constructed using conventional cloning and recombineering techniques. The targeting vector is confirmed by restriction analysis after each modification step, and by sequencing using primers designed to read from the Neo selection cassette into the 3′ end of the MA (AGTATGGCTTTCCTTCCCGATGG) and from Neo into the 5′ end of the SA (TCTAAGGCCGAGTCTTATG AGCAG). The distal LoxP site was confirmed by sequencing with primer (AAGTTGTGACCTGCACCCAGG). Primers (GAGTGCACCATATGGACA TATTGTC and TAATGCAGGTTAACCTGGCTTATCG) anneal to the vector sequence and read into the 5′ and 3′ ends of the homology arms, respectively. Ten micrograms of the targeting vector was linearized and then transfected by electroporation of FLP C57Bl/6 (BF1) embryonic stem (ES) cells. After selection with G418 antibiotic, surviving clones were expanded for PCR analysis to identify recombinant ES clones. The Neo cassette in targeting vector has been removed during ES clone expansion. Screening primer A1 (AGGATAA AGGTGTCCCTCACCAG) was designed downstream of the short homology arm (SA) outside the 3′ region used to generate the targeting construct. PCR reactions using A1 with the FRTN primer (TCGTTCGAACATAACTTCGTATAGC) amplify 2.03 kb fragment. Five clones were identified as positive and selected for further expansion

(Supplementary Fig. 2a). A PCR was performed on five clones to detect presence of the distal LoxP site using the LOX1 (AGTGACCTCAAG ATTGCATGTTGG) and SDL2 (AGCTCCATATTCATCAGGGTGTGC) primers. Confirmation of distal LoxP retention was performed by PCR using the LOX1 and FN2A (AACTTCGCGACACGGACACAATCC) primers. This reaction produces a product 2.65 kb in size. Sequencing was performed on purified PCR DNA to confirm presence of the distal LoxP cassette using the SDL2 primer.

Targeted iTL BF1 (C57BL/6 FLP) embryonic stem cells were microinjected into Balb/c blastocysts. Resulting chimeras with a high percentage black coat color were mated to C57BL/6 WT mice to generate Germline Neo Deleted mice. Tail DNA was analyzed as described below from pups with black coat color. Primer set PNDEL1 (TGTGCCTAACAGGCTCACAA) and PNDEL2 (AGCAGGACTTTACC TCTCCCT) was used to screen mice for the deletion of the Neo cassette. The PCR product for the wildtype is 505 bp. After Neo deletion, one set of LoxP-FRT sites remain (121 bp). A second band with a size of 626 bp indicates Neo deletion. Primer set newFLP1 (ACAGAGACAAAG ACAAGCGTTAGTAGG) and newFLP2 (ATTTCCCACAACATTAGTCAA CTCCGTTAGG) was used to screen mice for the FLP transgene. The amplified product for primer set newFLP1 and newFLP2 is 330 bp. As described above, A PCR was performed to detect presence of the distal LoxP site using the LOX1 and SDL2 primers. This reaction amplifies a wild type product 468 bp in size. The presence of a second PCR product 49 bp greater than the wild type product indicates a positive LoxP PCR. To confirm Short Homology Arm Integration, tail DNA samples from positive mice were amplified with primers NEOGT (GTCCGTGTCGCGAAGTTCCTATACTTTC) and A1. NEOGT is located inside the remaining Neo sequence and A1 is located downstream of the short homology arm, outside the region used to create the targeting construct. NEOGT / A1 amplifies a fragment of 2.07 kb in length. Next, confirmed F1 germline neo deleted mice were set up for mating with a C57BL/6 wildtype mouse to generate F2 heterozygotes and performed screening for Neo deletion and FLP transgene as described above. After extensively backcrossing the mice to C57BL/6, we generated the conditional KO model by crossing the mice with *Mx1*-cre mice obtained from The Jackson Laboratory.

### Genotyping of *Brd9* cKO mice
*Brd9* WT and *Brd9* floxed littermate mice were genotyped by PCR with primers PNDEL1 and PNDEL2 using the following parameters: 94 °C for 2 min, flowed by 30 cycles of 94 °C for 45 s, 60 °C for 30 s, and 72 °C for 1 min, and then 72 °C for 7 min. The *Brd9* floxed allele and WT allele were detected as a band of 505 bp and 626 bp, respectively. The excision of exon 4-6 were confirmed by PCR with primers LOX1 and PNDEL2 using the same PCR condition. The excised allele was detected as a band of 605 bp, respectively.

### Bone marrow (BM) transplantation
Freshly dissected femora and tibiae were isolated from *Mx1*-Cre control, *Mx1*-Cre;*Brd9*fl/wt, and *Mx1*-Cre;*Brd9*fl/fl CD45.2+ mice. BM was flushed with a 3 ml insulin syringe into PBS supplemented with 3% fetal bovine serum. The BM was spun at 0.5 g by centrifugation and RBCs were lysed in ammonium chloride-potassium bicarbonate lysis buffer for 5 min. After centrifugation, cells were resuspended in PBS plus 2% FBS, passed through a cell strainer, and counted. Finally, $1.0 \times 10^6$ total BM cells of *Mx1*-Cre control, *Mx1*-Cre;*Brd9*fl/wt, and *Mx1*-Cre;*Brd9*fl/fl CD45.2+ mice were mixed with $1.0 \times 10^6$ WT CD45.1+ support BM and transplanted via tail vein injection into lethally irradiated (two times 450 cGy) CD45.1+ recipient mice. Chimerism was measured by FACS from the peripheral blood at 4 weeks after transplant (week 0, pre-pIpC (polyI:polyC)). Chimerism was followed via FACS in the peripheral blood every 4 weeks. Additionally, for each bleeding, whole blood cell counts were measured on a blood analyzer, and peripheral blood smears were scored. For noncompetitive transplantation experiments,

$2.0 \times 10^6$ total BM cells of *Mx1*-Cre control, *Mx1*-Cre;*Brd9*^fl/wt^, and *Mx1*-Cre;*Brd9*^fl/fl^ CD45.2^+^ mice were injected into lethally irradiated CD45.1^+^ recipient mice.

## shRNA experiments

Cells were transduced with a doxycycline-inducible LT3GEPIR lentiviral vector, T3G-GFP-mirE-PGK-Puro-IRES-rtTA3, expressing shRNAs against BRD9 or a non-targeting shRNA against Renilla. shRNAs were induced with the addition of doxycycline (2.0 µg/ml, Sigma-Aldrich). All shRNAs were designed using the SplashRNA algorithm. The shRNA sequences are: BRD9 shRNA#1 (human, shBRD9_352):TTTATTATC ATTGAATATCCAG; BRD9 shRNA#2 (human, shBRD9_353):TTTTATT ATCATTGAATATCCA; BRD9 shRNA#3 (human, shBRD9_348): TTATC ATTGAATATCCAGGAGC; Brd9 shRNA#1 (mouse, shBrd9_511):TTTATT ATCATTGAATACCCAG; Brd9 shRNA#2 (mouse, shBrd9_512):TTTTATT ATCATTGAATACCCA; Brd9 shRNA#3 (mouse, shBrd9_1115): TTTATTTCTTCTTTCATCTTTG.

## Retroviral transduction and transplantation of primary hematopoietic cells

The bone marrow of 16-week-old *Mx1*-Cre control and *Mx1*-Cre;*Brd9*^fl/fl^ were treated with pIpC (20 mg/kg, three times) followed by BM harvest 4 weeks later. MACS-sorted c-Kit^+^ cells using CD117 MicroBeads (MACS, Miltenyi Biotec) were transduced with viral supernatants containing murine stem cell virus (MSCV)-driven MLL-AF9 fusion oncogene in a construct tagged with GFP reported from an internal ribosomal entry site (MSCV-MLL−AF9-IRES-GFP) for 3 days in Iscove's modified Dulbecco's medium (IMDM) with 20% fetal calf serum (FCS) supplemented with mouse stem cell factor (mSCF) (25 ng/ml), mouse interleukin (IL)-3 (10 ng/ml) and mouse IL-6 (10 ng/ml), followed by injection of ~1,000,000 cells per recipient mouse via tail vein injection into lethally irradiated (900 cGy) CD45.1 mice. Similarly, we generated pIpC-untreated *Mx1*-Cre;*Brd9*^fl/fl^ transplant model expressing MLL-AF9 cDNA and harvested CD45.2^+^ AML cells. Then, 6-week-old, sublethally irradiated (450 cGy) C57/BL6 recipient mice were injected with 1000 primary CD45.2^+^ MLL−AF9 leukemic cells. The recipient mice were treated with PBS or pIpC the next day after transplant (three times, every other day). For the BRD9-cDNA or shBrd9 transplant model, we utilized MSCV-BRD9-PGK-Puro-IRES-GFP (Addgene, 75114), MSCV-BRD9_dBD-PGK-Puro-IRES-GFP (Addgene, 75115), MSCV-BRD9_N216A-PGK-Puro-IRES-GFP (Addgene, 75116), MLS-E-GFP-shREN (Addgene, 105583), and two different vectors of MLS-E-GFP-shBrd9 whose antisense guide sequence are TTTATTATCATTGAATACCCAG and TTTTATTATCATTGAATACCCA.

## Cord blood HSC differentiation assays

Primary human CD34-enriched cord blood using CD34 MicroBead Kit (Miltenyi Biotec, 130-046-702) was transduced with lentivirus (SGEP shControl and shBRD9, Addgene, 111170) overnight in StemSpan™ SFEM II (StemCell Technologies, 9605) with StemSpan™ CD34^+^ Expansion Supplement (StemCell Technologies, 2691) and sorted for GFP expression using a FACSAria III (Becton Dickinson). For the differentiation assay, cells were washed and incubated in StemSpan™ SFEM II with CD34^+^ Expansion Supplement, Myeloid Expansion Supplement (StemCell Technologies, 2693), or Erythroid Expansion Supplement (StemCell Technologies, 2692). Myeloid differentiation was assessed by flow cytometry using anti human CD34, CD33 (WM53) and CD14 (MfP9) (BD Biosciences). Erythroid differentiation was assessed by flow cytometry using anti-human GPA (CD235a) (HIR2) and CD71 (OKT9) (BD Biosciences).

## In vitro colony-forming assays

Whole BM cells from *Mx1*-Cre control, *Mx1*-Cre;*Brd9*^fl/wt^, and *Mx1*-Cre;*Brd9*^fl/fl^ mice and seeded at a density of 20,000, 100,000, 200,000 cells/replicate into cytokine-supplemented methylcellulose medium

(Methocult M3434, M3436, and M3630, respectively; STEMCELL Technologies). Colonies propagated in culture were scored at day 7−10 and M3434 colonies were replated into new M3434 semisolid media. For cDNA expression experiment, c-Kit^+^ cells were selected by anti-mouse CD117 MicroBeads (Miltenyi Biotech) from 14-week-old primary CD45.1^+^ mice), and were cultured overnight in IMDM/10% FBS medium supplemented with 50 ng/ml recombinant murine SCF (250-03; Peprotech), 20 ng/ml recombinant murine IL-3 (213-13; Peprotech), 20 ng/ml recombinant murine IL-6 (216-16; Peprotech), and 20 ng/ml recombinant murine TPO (315-14; Peprotech). On the next day, those cells were infected with SGEP shRNA retrovirus. Three days after infection, $2.0 \times 10^4$ of GFP^+^ cells were FACS-sorted and plated in cytokine-supplemented methylcellulose medium (Methocult M3434; STEMCELL Technologies) in triplicate. For shBRD9 transduced human CD34^+^ cells, 3000 GFP^+^ cells were plated into cytokine-supplemented methylcellulose medium (Methocult H4434, STEMCELL Technologies). For wildtype whole BM cells treated with DMSO and BI-7273 (BRD9 bromodomain inhibitor, Selleck, S8179) for 24 h, 1,000,000 cells were plated into cytokine-supplemented methylcellulose medium (Methocult M3630; STEMCELL Technologies).

## RT−PCR and quantitative RT−PCR

Total RNA was isolated using RNeasy Mini or Micro kit (Qiagen). For cDNA synthesis, total RNA was reverse-transcribed to cDNA with SuperScript VILO cDNA synthesis kit (Life Technologies). The resulting cDNA was diluted 10−20 fold before use. Quantitative RT-PCR (qRT−PCR) was performed in 10-µl reactions with SYBR Green PCR Master Mix. All qRT−PCR analysis was performed on an Applied Biosystems QuantStudio 6 Flex Cycler (Thermo Fisher Scientific). Relative gene expression levels were calculated using the comparative CT method.

For Fig. 1h, Lin^+^ cells were depleted by lineage antibody cocktail (B220, CD19, CD3, CD4, CD8, CD11b, Gr1, Ter119, NK1.1, CD11c). LSK, Lin^−^Sca1^+^c-Kit^+^; HPC, Lin^−^Sca1^+^c-Kit^+^CD150^+^CD48^+^CD127^−^; LT-HSC, Lin^−^Sca1^+^c-Kit^+^CD150^+^CD48^−^CD127^−^; ST-HSC, Lin^−^Sca1^+^c-Kit^+^CD150^−^CD48^−^CD127^−^; MPP, Lin^−^Sca1^+^c-Kit^+^CD150^−^CD48^+^CD127^−^; CMP, Lin^−^Sca1^−^c-Kit^+^CD34^+^CD16/32^−^CD127^−^; CMP, Lin^−^Sca1^−^c-Kit^+^CD34^+^CD16/32^−^CD127^−^; GMP, Lin^−^Sca1^−^c-Kit^+^CD34^+^CD16/32^+^CD127^−^; MEP, Lin^−^Sca1^−^c-Kit^+^CD34^−^CD16/32^−^CD127^−^; CLP, Lin^−^Sca1^int^c-Kit^int^CD127^+^CD135^+^; B-cell Fr.A, Ter119^−^B220^+^CD19^−^CD43^−^IgM^−^IgD^−^; B-cell Fr.B, Ter119^−^B220^+^CD19^−^CD43^+^IgM^−^IgD^−^; B-cell Fr.C, Ter119^−^B220^+^CD19^+^CD43^+^IgM^−^IgD^−^; B-cell Fr.D, Ter119^−^B220^+^CD19^+^CD43^−^IgM^+^IgD^−^; B-cell Fr.E, Ter119^−^B220^high^CD19^+^CD43^−^IgM^+^IgD^−^; CD11b^+^Gr1^+^, Ter119^−^CD11b^+^Gr1^+^B220^−^CD19^−^; CD11b^+^Gr1^low^, Ter119^−^CD11b^+^Gr1^−^B220^−^CD19^−^; Blood B-cells, Ter119^−^CD11b^−^Gr1^−^B220^+^CD19^+^CD3^−^; Blood T-cells, Ter119^−^CD11b^−^Gr1^−^B220^−^CD19^−^CD3^+^; Blood CD11b^+^Gr1^+^, Ter119^−^CD11b^+^Gr1^+^B220^−^CD19^−^CD3^−^; Blood CD11b^+^Gr1^low^, Ter119^−^CD11b^+^Gr1^−^B220^−^CD19^−^CD3^−^; Spleen MZ (Marginal Zone) B-cells, Ter119^−^CD11b^−^Gr1^−^CD3^−^B220^+^CD19^+^CD21^+^CD23^−^; Spleen Follicular B-cells, Ter119^−^CD11b^−^Gr1^−^CD3^−^B220^+^CD19^+^CD21^low^CD23^+^; Spleen Transitional B-cells, Ter119^−^CD11b^−^Gr1^−^CD3^−^B220^+^CD19^+^CD21^−^CD23^−^; Erythroid Fr.I, CD45^−^CD11b^−^Gr1^−^Ter119^int^. Erythroid Fr.II to Erythroid Fr.IV population (CD45^−^CD11b^−^Gr1^−^Ter119^+^) were fractionated by CD44/FSC plot as previously reported[84]. Primers used in RT-PCR reactions were: *BRD9* (human) fwd, GCAATGACATACAATAGGCCAGA and rev, GAGCTGCC TGTTTGCTCATCA; *Brd9* (mouse) fwd, ATCCTATGGACTTTGGCACG and rev, CTGGTCTATTGTACGTCATCGC; *Stag1* (mouse) fwd, TCAGA GTTACCAGTGTTACAGGA and rev, CCAGGACGACCCCTTTTTC; *Stag2* (mouse) fwd, CTACAAGCATGACCGGGACAT and rev, GCCGTACTAAC ACACCAATGAAC; *Smc3* (mouse) fwd, CGAAGTTACCGAGACCAAACA and rev, TCACTGAGAACAAACTGGATTGC; *Smc1a* (mouse) fwd, TCGGACCATTTCAGAGGTTACC and rev, CAGGTGCTCCATGTATCA GGT; *Rad21* (mouse) fwd, ATGTTCTACGCACATTTTGTCCT and rev, TGCACTCAAATACATGGGCTTT; *Ctcf* (mouse) fwd, GATCCTA CCCTTCTCCAGATGAA and rev, GTACCGTCACAGGAACAGGT; *Gapdh*

(mouse) fwd, AGGTCGGTGTGAACGGATTTG and rev, TGTAG ACCATGTAGTTGAGGTCA.

## Antibodies, FACS, and Western blot analysis

All FACS antibodies were purchased from BD Pharmingen, eBioscience, or BioLegend. BM mononuclear cells were stained with a lineage cocktail comprised of antibodies targeting CD3, B220, Gr-1, CD11b, Ter119. Cells were also stained with antibodies against c-Kit, Sca1, CD150, and CD48. Cell populations were analyzed using a FAC-SLyric (BD Biosciences) and sorted with a FACSMelody (BD Biosciences). We used the following antibodies: B220-PECy7 (clone: RA3-6B2; BD Pharmingen; catalog #: 10322; dilution: 1:300); B220-PE (RA3-6B2; BD Pharmingen; 553090; 1:200); CD3-APCCy7 (145-2c11; BioLegend; 100330; 1:300); Gr1-APC (RB6-8C5; BD Pharmingen; 553129; 1:300); CD11b-PerCPCy5.5 (M1/70; Biolegend; 101228; 1:300); c-Kit-APC (2B8; BD Pharmingen; 553356; 1:200); c-Kit-PE (2B8; BD Pharmingen; 553356; 1:200); Sca1-PECy7 (D7; BioLegend; 108114; 1:400); Sca1-FITC (D7; BioLegend; 108106; 1:200); Streptavidin-PerCPCy5.5 (BioLegend; 405214; 1:400); Streptavidin-BV605 (BioLegend; 405229; 1:400);CD45.1-FITC (A20; BD Pharmingen; 553775; 1:300); CD45.2-PE (104; BD Pharmingen; 560695; 1:300); CD48-APCCy7 (HM48-1; BioLegend; 103432; 1:400); CD150-BV605 (TC15-12F12.2; BioLegend; 115927; 1:100); CD16/CD32 (FcγRII/III)-Alexa700 (93; eBioscience; 56-0161-82; 1:200); CD34-eFluor450 (RAM34; eBioscience; 48-0341-82; 1:50); CD135-PE (A2F10.1; BD Pharmingen; 553842; 1:100); CD127 (IL-7Rα)-APCCy7 (A7R34; BioLegend; 135039; 1:100); CD19- PECy7 (6D5; BioLegend; 115520; 1:200); IgM-FITC (RMM-1; BioLegend; 406505; 1:200); CD43-PerCPCy5.5 (S7; BD Pharmingen; 562865; 1:200); CD24-PECy7 (M1/69; BD Pharmingen; 560536; 1:50); CD93(AA4.1)-PerCPCy5.5 (AA4.1; Invitrogen; 45-5892-82; 1:50); MitoTracker Deep Red (M22426; Invitrogen; 200 nM). The following antibodies were used for Western Blot analysis: BRD9 (ab259839; Abcam; 1:1000), CTCF (07-729; Millipore; 1:1000), FLAG (F1804; Sigma; 1:1000), Actin (A-5441; Sigma-Aldrich; 1:5000), and IgG (ab37355; abcam). The following antibodies were used for ChIP-seq analysis: BRD9 (ab259839; Abcam), BRD4 (A301-985A100; Bethyl; 1:1000), BRG1 (Ab110641; Abcam), CTCF (07-729; Millipore), H3K4me3 (ab8580; abcam), H3K4me1(ab8895; abcam), and H3K27Ac (ab4729; abcam).

## Histological analysis

Mice were sacrificed and autopsied, and the dissected tissue samples were fixed for 24 h in 4% paraformaldehyde, dehydrated, and embedded in paraffin. Paraffin blocks were sectioned at 4 mm and stained with H&E. Images were acquired using an Axio Observer A1 microscope (Carl Zeiss).

## Peripheral blood analysis

Blood was collected by retro-orbital bleeding using heparinized microhematocrit capillary tubes (Thermo Fisher Scientific). Automated peripheral blood counts were obtained using a HemaVet 950 (Drew Scientific) according to standard manufacturer's instruction. Differential blood counts were realized on blood smears stained using Wright-Giemsa staining and visualized using an Axio Observer A1 microscope.

## In vitro cell viability assays

Cells were seeded in white flat-well 96-well plates (Costar) at a density of 10,000 cells per well. ATP luminescence readings were taken every 24 h after seeding, using Cell Titer Glo (Promega) according to the manufacturer's instructions.

## Phenylhydrazine treatment

Mice were injected intraperitoneally with phenylhydrazine (PHZ) at a dose of 50 mg/kg body weight. Peripheral blood was collected for RBC/Hb/Hct measurement.

## Cell lines and tissue culture

All cell lines were obtained from ATCC (Manassas, VA, USA) and Toshio Kitamura (University of Tokyo). HEK293T and PlatE cells were cultured in DMEM/10% FBS. K562, HL-60 and MOLM13 cells were cultured in RPMI/10% FBS. For the differentiation assay, K562 and HL-60 cells were treated with 30 μM hemin (Sigma-Aldrich, 51280) for 3 days, and with 1 μM ATRA (Sigma, R2625) for 3–6 days. All cell culture media include 100 U/ml penicillin and 100 μg/ml streptomycin (Gibco). Each differentiation was assessed by flow cytometry using anti CD235a, CD71, and CD11b.

## BrdU injection and cell cycle analysis

Mice received 100 μl of 10 mg/ml BrdU (ab142567; Abcam) dissolved in PBS by intraperitoneally injection to label dividing cells and were sacrificed 1-day post-injection. Harvest BM cells and MACS enrichment was performed by Lineage Cell Depletion Kit (Miltenyi Biotech, 130-090-858). Following enrichment, $1.0 \times 10^6$ cells were suspended in PBS containing antibodies against c-Kit, Sca1, CD150, and CD48 and incubated on ice for 1 h. After washing with 1 ml PBS, cells were fixed in BD Cytofix/Cytoperm Buffer (BD Biosciences, Cat. 51-2090KZ) and permeabilized by BD Cytoperm Permeabilization Buffer Plus (BD Biosciences, Cat. 51-2356KC) according to manufacturer's handbook. For DNA digestion, cell pellets were treated with DNase I and incubated at 37 °C for 1 h. After washing with 1 ml BD Perm/Wash Buffer (BD Biosciences, Cat. 51-2091KZ), resuspend the cells in 50 μl of BD Perm/Wash Buffer containing diluted anti-BrdU-FITC (1:10) for 30 min at room temperature. After washing with 1 ml BD Perm/Wash Buffer, pellets were resuspended in 1 ml PBS containing DAPI (1:10,000) and acquired with BD FACSLyric.

## mRNA isolation and analysis

For MOLM13, K562 cells after pretreatment with 50 nM dBRD9 (Tocris, 6606) and DMSO for four days and sorted mouse cell populations (live, lineage-negative c-Kit+ cells), RNA was extracted using RNeasy columns (Qiagen) per manufacturer's instructions. RNA was then Poly(A)-selected, and stranded Illumina libraries were prepared using TruSeq Stranded mRNA Library Prep Kit. The libraries were sequenced with illumina NovaSeq 6000 at a depth of ~30 M 2 × 151 bp reads per sample. For differential expression analysis, sequenced reads were mapped to mm10 using nf-core/rnaseq v3.0 pipeline[85] with star_rsem aligner. RSEM gene counts were normalized and differentially expressed genes were identified with Adjusted $p$ value < 0.1 by DESeq2 v1.26.0,[86] where independent hypothesis weighting was applied.[87] Variance stabilizing transformed gene counts was used for hierarchical clustering and principal component analysis. Hierarchical clustering was done using Euclidean distance and Complete linkage method. With differentially expressed gene sets, gene ontology and gene set enrichment analysis (GSEA) pathway analysis was performed by Enrichr.[88–90] and GSEA software. Fragments per 1 kb of transcript per one million mapped reads (FPKM) were computed by a custom program based on the number of read pairs mapped to each transcript.

For K562 and MOLM13 cells, sequenced reads were mapped to GRCh38 with gene annotations provided by GENCODE 43 using nf-core/rnaseq v3.10.1 pipeline[85] with star_rsem aligner. The latter analyses were performed with the same way for mouse cell populations.

## scRNA-seq

Single cell RNA-seq libraries of sorted LK cells from control Brd9fl/fl and Mx1-Cre;Brd9fl/fl mice, at 8 weeks post-deletion were prepared according to the manufacturer's protocol using the 10x Genomics Chromium Next GEM Single Cell 3' Kit v3.1 and Dual Index Kit. The libraries were pooled together and sequenced with NovaSeq 6000 (illumina) at a depth of ~ 448 M reads per sample. Raw sequencing data was demultiplexed and converted to the standard FASTQ files by executing cellranger mkfastq (6.0.1, 10x Genomics). Gene counts of

each sample were generated by running cellranger count (6.0.2, 10x Genomics) with mm10-2020A as the transcriptome dataset. For each sample, about 8000 cells were retrieved, where 50,000 mean reads and 4000 median genes were obtained per cell. The Gene count datasets were aggregated into a single gene count dataset by running cellranger aggr. The filtered feature barcode matrices of the samples were analyzed using Seurat (4.0.3) R package[91]. After calculating QC metrics including mitochondrial percentage, low quality cells without the following criteria were removed: nCount_RNA < 50,000, nCount_RNA > 1000, nFeature_RNA > 1000, nFeature_RNA < 6500 and percent.mt < 7. As a result 15,000 cells were retained. SCTransform function with the parameters variable.feature. $n = 3000$ and vars.to.regress = "percent.mt" was applied for the first normalization and cell cycle scores were calculated by CellCycleScoring function with cc.genes.updated.2019 cell cycle genes, where difference of cell cycle scores (CC.Difference) between S phase and G2M phase were obtained. Then, SCTransform function with the parameters variable.feature. $n = 3000$ and vars.to.regress = c("percent.mt", "CC.Difference") was applied for the second normalization. Finally, each data was integrated through executing SelectIntegrationFeatures with nfeatures = 3000, PrepSCTIntegration, FindIntegrationAnchors with normalization.method = "SCT" and reduction = "cca" and dims = 1:100 and IntegrateData. Principal component analysis (PCA) was performed with the integrated data followed by uniform manifold approximation and projection (UMAP) dimensional reduction, FindNeighbors and FincClusters. Cell clusters and UMAP coordinates were exported from Seurat object and imported into the Loupe Browser file for visualization. Maturation principal component (maturation PC) analyses were performed as described[92], where cells in LT-HSC, ST-HSC, and MPP3 (LT-HSC, ST-HSC, MPP2 and MEP) clusters were used when calculating the principal components for myeloid (erythroid) lineage.

## ChIP-seq

As previously reported[92], for BRD9, BRG1, CTCF, and BRD4 ChIP-seq, $2-5 \times 10^6$ cells were fixed with 2 mM DSG (Thermo Fisher Scientific, 20593) for 30 min at room temperature. DSG was then removed and replaced with fixing buffer (50 mM HEPES-NaOH (pH 7.5), 100 mM NaCl, 1 mM EDTA) containing 1% paraformaldehyde (Electron Microscopy Sciences, 15714) and crosslinked for 10 min at 37 °C. For histone modification ChIP-seq, $2 \times 10^6$ cells were fixed with 1% paraformaldehyde for 10 min at room temperature. Crosslinking was quenched by adding glycine to a final concentration of 0.125 M. Cells were washed with ice-cold PBS and harvested in PBS. The nuclear fraction was extracted by first resuspending the pellet in 1 ml of lysis buffer (50 mM HEPES-NaOH (pH 8.0), 140 mM NaCl, 1 mM EDTA, 10% glycerol, 0.5% NP-40, and 0.25% Triton X-100) for 10 min at 4 °C. Cells were pelleted, and washed in 1 ml of wash buffer (10 mM Tris-HCL (pH 8.0), 200 mM NaCl, 1 mM EDTA) for 10 min at 4 °C. Cells were then pelleted and resuspended in 1 ml of shearing buffer (10 mM Tris-HCl (pH 8), 1 mM EDTA, 0.1% SDS) and sonicated in a Covaris sonicator. Lysate was centrifuged for 5 min at 17,000 × g to purify the debris. Then 100 µl of 10% Triton X-100 and 30 µl of 5 M NaCl were added. The sample was then incubated with 20 µl of Dynabeads Protein G (Life-Technologies,10003D) for 1 h at 4 °C. Primary antibodies were added to each tube and immunoprecipitation (IP) was conducted overnight in the cold room. Cross-linked complexes were precipitated with Dynabeads Protein G for 2 h at 4 °C. The beads were then washed in low salt wash buffer (20 mM Tris-HCl pH 8, 150 mM NaCl, 10 mM EDTA, and 1% SDS) for 5 min at 4 °C, high salt wash buffer (50 mM Tris-HCl pH 8, 10 mM EDTA, and 1% SDS) for 5 min at 4 °C and LiCl wash buffer (50 mM Tris-HCl pH 8, 10 mM EDTA, and 1% SDS) for 5 min at 4 °C. DNA was eluted in elution buffer (100 mM sodium bicarbonate and 1% SDS). Cross-links were reversed overnight at 65 °C. RNA and protein were digested with 0.2 mg/ml RNase A for 30 min at 37 °C followed by 0.2 mg/ml Proteinase K for 1 h at 55 °C. DNA was purified with phenol-

chloroform extraction and isopropanol precipitation. ChIPseq libraries were prepared using the Rubicon ThruPLEX DNA-seq Kit from 1 ng of purified ChIP DNA or input DNA according to the manufacturer's protocol. The libraries for proteins and histones were sequenced with HiSeq 2500 and NovaSeq 6000 at a depth of ~13 M 2 × 100 bp and 40 M 2 × 150 bp reads per sample, respectively. The adapter sequences were removed from the ChIP-seq FASTQs using Trimmomatic (0.39) with the options ILLUMINACLIP: TruSeq3-PE-2.fa:2:30:10 LEADING:30 TRAILING:30 SLIDINGWINDOW:4:15 MINLEN:50 on the PE mode. The cleaned FASTQs were aligned using Bowtie2 (2.4.4) to the mouse reference genome mm10 with -X 2000 and the duplicated reads were removed using picard (2.26.2) MarkDuplicates with REMOVE_DUPLICATES = true. BigWig files were generated from the bam files using bamCoverage included in DeepTools (3.5.1) with the parameters --ignoreDuplicates --normalizeUsing RPGC --effectiveGenomeSize 2150570000 --binSize 1 --ignoreForNormalization chrX --extendReads. For detecting super enhancers (SE) and typical enhancers, findPeaks module in HOMER software was used with the following parameters: -style super -L 0 -fdr 0.001. The peaks on enhancers of control and knock out samples were merged and used in the ATAC-seq analyses. Enriched motifs associated with peaks on TE between control and knock out samples were searched by findMotifsGenome.pl script in HOMER software with parameters, mm10 -size given. For calling peaks stemming from BRD9, BRG1, CTCF and BRD4 compared to input, MACS2 (2.2.7.1) was used with the parameters -g mm --nomodel --keep-dup all, where the significance level was $q < 0.05$. MACS2 was also used for calling histone modification peaks with the parameters -g mm --nomodel –keep-dup all --broad. The mergePeaks module in HOMER was used to merge peaks and produce pairwise comparison statistics from which the venn diagram between peaks. ChIP-seq scores of peaks (average RPGC values) were calculated using multiBigwigSummary BED-file from the bigWig files. The correlation of peaks was visualized with plotCorrelation --corMethod spearman. For visualizing ChIP-seq peaks and downstream analyzes, computeMatrix reference-point --referencePoint center --skipZeros was used to proruce matrix files. Genomic regions corresponding to the active enhancers, active promoters Ctcf co-localized and primed sites were determined with plotHeatmap --sortUsing mean --kmeans 5. BRG1, CTCF and BRD4 peaks at the BRD9 peaks of the control sample were visualized with plotHeatmap --sortUsing mean. Ctcf peaks were classified into three groups, UP, Neutral and Down, by the fold-change 1.5 between control and BRD9 knock-out samples. The genomic positions of them were visualized with annotated with annotatePeaks.pl script included in HOMER software. After annotating Ctcf peaks, the relationship between mRNA fold change and Ctcf localization change at promoter-TSS was visualized by volcano plot and violin plot. The differences between $\log_2$ FC and zero in UP, Neutral, Down groups were examined by two-sided t-test and shown in the volcano plot. Gene ontology analysis was performed using enrichr. The normalized numbers of CTCF ChIP read pairs mapped to the loop anchors were computed by a custom program based on the alignment. In Fig. 5b and Supplementary Fig. 7d, we utilized the reported method[8].

## ATAC-seq

ATAC-seq library preparation was performed on cultured, FACS sorted Lin⁻Kit⁺ HSPCs (150,000 cells per sample) and K562 cells (pretreated with 50 nM dBRD9 or DMSO for four days) using the ATAC-seq kit from Active motif (#13150). In short, nuclei were isolated by adding 100 µl ice cold ATAClysis buffer to the cell pellet and resuspending cells with a pipette. After centrifugation (500 × g, 10 min at 4 °C), cells were washed and incubated with the tagmentation master mix in a shaking heat block at 37 °C/800 rpm for 30 min. Obtained DNA was taken up in DNA purification buffer, purified using the contained DNA purification columns, amplified for 10 cycles using indexed primers, and size selected using SPRI bead solution. A quality control (QC) was

performed in order to verify the size distribution of the PCR enriched library fragments. To this end, aliquots of the DNA libraries were analyzed on an Agilent Technologies 2100-Bioanalyzer, using a High Sensitivity DNA chip. This Bioanalyzer based QC was for qualitative purposes only. A precisely quantification of the libraries was done using the Qubit dsDNA HS Assay Kit (Thermo Fisher Scientific). The bar-corded amplicons from ATAC-Library preparation were sequenced in a HiSeq 2500 platform (Illumina) at a depth of ~16 M 2 × 100 bp reads per sample. Reads were quality filtered according to the standard Illumina pipeline, de-multiplexed and fastq files were generated. The adapter sequences were removed from the ATAC-seq FASTQs using Trimmomatic (0.39) with the options ILLUMINACLIP: NexteraPE-PE.fa:2:30:10 LEADING:30 TRAILING:30 SLIDINGWINDOW:4:15 MIN-LEN:50 on the PE mode. The cleaned FASTQs were aligned using Bowtie2 (2.4.4) to the mouse reference genome mm10 with the parameters -X 2000, --no-discordant, --no-mixed --dovetail --very-sensitive and the duplicated reads were removed using picard (2.26.2) Mark-Duplicates with REMOVE_DUPLICATES=true. BigWig files were generated from the bam files using bamCoverage with the parameters --ignoreDuplicates --normalizeUsing RPGC --effectiveGenomeSize 2000000000 --binSize 1 --ignoreForNormalization chrX. Signal intensities in SE and TE sites were calculated using multi-BigwigSummary BED-file with -bs1. Normalized ATAC signals were calculated by signal intensities * length of enhancers/1000 and shown in box plot.

## HiC library preparation

While conventional HiC approaches use restriction enzymes (RE) to digest chromatin, the Omni-C HiC library preparation uses a sequence-independent endonuclease for chromatin digestion prior to proximity ligation and library generation. The sequence-independent chromatin fragmentation of Omni-C raises the coverage for detecting chromatin contacts by up to 20% of the genome with low restriction enzyme density and thus increases the enrichment in long-range *cis* reads. The Omni-C library was prepared using the Dovetail® Omni-C® Kit according to the manufacturer's protocol. Briefly, the chromatin was fixed with disuccinimidyl glutarate (DSG) and formaldehyde in the nucleus. The cross-linked chromatin was then digested in situ with DNase I. Following digestion, the cells were lysed with SDS to extract the chromatin fragments and the chromatin fragments were bound to Chromatin Capture Beads. Next, the chromatin ends were repaired and ligated to a biotinylated bridge adapter followed by proximity ligation of adapter-containing ends. After proximity ligation, the crosslinks were reversed, the associated proteins were degraded, and the DNA was purified then converted into a sequencing library using Illumina-compatible adapters. Biotin-containing fragments were isolated using streptavidin beads prior to PCR amplification. The library was sequenced on an Illumina NovaSeq 6000 platform to generate 3,063,850,678 (control) and 3,103,317,792 (KO) Hi-C paired-end reads (151 bp).

## Mapping of Hi-C reads pairs and Hi-C contact map creation

Mapping of Hi-C reads pairs and Hi-C contact map creation were carried out according to the Omni-C analysis protocol (https://omni-c.readthedocs.io/en/latest).Totals of 3,063,850,678 (control) and 3,103,317,792 (KO) Hi-C paired-end reads (151 bp) were aligned to the mouse genome (mm10) by using bwa[93], of which 2,083,192,106 (control) and 2,133,962,430 (KO) were uniquely mapped. By using pairtools (https://github.com/open2c/pairtools), the ligation junctions were parsed, sorted, and dereplicated. Totals of 591,474,477 (coverage: 89%, depth: 73.1) (control) and 399,651,068 (coverage: 89%, depth: 51.0) (KO) dereplicated read pairs mapped on the same chromosome (cis read pairs) were identified. Statistics of the classified alignments are given in Supplementary Table 1.

## Identification of topologically associating domains (TADs) and related features

The Hi-C contact maps were created with the dereplicated cis read pairs by using Juicer (version 1.22.01)[67]. The topologically associating domains (TADs) were identified by Juicer tools Arrowhead function (Knight-Ruiz normalization at 10 kb resolution)[94]. The PC1 values were computed by using homer runHiCpca.pl at 25 kb resolution[95]. The eigenvectors were calculated by Juicer eigenvector tools function at 1 Mb resolution[67]. The heatmaps comparing the TAD profiles and chromatin contact frequencies between control and KO cells were visualized by using HiTC R script[96] based on the Knight-Ruiz balanced observed/expected (O/E) frequency ratios at 25 kb resolution[94]. The O/E ratios were extracted from Juicer hic files by Juicer tools dump function, and converted to HiC-Pro format by a custom program. Insulation scores were computed by using homer at 25 kb resolution[95]. The overlap of TADs between the control and KO cells (instances where a single control TAD overlaps a single KO TAD both bounded by overlapping non-TAD/boundary regions) was identified by a custom program.

## Identification of chromatin loops

The chromatin loops were called by Juicer tools Hiccups function. Based on the filtered alignment of HiC paired-end reads, Juicer computed observed contact frequencies (O) at 5 kb, 10 kb, and 25 kb resolutions[67]. The observed contact frequencies were normalized by expected contact frequencies (E) based on the previous report[94], and thus normalized contact matrices containing O/E ratio (loop intensity) in each pixel were built at each resolution. By comparing the loop intensity in a given pixel to loop intensities in four neighborhood regions [(1) pixels to its lower-left, (2) pixels to its left and right, (3) pixels above and below, and (4) a donut surrounding the pixel of interest (Fig. 3A in Rao et al.[67])], Juicer identified pixels of enriched contacts requiring that the pixel being tested contain at least 50% more contacts than expected based on each neighborhood region and the enrichment be statistically significant after correcting for multiple hypothesis testing (False discovery rate <10%). The resulting enriched pixels tend to form contiguous interaction regions comprising 5 to 20 pixels each. Of the contiguous interaction region, the pixel with the highest loop intensity was defined as "peak pixel", which tethers together two loci on the same chromosome, or equivalently forms a chromatin loop. The two loci are referred to as "loop anchors"[80]. The loops both of whose anchors overlapped were merged into a single loop. Of the merged peaks, a set of non-redundant loops was generated for each control and KO cells.

## Reproducibility of Control and KO HiC maps based on aggregate analysis of contact frequencies over the chromatin loops

We prepared three biological replicates of Omni-C HiC libraries independently from sampling of cells. Each biological replicate was split into two technical replicates before sequencing, and they were sequenced in two distinct sequencing runs. After confirming high reproducibility (Pearson $R = 0.93$–0.95) of the technical replicates based on comparison of the profile of mapped sequence depth in 10 kb bin on chr2, we merged the two technical replicates to a single set. Since we obtained sufficiently large numbers of dereplicated read-pairs in both Control and KO cells in the first biological replicate, we could detect 2199 and 4640 chromatin loops in Control and KO cells, respectively. We then measured the data reproducibility between the three biological replicates by comparing the overall profiles of the loop intensities over the loops detected in Control and KO cells in the first biological replicate. As shown in Supplementary Fig. 9, the loop intensities of HiC maps of Control and KO cells aggregated over the Control and KO loops show significantly distinct profiles between the four combinations of loops and HiC maps in the first biological

replicate. The loop intensities of both Control and KO loops were raised from Control to KO cells, reflecting the overall increase of contact frequencies from Control to KO cells. The loop intensities of KO loops were the weakest in Control, and were most dynamically raised to KO cells, in agreement of the drastic increase in the number of loops from Control to KO cells. These two features of the overall profiles of contact frequencies observed in the first biological replicate were maintained in the second and third biological replicates. Then we merged the dereplicated read-pairs of all three biological replicates into a single dataset in each Control and KO cells, and carried out all the downstream analyses based on the merged datasets, including the creation of contact maps (hic files), TAD determination, and chromatin loop calling.

### Numbers and sizes of loops at distinct sequence depths and resolutions

To examine the influence of sequence depths on the numbers and sizes of loops, we loop called for decreased amounts (3/4 and 1/2) of whole Control and KO datasets of Biological replicate 1 (Supplementary Fig. 10 and Supplementary Tables 2 and 3). The number of loops decreased accordingly at nearly equal rates both in Control and KO datasets with decreasing sequence depths. The sizes of loops somewhat decreased at lower sequence depths. We also presented the numbers and sizes of loops called independently at the three resolutions, e.g., 5, 10 and, 25 kb (Supplementary Fig. 10 and Supplementary Table 4). More KO loops were detected at finer resolution than Control loops, which likely reflects the overall increase of contact frequencies in KO cells than Control. We observed certain variations in the numbers and sizes of loops detected at the distinct resolutions. The largest number and average sizes of loops were detected at 25 kb resolution. This is likely due to the fact that at relatively shallow mapped depth like ours, large bin size (coarse resolution) is required to detect loops. The largest average loop size at 25 kb resolution may be simply due to the largest number of loops detected at this resolution, e.g., if more loops are detected, more large loops are likely to be included. Then we consider this size variation largely computational, although it may involve in part biological and/or influence of HiC experimental and sequencing protocols.

### Computation of insulation score

The insulation scores[70] were computed by adding up normalized contact frequencies in a sliding window of 1 Mb aligned to the HiC matrix diagonal.

### Computation of eigenvector

Suppose M is an intrachromosomal normalized contact matrix of $n \times n$ size, and X($i,j$) is correlation between columns $i$ and $j$ of M ($1 \le i, j \le n$). A principal component analysis is performed on the correlation matrix, X. The eigenvector computed by juicer is the first eigenvector (principal component) of X, corresponding to the largest eigenvalue. The $\pm$ sign of an eigenvector value is arbitrarily given on each chromosome, then the sign is to be flipped according to the level of contact enrichment in a locus of the chromosome.

### Reporting summary

Further information on research design is available in the Nature Portfolio Reporting Summary linked to this article.

## Data availability

Sequencing data generated in this study have been deposited in the Gene Expression Omnibus (GEO) under accession number GSE203322 (RNA-seq, scRNA-seq, ChIP-seq, ATAC-seq) and GSE236960 (RNA-seq, ChIP-seq). The HiC sequences were deposited to DDBJ under accession DRA014202 and DRA016627. The publicly datasets reused in this paper are available in the GEO database under accession code GSE104406.

The remaining data are available within the Article, Supplementary Information or Source Data file. Source data are provided with this paper.

## Code availability

Code used to analyze the datasets in this paper is available from corresponding author upon reasonable request.

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

## Acknowledgements

This work was supported by American Society of Hematology (S.C.L. and D.I.), Japanese Society of Hematology (D.I.), Leukemia & Lymphoma Society (D.I. and O.A.-W.), NIH R01 CA242020, R01 CA251138, R01HL128239, and P50 254838 (O.A.-W.), R00 CA218896 (S.C.L.), the Edward P. Evans MDS Foundation (O.A.-W. and S.C.L.), the Vera and Joseph Dresner Foundation (S.C.L.), JSPS KAKENHI JP20H00537 (T.K.), JP20KK0184 (K.H.), JP20H03512 (K.H.), and AMED JP20cm0106166h0002 (K.H.). D.I. is also supported by Senri Life Science Foundation, The Sumitomo Foundation, The Mitsubishi Foundation, KAKETSUKEN, Bristol Myers Squibb Foundation, Astellas Foundation for Research on Metabolic Disorders, Mochida Memorial Foundation for Medical and Pharmaceutical Research, Japan Leukemia Research Fund, Princess Takamatsu Cancer Research Fund, JSPS KAKENHI (JP20H00537, JP20H03717, and 16H06279 (PAGS)) and AMED (21ck0106697h0001, 22ck0106697h0002, 23ck0106697h0003, and 23ama221126h0001). The super-computing resource was provided by Human Genome Center (the University of Tokyo) and the DDBJ super-computer system[97].

## Author contributions

D.I., O.A.-W. and M.X. designed the study. M.N., W.Z., A.K. and M.X. performed computational analyses of RNA-seq data. H.No. and A.To. performed scRNA-seq. M.N. and D.I. performed computational analyses of scRNA-seq data. S.Ka., T.A. and K.H. performed ChIP-seq and H.Ni. supervised ChIP-seq experiments. M.N. and D.I. performed computational analyses of ChIP-seq data. M.N., W.Z. and D.I. performed computational analyses of ATAC-seq data. S.Ko., D.I. and A.V. performed computational analyses of Hi-C data. M.X., M.F., T.S., T.I., H.Y., A.Ta., Y.Z., W.Z., K.N., Y.H., Y.K., Y.A., T.K., T.F., A.Y., H.I., S.C.L., S.G. and D.I. performed animal and cell line experiments. M.X., D.I., M.N., S.Ko. and O.A.-W. wrote the manuscript with approval from all co-authors.

## Competing interests

O.A.-W. has served as a consultant for H3B Biomedicine, Foundation Medicine Inc, Merck, and Janssen, Loxo Oncology/Lilly and is on the Scientific Advisory Board of Envisagenics Inc and Harmonic Discovery Inc.; O.A.-W. has received prior research funding from H3B Biomedicine, Loxo Oncology/Lilly, Minovia Therapeutics, and Nurix Therapeutics unrelated to the current manuscript. D.I. has received prior research funding from Abbvie and Sumitomo Dainippon Pharma unrelated to the current manuscript. The remaining authors declare no competing interests.

## Additional information

[1]Department of Hematology-Oncology, Institute of Biomedical Research and Innovation, Foundation for Biomedical Research and Innovation at Kobe, Kobe, Hyogo, Japan. [2]Division of Cellular Therapy, The Institute of Medical Science, The University of Tokyo, Tokyo, Japan. [3]Center for Genome Informatics, Joint Support-Center for Data Science Research, Research Organization of Information and Systems, National Institute of Genetics, Mishima, Japan. [4]Advanced Genomics Center, National Institute of Genetics, Mishima, Japan. [5]Facility for iPS Cell Therapy, CiRA Foundation, Kyoto, Japan. [6]Department of Immunology, Nagoya University Graduate School of Medicine, Nagoya, Japan. [7]Institute for Advanced Study, Nagoya University, Nagoya, Japan. [8]Center for 5D Cell Dynamics, Nagoya University Graduate School of Medicine, Nagoya, Japan. [9]Department of Hematology and Oncology, Graduate School of Medicine, Kyoto University, Kyoto, Japan. [10]Division of Systems Biology, Center for Neurological Diseases and Cancer, Graduate School of Medicine, Nagoya University, Nagoya, Japan. [11]Department of Medicine, Division of Hematology and Oncology, and Department of Genetics and Development, Columbia University Irving Medical Center, New York, NY, USA. [12]Division of Immunobiology, Research Institute for Biomedical Sciences, Tokyo University of Science, Noda, Chiba, Japan. [13]Laboratory of Immunology, Institute for Frontier Life and Medical Sciences, Kyoto University, Kyoto, Japan. [14]Division of Cancer Immunology, Research Institute/Exploratory Oncology Research & Clinical Trial Center (EPOC), National Cancer Center, Tokyo/Chiba, Japan. [15]Department of Molecular Oncology and Leukemia Program Project, Research Institute for Radiation Biology and Medicine, Hiroshima University, Hiroshima, Japan. [16]Department of Computational Biology and Medical Sciences, Graduate School of Frontier Sciences, The University of Tokyo, Chiba, Japan. [17]Tsuruoka Metabolomics Laboratory, National Cancer Center, Yamagata, Japan. [18]Department of Hematology and Rheumatology, Tohoku University Graduate School of Medicine, Sendai, Japan. [19]Laboratory Diagnostics, Tohoku University Hospital, Sendai, Japan. [20]Division of Molecular Oncology, Department of Computational Biology and Medical Sciences, Graduate School of Frontier Sciences, The University of Tokyo, Tokyo, Japan. [21]Clinical Research Division, Fred Hutchinson Cancer Center, Seattle, WA, USA. [22]Department of Laboratory Medicine and Pathology, University of Washington, Seattle, WA, USA. [23]Comparative Genomics Laboratory, National Institute of Genetics, Mishima, Japan. [24]Molecular Pharmacology Program, Sloan Kettering Institute, Memorial Sloan Kettering Cancer Center, New York, NY, USA. [25]These authors contributed equally: Muran Xiao, Shinji Kondo, Masaki Nomura. ✉e-mail: d-inoue@fbri.org

