## [Peer Review File · Nature Communications]

BRD9 determines the cell fate of hematopoietic stem cells by regulating chromatin stateREVIEWER COMMENTS

Reviewer #1 (Remarks to the Author):

The manuscript by Xiao et al. evaluates the role of BRD9, a component of the non-canonical SWI-SNF BAF complex, in normal hematopoiesis and MLL/AF9-driven leukemia. The authors present data for a new Brd9 conditional knockout mouse model in the hematopoietic system. They identify myeloid skewing, defects in B lymphopoiesis and using the retroviral MLL-AF9 model of AML show that loss of Brd9 interferes with leukemia initiation. They perform ChIP-Seq, Omni-C/Hi-C and ATAC-Seq to further investigate mechanisms by which Brd9 loss may affect gene expression, however, these lack detailed notation about numbers of replicates which currently limits interpretation and reproducibility of these studies in my mind.

The development of the Brd9 conditional knockout mouse model and its characterization is of general interest to the field. The manuscript would benefit from addition of experimental details, especially as it pertains to analysis of aged mouse models described as developing MDS, details regarding technical and biological replicates used to support reproducibility of the findings, and a more detailed methods section with sufficient detail to reproduce the work presented.

Major comments:

1. The authors should include western blots to confirm the level of depletion of BRD9 for experiments shown in Figures 1A, B and F in the different cell models.
2. In panel 1E, the authors show representative flow cytometry plots for the markers of interest, however it is unclear how many replicates were performed and the reproducibility of this result. In addition to showing the flow plots, the authors should summarize the fraction of cells within each population from each replicate experiment as was done in panel 1G.
3. The methods section would benefit from additional details. For example, the experiment described in Figure 1I is not supported with requisite information, such as number of cells that were sorted and sequenced, how many mice of what age and gender were included, and what controls were included in the qRT-PCR reactions.
4. I did not find the presentation of the scRNA-Seq data very helpful. How does panel 3G demonstrate granulocytic maturation? The authors should include more quantitative analyses of these data. Fraction of WT and KO cells should be reported for each population of interest to highlight the differences the authors claim in the text. It is difficult to determine from the images shown in Figure 3G the proportional increase or decrease in number of cells from each condition in the populations as they are highlighted.
5. The competitive transplant presented in Figure 4F is confusing. The authors did not normalize for the number of LSK or LT-HSC cells at the time of transplantation but rather injected the whole bone marrow. As a result, and in light of the data presented in Figure 3A which shows a significant reduction of LT-HSC and expansion of LSK cells, how do the authors explain that there is no difference in LT-HSC and minimal difference in LSK chimerism of Brd9 null cells competing against wild type competitor cells? What does this experiment tell us about the stem cell function of Brd9 null hematopoietic stem cells?
6. The survival figure suggesting significantly shorter latency of mice transplanted with Brd9 null bone marrow in panel 4D is very exciting. However, data describing the phenotype at the time of death for these mice is missing outside of peripheral blood smear pictures in Supplementary Figure 4. The authors should include morphologic and immunophenotypic characterization of bone marrow and spleen of both arms of this experiment at the time of sacrifice, as well as show the Brd9 excision PCR for these mice. Do these mice demonstrate dysplastic features in the bone marrow? Is there expansion of immature stem/progenitor cells? Is there abnormal myeloid differentiation?

7. Following up on my previous point, how long have the authors monitored the Brd9 conditional knockout mice after pIpC? Do these mice develop MDS in a non-transplant setting?

8. The authors do not consistently report the number of biological or technical replicate samples that were used in each experiment, including both the figure legends and their methods section. This is particularly confusing when individual data points are not shown like in figures 2E, 2G, 3H, 3J. How many biological and technical replicates were used here? How many biological replicates were used to generate average ChIP enrichment profiles in Figure 5D? It would be important to see how consistent these individual replicates are, especially for CTCF ChIP-Seq where the authors suggest a difference in signal between control and KO cells.

9. It is not clear how the correlation between CTCF-up peaks and increased gene expression was performed in Figure 5G. The authors do not state in the legend what statistical test was used, or the number of genes that fall into each category and are also up/down regulated. The authors could consider plotting the fold change in gene expression as a cumulative distribution function for each category of genes and using a KS test to show increased or decreased expression of all genes within each group.

10. The Hi-C experiments which were performed using an Omni-C library preparation (using DNase I rather than standard restriction enzymes) lack sufficient details. How many cells and how many replicates were used for these experiments? If the data presented is based on a single replicate, I am greatly concerned about the interpretation and reproducibility of these data. The legend for Figure panel 6F seems confusing – how did the authors calculate a log₂ fold change KO/control for the control loops? Did the authors actually have a control replicate to do log₂ fold change Rep1/Rep2 for the red dots/control loops? The label for the x-axis is identical for the control loops (there should be 2169 instead of 2199) and KO loops, however the position of the data points is not the same. Is this mislabeled?

Minor comments:

1. I found the flow of the paper to be somewhat confusing at times. For example, the data presented in Figure 1 is a conglomeration of different models, questions and analyses without a clear message. I would suggest better organizing these data and moving some of these data to the supplement.

2. Contrary to the title of Figure 1, none of the in vitro experiments shown evaluate the effect of BRD9 on “stemness of HSCs” which can only be evaluated using in vivo in transplant experiments, as the authors proceed to show in Figure 4. I would suggest avoiding the use of this term in this context and when describing colony forming assays, which are not stemness but rather differentiation assays.

3. Complete blood count data presented in Figures 2D, 2F, 4C and the Supplementary panel 2D need to include absolute numbers in addition to the % provided. In addition, parent populations for which % are presented need to be clearly defined on the y axes.

4. The authors should include a representative image of γ H2AX staining to accompany the quantification shown in panel 2H and indicate the number of cells imaged in each condition.

5. Y-axis labels should be included on Figures 3A and Supplemental Figure 3A.

6. The text on the plots in Figure 3C-G, 5D-H are very difficult to read; the labels are too small.

Reviewer #2 (Remarks to the Author):

Xiao et al. investigated the role of BRD9 in hematopoietic stem cell differentiation and explored the molecular mechanisms in depth. They found that loss of BRD9 altered HSC differentiation, with more myeloid commitment and fewer B cells. Single-cell RNA-seq data showed down-regulation of lymphoid- and erythroid-committed gene expression in lin⁻ cells, whereas myeloid-committed genes were up-regulated. ATAC-seq analysis revealed that loss of BRD9 increased chromatin accessibility. The authors explored the mechanism and found that BRD9 was co-localized with CTCF, and loss of BRD9 increased CTCF binding. The authors also performed Hi-C assays and found that loss of BRD9 increased intensities of CTCF-mediated chromatin loops. Finally, they also reported that BRD9 loss reduced the leukemic potential of the cells. All in all, they carried out an extensive analysis of the function of BRD9 in hematopoietic stem and progenitor cells and conducted an extensive exploration of the mechanism, which is valuable for understanding the function of BRD9. However, some data do not support its conclusions, especially the relationship between chromatin loops and gene expression.

Major comments:

Fig. 1a. Why is the starting point of cell growth at 0? If it is 0, theoretically there are no cells. Fig1b starts at 1. These need to be consistent. It is necessary to provide the detection results of knockdown efficiency, which is very important for readers to understand the results.

As shown in Fig. 1b, it is necessary to provide the detection of knockout efficiencies, such as Western blot results. The effects of the two sgRNAs are significantly different, what is the reason? If it is not due to the difference in knockout efficiency, why does the effect of different knockout positions have such a large difference in cell growth? It's interesting. Is it caused by the different functions of the deleted domains?

Fig. 1c and 1d. The third shRNA was added in the experiment, is the knockdown efficiency confirmed? The authors detected the colony-forming ability of BRD9 knockdown cells. What is the difference in significance in the graph? Is it the number of colony numbers overall or the difference in the number of each lineage? This is not very clear. The percentage of each lineage is also important data to understand the function of BRD9 in lineage differentiation and should be provided by the authors.

Fig. 1F and S1A. A scale bar is needed for microscope pictures.

Fig. 1G. Are the 6 knockouts generated by the same sgRNA as 6 clones or by 6 different sgRNAs? The author should make it clear in figure legends.

Fig.1H. The result is confusing. It is plausible that BRD9 knockdown reduced B cells and increases CD11b⁺ Gr1⁺ neutrophils. But it is expected to see that WT increased B cells while reducing neutrophils in BRD9 overexpressing cells, whereas BRD9 mutants did not. But the authors' results were that WT had no effect, while the mutants reduced B cell numbers. This doesn't seem to make sense. The authors cannot ignore this observation and should discuss it.

Figure 1 used many shRNA knockdowns and sgRNA knockouts but lacks protein-level validation data. And it's confusing: some two, some three, some six. The information needs to be provided in figure legends. This is very important for the reader to understand the results.

Figure 1. Some of the results are somewhat redundant and can be placed in supplementary figures. For example, 1A and 1B both show that BRD9 regulates cell growth, and one of them can be placed in the supplementary figure. 1C and 1D are also somewhat similar, and 1D can be placed in the supplementary figure.

Fig. 2D. Authors provided the proportion of T cells in the spleen (Fig. S2D) but not in peripheral blood. Authors claimed that myeloid-lineage skewing at the expense of B cell development. However, the data doesn't support the conclusion because B cell commitment is after the separation of myeloid and lymphoid during HSC differentiation and T cell development is relatively normal (Fig. S2D). Fig. 2F also showed that BRD9 loss didn't affect the development of pre-pro- and pro-B cells, where B cell commitment is done. The reduction of B cells that the authors observed in BRD9 deleted mice was due to impaired pro-B to pre-B transition rather than CLP commitment to B cells. We also can see that proportion of T cells increased (Fig. 2D), which raises the question of whether the increased percentage in CD11b⁺ cells is intrinsic or simply due to the decreased percentage of B cells. Altogether, there is a lack of solid evidence that BRD9 loss produces an increase in myeloid cells at the expense of B cell development.

Fig. 3A and 3B. The two panels should be merged into one, with the flow plots on the left.

Fig. 3C can be moved to supplementary.

Fig. 3D. The authors need to classify these differentially expressed genes, which are myeloid commitment genes and which are lymphoid commitment genes. It will help readers understand. Fig. 3E. The fonts of the x and y axes are too small to be legible. Add the numbers of up- and down-regulated genes. The authors describe differentially expressed genes in the article with the condition of adjusted P value < 0.05. However, Figure 3E uses an adjusted P value < 0.1. The two should be consistent.

Fig. 3G. Fig S4G described in the article should be Fig. S3G. The authors should provide a percentage of each cell subpopulation. Despite the authors' repeated emphasis on increased myeloid development, the scRNA-seq data did not support an increase in the proportion of GMP, nor was there a significant decrease in CLP. Authors need to think carefully about their data and draw a conclusion that is supported by the data.

Fig. 3H, 3I, and S3H. For scRNA-seq data, two pieces of information about differentially expressed genes are important: log fold-change of the average expression between the two groups and the percentage of cells where the feature is detected. The authors should provide the two information for these critical genes.

Fig. S3I. The authors used H3K27 acetylation ChIP-seq data to identify enhancers and super-enhancers, which help understand the mechanism of gene expression regulation. The authors mentioned that some enhancers are increased in KO, which is somewhat vague. The authors do not indicate how to analyze changes in enhancers and super-enhancers, nor do they indicate which enhancers are up-regulated and which are down-regulated in KO. How is the expression of these enhancer-related genes altered? These are important pieces of information that the authors should analyze in depth.

Fig. S3J. The authors need to mark which color is SE and which is TE, although it looks like the red color might be SE.

Fig. 4. The competitive BM transplantation assay showed that BRD9 loss lead to a substantial reduction of B and T lymphocytes, but a subtle increase of myeloid cells. No changes in CMP, GMP, and MEP were observed in KO, which is different from increased MEP observed in scRNA-seq data. Authors should explain the difference. Fig. S2D showed subtly increased T cells in the spleen of KO mice, which is the opposite of Fig. 4G. The authors cannot ignore these inconsistencies.

The authors again attributed the observed reduction in B cells caused by BRD9 loss to lineage selection, however, this is not plausible. Because the authors did observe the change of CLP, the reduction in B and T cells is more likely to be impaired lymphocyte development rather than distorted lineage selection. The authors mentioned earlier the significantly down-regulated *Dntt* gene in BRD9 deleted cells, and its function is not lineage commitment, but V(D)J rearrangement. Decreased expression of *Dntt* may prevent the normal rearrangement of antigen receptor genes in pro-B and pro-T cells, thereby impairing the development of B and T lymphocytes.

Fig. S5A. The percentage of each proportion needs to be labeled on the graph.

Fig. 6. Authors need to briefly describe Hi-C experiments in the Results section, including biological replicates, sequencing depth, etc.

Fig. 6A. It is not enough that the author only provides a small region as an example where the compartment has not changed. Authors need to provide information on overall compartments. In addition, it is necessary to mark on which chromosome this region is located.

Fig. 6C and 6D. The authors need to briefly describe how the eigenvector and insulating scores were made, and what biological significance these values represent. The high correlation of these two features between WT and KO indicates that chromatin interacting features are unchanged. Chromatin loops are highly dynamic, and both the sequencing depth and the percentages of effective reads may affect the number and length of chromatin loops. Different resolutions also produce different loop lengths and numbers. Sometimes there are large differences in loop length and number between biological repeats. How consistent is the loop between the authors' biological repeats?

The loop between enhancer and promoter is closely related to the regulation of gene expression. The authors should provide how many of these loops mediate enhancer-promoter interactions.

Fig. 6E and 6F. The results are confusing. Among the 2199 control loops, 1744 loops have increased intensity in KO, but only 548 loops are still loops in KO, so why are these loops with an increased intensity not loops in KO? What is the standard of chromatin loop? Why do the interactions with an increased intensity not meet the loop criteria? Then there must be many control loops, which have increased interaction intensity in KO but were not identified as loops in KO. So do the author's two examples in Fig 6G fit this case? Authors should explain it.

Fig. 6G. Experimental results cannot be presented in cartoon form. The plots didn't give

information about the regulation of the two genes by chromatin loops. There are several problems: 1. The schematic diagram of the gene is too vague to see the gene structure information, such as direction, promoter, and exons; 2. The information on BRD9 and CTCF occupancy and histone modification are lacking; 3. The expression information is lacking; 4. The promoter-enhancer loop cannot be seen, not just on the Hi-C heat map. In addition, the authors seem to be wrong about the relationship between chromatin loops and gene expression. An increase in chromatin interactions does not affect gene expression, only if the interaction affects the proximity of enhancers and promoters. The authors should show their RNA-seq, ChIP-seq of BRD9, CTCF, H3K27ac, and promoter-enhancer interactions (not just Hi-C heatmap) to prove that chromatin loops may directly regulate the expression of these two genes, otherwise, it is very difficult to know whether changes in gene expression are directly or indirectly regulated by BRD9. Fig. 6J and S6G S6H. For the reduced expression of the *Dntt* gene in KO cells, the authors interpreted it as excessive recruitment of transcription factor complexes to the loop-anchored gene. But this is wrong. CTCF/Cohesin-mediated chromatin loops are generated by DNA extrusion, which is a dynamic process. The increased loop intensity in KO cells is due to enhanced CTCF binding, which blocked extrusion, but this process does not necessarily lead to an increase of transcription factor complex in loop anchors. Moreover, the two loop anchors in KO near *Dntt* in the figure also did not have an obvious increase in transcription factor complexes. It is not convincing that the loop anchors compete for transcription factor complexes to repress the *Dntt* gene. It is more likely that the reduction of *Dntt* expression in KO is an indirect consequence of BRD9 deletion.

The authors' repeated emphasis on *Dntt* expression in KO cells seems to explain the reduced differentiation of HSCs into B cells. However, the authors do not seem to understand the function of the protein encoded by *Dntt*, and thus have a misunderstanding of the role of this gene expression in B cell commitment. Terminal deoxynucleotidyl transferase (TdT), also known as DNA nucleotidylexotransferase (DNNT) or terminal transferase, is a specialized DNA polymerase expressed in pro-B, pre-B, pro T, and pre-T lymphoid cells. TdT adds N-nucleotides to the V, D, and J exons of the TCR and BCR genes during antibody gene recombination, enabling the phenomenon of junctional diversity. So *Dntt* doesn't play any role in HSCs.

Fig. 7J. There is no need to put citation information on the y-axis.

Fig. 7. The authors examined the role of BRD9 in myeloid leukemia and found that BRD9 is essential for both the initiation and maintenance of myeloid leukemia. The authors also performed ATAC-seq assays and observed that BRD9 loss resulted in increased chromatin accessibility, while motif analysis of up-regulated chromatin accessible regions also revealed enrichment of CTCF. But the results cannot explain the role of BRD9 in the initiation and maintenance of leukemia, except for the known BRD9 target *myc* gene. The author needs to explain this.

Altogether, the authors present an extensive study on the role of BRD9 in HSC differentiation, providing many valuable data. But the authors' interpretation of their experimental results is not plausible, especially regarding the role of BRD9 in B cell commitment and development. The authors observed that BRD9 loss resulted in increased CTCF occupancy in the promoter region. The authors performed HiC assays to detect changes in chromatin organization. But the authors' analysis of chromatin loops has something unreasonable. The authors' examples of chromatin loops affecting expression are either incomplete or misinterpreted. Overall, the conclusion presented by the authors is unconvincing: BRD9 loss promotes myeloid lineage skewing at the expense of B cell development.

Reviewer #3 (Remarks to the Author):

In their manuscript, Xiao, Kondo, Nomura et al. showed that BRD9 is important for human HSC differentiation and stemness. Importantly, they generated a novel conditional BRD9 KO mice to study in vivo effects of BRD9 loss on haematopoiesis. Through multi-omics studies including scRNA-Seq, they identified changes at the cell types and CTCF binding upon BRD9 KO. They showed an interplay between BRD9 depletion-mediated CTCF binding and chromatin looping enhancement that result in mainly gene activation. The text and data presentation are clear, and the manuscript is likely be of interest to the fields of gene regulation, leukemia and

haematopoiesis. I have a few suggestions to help strengthen the main take-home messages.

Major Points:

1. As Brd9 KO cells are analysed 8 week after pIpC injection, this raises the concern of indirect/secondary effects of BRD9 depletion. Could authors validate their main conclusions in Figure 5F-H and 6E-J by using short term BRD9 perturbation for e.g. 12h or 24h either by BRD9 bromodomain inhibitors, BRD9 degrader or siRNA mediated BRD9 depletion in cultured cells? This would be of crucial importance to validate the authors' mechanistic explanations regarding CTCF enrichment and gene regulation.

2. The following sentence in summary is not correct: "While cBAF and PBAF regulate fates of multiple cell types, roles for ncBAF in stem cells have not been investigated." Please see <https://doi.org/10.1038/s41467-018-07528-9> which identified a role for ncBAF in regulating mouse naïve pluripotent stem cells as well as <https://www.biorxiv.org/content/10.1101/2021.05.27.445940v1> for the description of a role of BRD9 in human pluripotent stem cells. Therefore, this sentence should be revised, and the relevant references should be added to the introduction or discussion.

3. As BRD9 bromodomain was shown to be important in B-cell development and stemness in Figure1, it would be valuable to show if BRD9 bromodomain inhibitors exhibit similar effects.

4. Please revise the analysis in Figure 5B given that H3K4me1 is not a mark for poised sites if it co-occupies the regulatory element with H3K27ac. Additionally, it would be necessary to show genomic distribution of BRD9+BRG1- peaks as the authors claimed that BRD9 localized to the genome via ncBAF.

5. Could authors show the average expression levels of genes whose promoters show CTCF up, neutral and down peaks upon BRD9 loss as boxplots and perform relevant statistical tests to ensure CTCF enrichment has any correlation with gene expression (instead of Figure 5G)?

6. Similar to what authors performed for CTCF enrichment, loop formation and gene regulation upon BRD9 KO as in Figure 6H-J, could the authors perform similar analysis on YY1 (since they showed a decrease in chromatin accessibility around YY1 binding motifs upon BRD9 KO)? If this is not feasible during revision stage, could authors provide YY1 enrichment for BRD9 bound genomic loci as heatmaps (by using a publicly available data)?

7. The following sentence is not supported by data "Altogether, these findings indicate that the CTCF enrichment after BRD9 depletion occurs selectively and enhanced CTCF peaks at gene promoters results in active transcription for the observed phenotype." Could authors calculate the significance of enhanced CTCF peaks upon BRD9 loss compared to weakened CTCF peaks? Additionally, could authors calculate the significance of observed differential CTCF peaks around promoters upon BRD9 loss compared to expected ones?

Minor points:

1. Y-axis in Figure 3K should be equalized for both wild type and Brd9 KO.

REVIEWER COMMENTS

Reviewer #1 (Remarks to the Author):

The manuscript by Xiao et al. evaluates the role of BRD9, a component of the non-canonical SWI-SNF BAF complex, in normal hematopoiesis and MLL/AF9-driven leukemia. The authors present data for a new Brd9 conditional knockout mouse model in the hematopoietic system. They identify myeloid skewing, defects in B lymphopoiesis and using the retroviral MLL-AF9 model of AML show that loss of Brd9 interferes with leukemia initiation. They perform ChIP-Seq, Omni-C/Hi-C and ATAC-Seq to further investigate mechanisms by which Brd9 loss may affect gene expression, however, these lack detailed notation about numbers of replicates which currently limits interpretation and reproducibility of these studies in my mind.

The development of the Brd9 conditional knockout mouse model and its characterization is of general interest to the field. The manuscript would benefit from addition of experimental details, especially as it pertains to analysis of aged mouse models described as developing MDS, details regarding technical and biological replicates used to support reproducibility of the findings, and a more detailed methods section with sufficient detail to reproduce the work presented.

RESPONSE: We thank the Reviewer for their time, careful consideration of our comments, and positive comments. Our point-by-point responses to the Reviewer's comments are noted below.

Major comments:

1. The authors should include western blots to confirm the level of depletion of BRD9 for experiments shown in Figures 1A, B and F in the different cell models.

RESPONSE: We thank the Reviewer for this question. We have now added western blots to demonstrate the efficient reduction of BRD9 at the protein level. These data are also now described in the revised Results section (Revised Figure S1a-b, and S1k).

2. In panel 1E, the authors show representative flow cytometry plots for the markers of interest, however it is unclear how many replicates were performed and the reproducibility of this result. In addition to showing the flow plots, the authors should summarize the fraction of cells within each population from each replicate experiment as was done in panel 1G.

RESPONSE: We thank the Reviewer for this question. The data were performed in biological triplicate. Thus, we generated a bar graph with statistical significance for the triplicate independent experiments (Revised Figure S1i, CD34⁺, HSPCs; CD34⁺38⁻, more enriched HSCs; CD33⁺, myeloid-lineage cells; CD14⁺, monocytic-lineage cells). These data are also now described in the revised Results section.

3. The methods section would benefit from additional details. For example, the experiment described in Figure 1I is not supported with requisite information, such as number of cells that were sorted and sequenced, how many mice of what age and gender were included, and what controls were included in the qRT-PCR reactions.

RESPONSE: According to the Reviewer's comment, we added the indicated information for Revised Figure 1h. We collected at least 0.1M cells, used three female mice (12-week-old) per sample in biologically triplicate (9 mice were used). The FACS sorting strategy is also indicated in Revised Supplementary Data 1. For the erythroid fractions, we utilized the previously described method using CD44, Ter119, and FSC (Liu et al. Blood 2013. PMID: 23287863).

4. I did not find the presentation of the scRNA-Seq data very helpful. How does panel 3G demonstrate granulocytic maturation? The authors should include more quantitative analyses of these data. Fraction of WT and KO cells should be reported for each population of interest to highlight the differences the authors claim in the text. It is difficult to determine from the images shown in Figure 3G the proportional increase or decrease in number of cells from each condition in the populations as they are highlighted.

RESPONSE: We agree with the Reviewer that these are important points. We have now added the quantitative data on each population (Revised Figure S4h) based on the transcriptome analysis, in addition to the data of each fraction evaluated by the surface antigens by flow cytometric analysis (Revised Figure 3a and S4a). Both analyses indicated the enhanced commitment of HSCs toward myeloid-associated MPP3. Moreover, we newly performed the maturation principal component (maturation PC) analysis (Viny et al. Cell Stem Cell 2019. PMID: 31495782) to demonstrate increased expression of early myeloid commitment signatures and decreased signatures of erythroid commitment (Revised Figure 3f and S5a). These data are also now described in the revised Results section.

5. The competitive transplant presented in Figure 4F is confusing. The authors did not normalize for the number of LSK or LT-HSC cells at the time of transplantation but rather injected the whole bone marrow. As a result, and in light of the data presented in Figure 3A which shows a significant reduction of LT-HSC and expansion of LSK cells, how do the authors explain that there is no difference in LT-HSC and minimal difference in LSK chimerism of Brd9 null cells competing against wild type competitor cells? What does this experiment tell us about the stem cell function of Brd9 null hematopoietic stem cells?

RESPONSE: We thank the Reviewer for this insightful comment. We agree with the Reviewer that this is a critical point, and are very interested in maintaining the LT-HSC fraction concerning the nature of MDS. Although using whole BM cells is the standard way of competitive transplant experiments to fairly reflect the relative reduction of the cells harboring stem cell properties (Inoue et al. *Nature Genetics* 2021. PMID: 33846634; Lee et al. *Cancer Cell* 2018, PMID: 30107174; Kim et al. *Cancer Cell* 2015, PMID: 25965569), we observed comparable donor-derived chimerism in LT-HSC and LSK fractions. This was well recapitulated in another set of competitive transplants (data not shown). The decreased donor-derived chimerism in peripheral blood and spleen are due to the competitive disadvantage of Brd9 null mature B and T lymphocytes; however, chimerism amongst bone marrow mononuclear cells was comparable among the control, heterozygous KO, and homozygous KO group due to the proportion of lymphocytes being relatively lower in this population. These data are supported by additional experimental data (described in response to Reviewer #2) demonstrating that BRD9 is essential for the proliferative stages in B- and T-cell development. As for unchanged chimerism in LT-HSC, we speculate that the lower number of “phenotypic” LT-HSCs is due to the decrease of cell-cycling LT-HSCs in the Brd9 KO setting and that more dormant LT-HSCs can reconstitute the HSPC fraction in the transplant setting. Another possibility is that Brd9 KO HSCs modify the BM microenvironment to overcome their disadvantage against the normal HSCs. Indeed, we recently reported that MDS cells impair osteolineage differentiation of MSCs via MDS-derived extracellular vesicles, leading to less supportive BM niche function for normal hematopoiesis and to the relative dominance of MDS cells in the BM (Hayashi et al. *Cell Reports* 2022. PMID: 35545056). Our observation in the Brd9 KO model may motivate future research to evaluate further why and how less proliferative MDS cells became dominant against normal hematopoiesis. We have now described these points in the Discussion.

6. The survival figure suggesting significantly shorter latency of mice transplanted with Brd9 null bone marrow in panel 4D is very exciting. However, data describing the phenotype at the time of death for these mice is missing outside of peripheral blood smear pictures in Supplementary Figure 4. The authors should include morphologic and immunophenotypic characterization of bone marrow and spleen of both arms of this experiment at the time of sacrifice, as well as show the Brd9 excision PCR for these mice. Do these mice demonstrate dysplastic features in the bone marrow? Is there expansion of immature stem/progenitor cells? Is there abnormal myeloid differentiation?

RESPONSE: We agree with the Reviewer that these are important points. The mice in the KO group exhibited striking anemia and myeloid lineage skewing with dysplastic features in bone marrow and spleen, but they did not develop leukemic transformation. In immunophenotypic characterization by flow cytometry, we found a significant expansion of GMP and myeloid-biased MPP3 cells (contrary to the reduced lymphoid-biased MPP4 cells), in line with the KO phenotype shown in Figure 2 and Figure 3. We also observed differentiation block at the level of metamyelocytes (group IV) evaluated by Ly6G/c-Kit staining. Moreover, histological analysis of the spleen in the KO group exhibited destruction of normal spleen architecture with the infiltration of CD11b⁺ cells, occupying a large part of spleen cells. We also confirmed that the perfect Brd9 depletion was maintained at the time of analysis. We have now added these data in Revised Supplementary Figure 6.

7. Following up on my previous point, how long have the authors monitored the Brd9 conditional knockout mice after plpC? Do these mice develop MDS in a non-transplant setting?

RESPONSE: We thank the Reviewer for this insightful comment. The primary mice developed significant myelodysplasia (Revised Figure S3b); however, the mice did not die of severe anemia during the observation period of ~12 months. Regarding the fact that the transplanted mice died earlier (Figure 4d), one possible explanation for this is prolonged stress condition under the transplant setting (such as lethal irradiation) may enhance the phenotype of Brd9 KO. We have now added these points in Discussion.

8. The authors do not consistently report the number of biological or technical replicate samples that were used in each experiment, including both the figure legends and their methods section. This is particularly confusing when individual data points are not shown like in figures 2E, 2G, 3H, 3J. How many biological and technical replicates were used here? How many biological replicates were used to generate average ChIP enrichment profiles in Figure 5D? It would be important to see how consistent these individual replicates are, especially for CTCF ChIP-Seq where the authors suggest a difference in signal between control and KO cells.

RESPONSE: Based on the suggestions, we modified the manuscript and figures as described below.

Figure 2E: We showed the individual data points (Figure 2e) and we have now added the independent experimental data of colony formation assay in Revised Figure S2f.

Figure 2G: The original data are derived from 4 biological replicated samples. Thus, we believe that our findings are highly reliable. We also showed the individual data points in Revised Figure 2h.

Figure 3H: Since the data are obtained from scRNA-seq analysis from the combined control and KO Lineage^{Kit}⁺ BM cells (3 mice into 1 sample each), we newly FACS-sorted MPP4 fraction (2 control vs. 3 KO, biological replication) and performed qRT-PCR of *Ilf7* and *Dntt* mRNA. The reduction of these genes was recapitulated by the qRT-PCR (Revised Figure 5b).

Figure 3J: We newly performed ATAC-seq (two independent biological samples in Control vs. KO Lineage⁻ BM cells) and obtained similar results as we presented in the original Figure 3J. We added the information derived from the independent ATAC dataset to describe the increase of ATAC signals at super-enhancers and typical-enhancers in KO condition (Revised Figure 5f). Performing the analysis of variance, we demonstrated that the normalized ATAC signals in the Brd9 KO condition were significantly enriched compared to Control both in SE (super-enhancers) and TE (typical-enhancers) ($p < 2e-16$). We also newly generated H3K27Ac ChIP-seq data (control and KO Lineage⁻ BM cells). We confirmed that the original H3K27Ac rank/signal plots were reproducible by combining the originally mapped sequencing reads and new sequencing data (Revised Figure S5g).

Figure 5D: We newly performed another set of CTCF ChIP-seq analyses and further demonstrated that the CTCF increment was consistent in the two biologically independent samples (Figure 5 and Revised Figure S7g). Moreover, based on the comment of Reviewer #3, we carried out CTCF ChIP-seq of K562 cells (human leukemia cell line) a few days after BRD9 depletion and demonstrated the enrichment of CTCF peaks, especially at opened chromatin evaluated by ATAC-seq (Revised Figure S8a-d). These data indicate the robust relationship between BRD9-CTCF alteration.

9. It is not clear how the correlation between CTCF-up peaks and increased gene expression was performed in Figure 5G. The authors do not state in the legend what statistical test was used, or the number of genes that fall into each category and are also up/down regulated. The authors could consider plotting the fold change in gene expression as a cumulative distribution function for each category of genes and using a KS test to show increased or decreased expression of all genes within each group.

RESPONSE: We agree with the Reviewer that these are essential points and we apologize for the confusion. First, we utilized two-sided t-test to evaluate the average value of log₂ fold changes compared to 0 in Figure 5g, where the altered peaks in the Up group located at promoter-TSS sites were associated with enhanced transcription ($p=1.2838e-08$). Next, according to the Reviewer's suggestion, we added the violin plot figure to show a distribution of log₂ fold-change on gene expression for each category (revised Figure S7j). When we performed a two-sided KS test as suggested, we observed a modest difference in distributions of log₂ fold change of gene expressions between CTCF-up and CTCF-neutral ($p=0.062$). However, we could not see much difference between CTCF-up and CTCF-down groups ($p=0.457$). The reason for this is basically due to the limited sample size in CTCF-down (<100).

10. The Hi-C experiments which were performed using an Omni-C library preparation (using DNase I rather than standard restriction enzymes) lack sufficient details. How many cells and how many replicates were used for these experiments? If the data presented is based on a single replicate, I am greatly concerned about the interpretation and reproducibility of these data. The legend for Figure panel 6F seems confusing – how did the authors calculate a log₂ fold change KO/control for the control loops? Did the authors actually have a control replicate to do log₂ fold change Rep1/Rep2 for the red dots/control loops? The label for the x-axis is identical for the control loops (there should be 2169 instead of 2199) and KO loops, however the position of the data points is not the same. Is this mislabeled?

RESPONSE: We thank the Reviewer for the insightful comments. To further help readers understand the HiC experiments, first, we summarized the features/advantages of the Omni-C HiC protocol in the revised Methods section (as noted below). We also modified the legend for Figure 6f (as described below). As the Reviewer pointed out, the number of Control loops was 2199, not 2169 (we apologize for the incorrect number). This number was updated to 2,239 after merging the three biological replicates.

We independently prepared three biological replicates of Omni-C HiC libraries from 1M cells each. Since we obtained sufficiently large numbers of dereplicated read-pairs in both Control and KO cells in the first biological replicate, we could detect 2199 and 4640 chromatin loops in Control and KO cells, respectively. We then measured the data reproducibility between the three biological replicates by comparing the overall profiles of the loop intensities over the loops detected in Control and KO cells in the first biological replicate as described in the Methods section (reproducibility of Control and KO HiC maps based on aggregate analysis of contact frequencies over the chromatin loops) and Supplementary Fig. 9. We merged the three biological replicates into a single set and created a contact map (hic file) based on each merged dataset. The loop intensities, insulation scores, and eigenvectors were computed based on the contact map in each sample. The chromatin loops and TADs were determined based on the loop intensities and insulation scores in each Control and KO cells.

(Omni-C HiC library preparation)

While conventional HiC approaches use restriction enzymes (RE) to digest chromatin, the Omni-C HiC library preparation uses a sequence-independent endonuclease for chromatin digestion prior to proximity ligation and library generation. The sequence independent chromatin fragmentation of Omni-C raises the coverage for detecting chromatin contacts by up to 20% of the genome with low restriction enzyme density, and thus increases the enrichment in long-range *cis* reads.

(Figure 6f legend)

(f) Fold change of loop intensities from Control to KO cells of chromatin loops detected in Control (left, red) and KO (right, blue) cells. X and y axes represent the loop intensities detected in Control and the fold change of loop intensities (KO/Control), respectively.

Minor comments:

1. I found the flow of the paper to be somewhat confusing at times. For example, the data presented in Figure 1 is a conglomeration of different models, questions and analyses without a clear message. I would suggest better organizing these data and moving some of these data to the supplement.

RESPONSE: According to the comments of Reviewer 1 and Reviewer 2, we moved several data previously located in Figure 1 into Figure S1.

2. Contrary to the title of Figure 1, none of the in vitro experiments shown evaluate the effect of BRD9 on “stemness of HSCs” which can only be evaluated using in vivo in transplant experiments, as the authors proceed to show in Figure 4. I would suggest avoiding the use of this term in this context and when describing colony forming assays, which are not stemness but rather differentiation assays.

RESPONSE: We agree with the Reviewer and modified the manuscript accordingly.

3. Complete blood count data presented in Figures 2D, 2F, 4C and the Supplementary panel 2D need to include absolute numbers in addition to the % provided. In addition, parent populations for which % are presented need to be clearly defined on the y axes.

Describe the absolute number and parent populations.

RESPONSE: We added the figures for absolute numbers in revised Figure S2d, S2e, S2j, and S6a and modified the Y axes accordingly.

4. The authors should include a representative image of γ H2AX staining to accompany the quantification shown in panel 2H and indicate the number of cells imaged in each condition.

RESPONSE: We added the representative γ H2AX staining panel in Figure S3d. We evaluated 522 and 741 cells in control and KO Lineage-Kit⁺ BM cells, respectively.

5. Y-axis labels should be included on Figures 3A and Supplemental Figure 3A.

RESPONSE: We added the Y-axis labels in revised Figure 3a and S4a accordingly.

6. The text on the plots in Figure 3C-G, 5D-H are very difficult to read; the labels are too small.

RESPONSE: We modified accordingly.

Reviewer #2 (Remarks to the Author):

Xiao et al. investigated the role of BRD9 in hematopoietic stem cell differentiation and explored the molecular mechanisms in depth. They found that loss of BRD9 altered HSC differentiation, with more myeloid commitment and fewer B cells. Single-cell RNA-seq data showed down-regulation of lymphoid- and erythroid-committed gene expression in lin- cells, whereas myeloid-committed genes were up-regulated. ATAC-seq analysis revealed that loss of BRD9 increased chromatin accessibility. The authors explored the mechanism and found that BRD9 was co-localized with CTCF, and loss of BRD9 increased CTCF binding. The authors also performed Hi-C assays and found that loss of BRD9 increased intensities of CTCF-mediated chromatin loops. Finally, they also reported that BRD9 loss reduced the leukemic potential of the cells. All in all, they carried out an extensive analysis of the function of BRD9 in hematopoietic stem and progenitor cells and conducted an extensive exploration of the mechanism, which is valuable for understanding the function of BRD9. However, some data do not support its conclusions, especially the relationship between chromatin loops and gene expression.

RESPONSE: We thank the Reviewer for the careful evaluation of our manuscript. We have now addressed the points raised in the initial review below.

Major comments:

Fig. 1a. Why is the starting point of cell growth at 0? If it is 0, theoretically there are no cells. Fig1b starts at 1. These need to be consistent. It is necessary to provide the detection results of knockdown efficiency, which is very important for readers to understand the results.

RESPONSE: We apologize to the Reviewer for this oversight. In the original Figure 1a, the starting point of cell growth was mistaken as “0.1”. Thus, we have now revised the Y-axis scale of Figure 1a, where the value at day 0 started at 1.0 (although it may appear that the very left circle is on 0, it actually is on 1).

As shown in Fig. 1b, it is necessary to provide the detection of knockout efficiencies, such as Western blot results. The effects of the two sgRNAs are significantly different, what is the reason? If it is not due to the difference in knockout efficiency, why does the effect of different knockout positions have such a large difference in cell growth? It's interesting. Is it caused by the different functions of the deleted domains?

RESPONSE: We thank the Reviewer for this insightful suggestion. To evaluate if the sgRNA-targeted sites may affect the impact on the growth, we compared the consequences of distinct sgRNAs targeting the bromodomain (BD) or DUF domain. Although we obtained perfect KO in both cells (as shown in revised Figure 1b), we observed that BD-targeting had a robust effect on growth. Similar to the effect of BRD9 protein on B-cell differentiation, these data suggest that the BD is essential for the role of BRD9 in the proliferating state.

Fig. 1c and 1d. The third shRNA was added in the experiment, is the knockdown efficiency confirmed? The authors detected the colony-forming ability of BRD9 knockdown cells. What is the difference in significance in the graph? Is it the number of colony numbers overall or the difference in the number of each lineage? This is not very clear. The percentage of each lineage is also important data to understand the function of BRD9 in lineage differentiation and should be provided by the authors.

RESPONSE: The efficient knockdown efficiency was shown in revised Figure S1c and f. We also added the statistical significance in the number of each lineage (revised Figure 1c and S1e) and the percentage of each lineage (revised Figure S1d and S1g).

Fig. 1F and S1A. A scale bar is needed for microscope pictures.

RESPONSE: We revised the figures accordingly (revised Figure 1e and S1h).

Fig. 1G. Are the 6 knockouts generated by the same sgRNA as 6 clones or by 6 different sgRNAs? The author should make it clear in figure legends.

RESPONSE: We used two independent sgRNAs and expanded three clones per sgRNA. We clarified this point and added the WB pictures in revised Figure S1j.

Fig.1H. The result is confusing. It is plausible that BRD9 knockdown reduced B cells and increases CD11b+ Gr1+ neutrophils. But it is expected to see that WT increased B cells while reducing neutrophils in BRD9 overexpressing cells, whereas BRD9 mutants did not. But the authors' results were that WT had no effect, while the mutants reduced B cell numbers. This doesn't seem to make sense. The authors cannot ignore this observation and should discuss it.

RESPONSE: We thank the Reviewer for this suggestion. Our data suggest that the endogenous level of BRD9 is sufficient for B-cell development and maturation and that the B-cell phenotype is observed only when BRD9 is depleted or functionally impaired. Moreover, the well-studied BRD9 inhibitor (BI-7273, Martin et al. J Med Chem 2016. PMID: 26914985) reduced the pre-B colonies and pre-B cell proportion in a dose-dependent manner, indicating the functional roles of the bromodomain in B-cell development (revised Figure 2f and S2h). We added these points to the Discussion and revised Figure S2g.

Figure 1 used many shRNA knockdowns and sgRNA knockouts but lacks protein-level validation data. And it's confusing: some two, some three, some six. The information needs to be provided in figure legends. This is very important for the reader to understand the results.

RESPONSE: We revised the figures accordingly and added the WB data in revised Figure S1a-c, and f.

Figure 1. Some of the results are somewhat redundant and can be placed in supplementary figures. For example, 1A and 1B both show that BRD9 regulates cell growth, and one of them can be placed in the supplementary figure. 1C and 1D are also somewhat similar, and 1D can be placed in the supplementary figure.

RESPONSE: According to the comments of Reviewer 1 and Reviewer 2, we moved several data, such as original Figure 1D into revised Figure S1.

Fig. 2D. Authors provided the proportion of T cells in the spleen (Fig. S2D) but not in peripheral blood. Authors claimed that myeloid-lineage skewing at the expense of B cell development. However, the data doesn't support the conclusion because B cell commitment is after the separation of myeloid and lymphoid during HSC differentiation and T cell development is relatively normal (Fig. S2D). Fig. 2F also showed that BRD9 loss didn't affect the development of pre-pro- and pro-B cells, where B cell commitment is done. The reduction of B cells that the authors observed in BRD9 deleted mice was due to impaired pro-B to pre-B transition rather than CLP commitment to B cells. We also can see that proportion of T cells increased (Fig. 2D), which raises the question of whether the increased percentage in CD11b+ cells is intrinsic or simply due to the decreased percentage of B cells. Altogether, there is a lack of solid evidence that BRD9 loss produces an increase in myeloid cells at the expense of B cell development.

RESPONSE: We agree with the Reviewer that these are essential points. Firstly, the results of the differentiation assay of HSCs (revised Figure 1d) and AML cells (revised Figure 1e) indicate the intrinsic effects of BRD9 loss on promoting myeloid differentiation, which is supported by the maturation PC analysis (described below in revised Figure 3f) of scRNA-seq data. Secondly, as the Reviewer suggested, we agree that B cell reduction is attributed to the differential block after B cell lineage commitment rather than impaired CLP commitment to B cells, which is supported by the fact that CLP proportion was unaffected (revised Figure S4a). Notably, the block occurred at the expansion stage between PreB to ProB and the cell-cycling population (S phase) was found to be decreased at the stage of ProB cells or later (revised Figure S2k). Thus, we modified our description like "myeloid-lineage skewing at the expense of B cell development."

Similarly, in the thymus of transplanted mice, we also detected that T cell development was severely impaired at the DN1-DN2/3 transition stage where rapid expansion of cells is required (data shown below), although the ratio of splenic CD3⁺ cells was not affected (revised Figure S2e). Given that T cell development mainly occurs at the thymus, these findings indicate roles of BRD9 in the maturation of both B and T cells. Indeed, Brd9 KO LSK cells hardly grow and differentiate into mature B- or T-cells under specific cytokine conditions (data shown below). These data are part of our ongoing studies evaluating a potential role for BRD9 in immune recognition of tumor cells. As the Reviewer suggested, we understand that the unaffected CLP commitment does not result in the increase in myeloid cells, while BRD9 severely affects lymphoid expansion stages. Based on the Reviewer's insightful and reasonable comments, we deleted the term "at the expense of B cell development" in the Abstract and Manuscript.

DN1 (CD4-CD8-CD25-CD44+), DN2 (CD4-CD8-CD44+CD25+), DN3 (CD4-CD8-CD25+CD44-), and DN4 (CD4-CD8-CD25-CD44-).

Fig. 3A and 3B. The two panels should be merged into one, with the flow plots on the left.

RESPONSE: We revised the figures accordingly (revised Figure 3a).

Fig. 3C can be moved to supplementary.

RESPONSE: We revised the figures accordingly (revised Figure S4d).

Fig. 3D. The authors need to classify these differentially expressed genes, which are myeloid commitment genes and which are lymphoid commitment genes. It will help readers understand.

RESPONSE: Given that it was hard to classify these top differentially expressed genes (DEGs, revised Figure 3b) into two groups, myeloid vs. lymphoid, since the RNA-seq was performed using bulk Lineage-Kit⁺ HSPC fraction and DEGs contain a variety of pathways. Instead, we analyzed transcriptional factor (TF) programs using myeloid and lymphoid-committed multi-potent progenitors, called MPP3 and MPP4, respectively (revised Figure 3h). Moreover, the enhanced myeloid commitment was supported by the maturation principal component (PC) analysis (described below in revised Figure 3f) of scRNA-seq data. These data prove that myeloid and lymphoid programs are up- and down-regulated, respectively, at the MPP level or lineage-committed HSCs.

Fig. 3E. The fonts of the x and y axes are too small to be legible. Add the numbers of up-and down-regulated genes. The authors describe differentially expressed genes in the article with the condition of adjusted P value < 0.05. However, Figure 3E uses an adjusted P value < 0.1. The two should be consistent.

RESPONSE: We revised the figures accordingly. We consistently used an adjusted P value < 0.1 and revised the related description in “BRD9 loss transcriptionally alters the cell fate specification of HSCs *in vivo*” section of the manuscript (744 genes, $\log_2FC > \log_2(1.5)$; 277 genes, $\log_2FC < -\log_2(1.5)$) (revised Figure 3c).

Fig. 3G. Fig S4G described in the article should be Fig. S3G. The authors should provide a percentage of each cell subpopulation. Despite the authors' repeated emphasis on increased myeloid development, the scRNA-seq

data did not support an increase in the proportion of GMP, nor was there a significant decrease in CLP. Authors need to think carefully about their data and draw a conclusion that is supported by the data.

RESPONSE: We agree with the Reviewer that these are important points. As the Reviewer suggested, we could not show that the number of the GMP/CLP fraction estimated by single-cell transcriptome clustering analysis was robustly affected. Thus, we calculated the number and proportion of multi-potent progenitors (MPPs) to determine the lineage bias at more immature progenitors (revised Figure S4h). MPP4 was not found to be decreased, suggesting that myeloid lineage skewing is not simply attributed to the impaired lymphoid-lineage commitment as described above. To further demonstrate the differentiation program alteration, we newly performed the maturation principal component (maturation PC) analysis to demonstrate increased expression of early myeloid commitment signatures and decreased signatures of erythroid commitment (revised Figure 3f and S5a). These data are also now described in the revised Results section.

Fig. 3H, 3I, and S3H. For scRNA-seq data, two pieces of information about differentially expressed genes are important: log fold-change of the average expression between the two groups and the percentage of cells where the feature is detected. The authors should provide the two information for these critical genes.

RESPONSE: We added the data of log2 fold-change for *Ilf7r* (-3.52) and *Dntt* (-2.68) mRNA expression in Figure 3g ($p=3.37e-7$ and $p=3.98e-10$, respectively). The significance was calculated by a negative binomial test. We also added the number of cells where the feature is detected, such as MPP4 (revised Figure S4h). Interestingly, although the number of MPP4 is comparable between the control and KO group, these two genes were severely downregulated in the KO MPP4, which was reproduced by qRT-PCR in FACS-sorted MPP4 (revised Figure S5b). As for revised Figures 3h and S5c, the TF enrichment data are based on the genes positively or negatively correlated with BRD9 KO in the indicated clusters of UMAP analysis. To show the DEGs for revised Figure 3h and S5c, we added Supplementary Data 4.

Fig. S3I. The authors used H3K27 acetylation ChIP-seq data to identify enhancers and super-enhancers, which help understand the mechanism of gene expression regulation. The authors mentioned that some enhancers are increased in KO, which is somewhat vague. The authors do not indicate how to analyze changes in enhancers and super-enhancers, nor do they indicate which enhancers are up-regulated and which are down-regulated in KO. How is the expression of these enhancer-related genes altered? These are important pieces of information that the authors should analyze in depth.

RESPONSE: We thank the Reviewer for the comment. The locations of super-enhancers and typical-enhancers were called using the findPeaks module in HOMER with parameters -style super -L 0 -fdr 0.001 as mentioned in the “ChIP-seq” portion of the Methods section. We newly analyzed differential peaks using getDifferentialPeaks module in HOMER with parameters -F 1.5 -P 0.05. For example, we found that 15 locations in SEs were up under KO conditions. Using the nearest gene name of each peak, we extracted eight genes, such as *Myb*, upregulated in the BRD9 KO condition. We also newly generated H3K27Ac ChIP-seq data (control and KO Lineage⁻ BM cells). We confirmed that the original H3K27Ac rank/signal plots were reproduced by combining the originally mapped sequencing reads and new ones. This information is now shown in a revised Figure S5g.

Fig. S3J. The authors need to mark which color is SE and which is TE, although it looks like the red color might be SE.

RESPONSE: We revised the figures accordingly: SE, red; TE, blue (Revised Figure S5d).

Fig. 4. The competitive BM transplantation assay showed that BRD9 loss lead to a substantial reduction of B and T lymphocytes, but a subtle increase of myeloid cells. No changes in CMP, GMP, and MEP were observed in KO, which is different from increased MEP observed in scRNA-seq data. Authors should explain the difference. Fig. S2D showed subtly increased T cells in the spleen of KO mice, which is the opposite of Fig. 4G. The authors cannot ignore these inconsistencies.

The authors again attributed the observed reduction in B cells caused by BRD9 loss to lineage selection, however, this is not plausible. Because the authors did observe the change of CLP, the reduction in B and T cells is more likely to be impaired lymphocyte development rather than distorted lineage selection. The authors mentioned earlier the significantly down-regulated *Dntt* gene in BRD9 deleted cells, and its function is not lineage commitment, but V(D)J rearrangement. Decreased expression of *Dntt* may prevent the normal rearrangement of antigen receptor genes in pro-B and pro-T cells, thereby impairing the development of B and T lymphocytes.

RESPONSE: We thank the Reviewer for this critical point. First, our scRNA-seq data clearly indicated the decreased MEP and impaired erythroid commitment based on revised Figure 3e and S5a (maturation PC analysis), consistent with the FACS analysis result in revised Figure 3a. Please note that we did not show increased MEP data in the original figures or manuscript. As the Reviewer pointed out, we found discrepant

results of the donor-derived chimerism in the competitive BM transplant (Figure 4f-h) with those of primary mice, such as HSPC fractions and CD3⁺ fractions (Figure 3a and revised Figure 2e and 4b). However, the cell-autonomous effects of *Brd9* loss in the development of myeloid and B cells were consistent, all of which were well recapitulated in the latest competitive transplant (data not shown). The decreased donor-derived chimerism in PB and spleen are mainly due to the competitive disadvantage of mature B/T lymphocytes; however, the chimerism in the whole BM was comparable among the control, heterozygous KO, and homozygous KO groups. Based on our data showing that BRD9 is essential for the proliferative stages (e.g., B/T development) despite the increased S phase in BRD9 KO LSK cells, we speculate that the lower number of “phenotypic” and dormant LT-HSC can reconstitute the HSPC fraction in the transplant setting. Such observation will hopefully stimulate future research to evaluate why and how less proliferative MDS cells became dominant over normal hematopoiesis. We have now described these points in Discussion.

As the Reviewer mentioned, we agree that the remarkably decreased *Dntt* in KO may prevent the normal rearrangement of antigen receptor genes in pro-B and pro-T cells, thereby impairing the development of B and T lymphocytes after lineage commitment. In addition to our extensive B-lineage data showing the block at the preB to proB stage with decreased S phase (revised Figure S2k), we precisely evaluated T cell development in the thymus of transplanted mice. Of interest, T cell development (pro-T or later) was significantly impaired at the DN1-DN2/3 stage in the KO group, although the ratio of splenic CD3⁺ cells was not affected (revised Figure S2e). Based on these findings, we revised the manuscript accordingly not to emphasize the role of *Dntt* in lineage commitment. We have also described these points in the revised Discussion.

Fig. S5A. The percentage of each proportion needs to be labeled on the graph.

RESPONSE: We revised the figures accordingly (revised Figure S7a).

Fig. 6A. Authors need to briefly describe Hi-C experiments in the Results section, including biological replicates, sequencing depth, etc.

RESPONSE: We thank the Reviewer for the insightful comments about Figure 6. To further help readers understand the HiC experiments, we modified the figures and manuscript accordingly. The point-by-point response is described below.

Fig. 6A. It is not enough that the author only provides a small region as an example where the compartment has not changed. Authors need to provide information on overall compartments. In addition, it is necessary to mark on which chromosome this region is located.

RESPONSE: We replaced this plot with one for the whole chr1. We also clarified that the eigenvectors to define A/B compartments and insulation scores to define TADs are highly conserved genome-genome, as shown in Revised Figure 6c and 6d, respectively.

Fig. 6C and 6D. The authors need to briefly describe how the eigenvector and insulating scores were made, and what biological significance these values represent. The high correlation of these two features between WT and KO indicates that chromatin interacting features are unchanged.

RESPONSE: We briefly described the definitions and utilities (biological significance) of insulation scores and Eigenvectors in the text. We also presented how to compute them in Methods section. These are noted below:

Methods [Computation of Insulation score]

Insulation scores, i.e., the aggregates of normalized contact frequencies in a sliding window of a certain size (we used 1 Mb) along the HiC matrix diagonal measure degrees of enrichment of chromatin contacts in a given genomic region (Crane et al. Nature. 2015, PMID: 26030525). Minima and peaks of the insulation score profile denote areas of high and low insulation, e.g., boundaries and centers of TADs, respectively. The insulation scores were computed by adding up normalized contact frequencies in a sliding window of 1 Mb aligned to the HiC matrix diagonal.

Methods [Computation of eigenvector]

The eigenvector computed by juicer, e.g., the first eigenvector (principal component) of the correlation matrix of normalized chromatin contacts (binned at 10 kb resolution) clusters the binned genomic regions according to the enrichment profiles of chromatin contacts of a genomic region clusters with surrounding regions, and basically splits the genomic regions into active (A) and inactive (B) compartments in terms of chromatin contacts. As for the computation of the eigenvector, suppose M is an intrachromosomal normalized contact matrix of $n \times n$ size, and $X(i,j)$ is the correlation between columns i and j of M ($1 \leq i, j \leq n$). A principal component analysis is performed on the correlation matrix, X . The eigenvector computed by juicer is the first eigenvector (principal component) of X , corresponding to the largest eigenvalue. The +/- sign of an eigenvector value is arbitrarily given on each chromosome; then the sign is to be flipped according to the level of contact enrichment in a locus of the chromosome.

Chromatin loops are highly dynamic, and both the sequencing depth and the percentages of effective reads may affect the number and length of chromatin loops. Different resolutions also produce different loop lengths and numbers. Sometimes there are large differences in loop length and number between biological repeats. How consistent is the loop between the authors' biological repeats?

RESPONSE: We identified loops for ~3/4 and half the sizes of the Control and KO datasets of the first biological replicate, and the numbers and sizes of the called loops are presented in revised Figure S10b and Tables S2-3. The numbers and sizes of the loops called at three distinct resolutions, 5kb, 10kb, and 25kb are presented for the merged Control and KO datasets in revised Figure S10c and Table S4. We did not find notable changes in loop sizes at the three distinct depths and three resolutions between Control and KO cells. At the same time, we used the overall profile of normalized contact frequencies (loop intensities) of the second and third biological replicates over the Control and KO loops called in the first biological replicate to verify the reproducibility of HiC across the three biological replicates (this is noted in the Methods section “Reproducibility of Control and KO HiC maps based on aggregate analysis of contact frequencies over the chromatin loops” and revised Figure S9).

Supplementary Figure 9

The loop between enhancer and promoter is closely related to the regulation of gene expression. The authors should provide how many of these loops mediate enhancer-promoter interactions.

RESPONSE: We presented the numbers of enhancer-promoter loops (934 Control and 1847 KO loops) in the text. We have also now added the information in the Result section.

Fig. 6E and 6F. The results are confusing. Among the 2199 control loops, 1744 loops have increased intensity in KO, but only 548 loops are still loops in KO, so why are these loops with an increased intensity not loops in KO? What is the standard of chromatin loop? Why do the interactions with an increased intensity not meet the loop criteria? Then there must be many control loops, which have increased interaction intensity in KO but were not identified as loops in KO. So do the author's two examples in Fig 6G fit this case? Authors should explain it.

RESPONSE: To help readers understand the strategies used, we summarized the threshold parametrization used in Juicer to call chromatin loops in the Methods section entitled "Identification of chromatin loops" as described below. Of the 2,239 Control loops (based on the final merged Control dataset), 1,878 loops overlapped KO loops in both anchors (common loops called in both Control and KO cells). 361 did not overlap with KO loops in both anchors (Control specific loops). The ratio (KO/Control) of loop intensities of the Control specific loops was 0.99 ± 0.2 while that of the common loops was 1.27 ± 0.26 (Mann-Whitney test $p < 5.23e-71$). We used loop intensity as a measure to indicate the overall increase of contact frequencies in groups of loops. The loop call threshold parametrization of Juicer, however, involves additional thresholds such as the raise of loop intensity of the peak pixel relative to the four neighborhood regions as described below [(1) pixels to its lower-left, (2) pixels to its left and right, (3) pixels above and below, and (4) a donut surrounding the pixel of interest]. Then some discrepancy between the up and down of loop intensities and loop calling occurs in certain loops. Loop intensities increased in 170 (47%) of the 361 Control specific loops, but they were not called as loops in KO cells. These 170 are the Control specific loops not called in KO cells, although their loop intensities were raised. Please note that the loop intensity increased in a much more significant portion, 1,625 (87%) of the 1878 common loops.

Methods [Identification of chromatin loops]

Based on the filtered alignment of HiC paired-end reads, Juicer computed observed contact frequencies (O) at 5kb, 10kb, and 25kb resolutions (Rao et al. Cell. 2014. PMID: 25497547). The observed contact frequencies were normalized by expected contact frequencies (E) based on Knight and Ruiz (2012), and thus normalized contact matrices containing O/E ratio (loop intensity) in each pixel were built at each resolution. By comparing the loop intensity in a given pixel to loop intensities in four neighborhood regions [(1) pixels to its lower left, (2) pixels to its left and right, (3) pixels above and below, and (4) a donut surrounding the pixel of interest (Figure 3A in Rao et al. Cell. 2014. PMID: 25497547)], Juicer identified pixels of enriched contacts requiring that the pixel being tested contain at least 50% more contacts than expected based on each neighborhood region and the enrichment be statistically significant after correcting for multiple hypothesis testing (False Discovery Rate < 10%). The resulting enriched pixels tend to form contiguous interaction regions comprising 5 to 20 pixels each. Of the contiguous interaction region, the pixel with the highest loop intensity was defined as the 'peak pixel', which tethers together two loci on the same chromosome or equivalently forms a chromatin loop. The two loci are called "loop anchors" (Rao et al. Cell. 2017. PMID: 28985562). The loops whose anchors overlapped were merged into a single loop with the largest loop intensity.

Fig. 6G. Experimental results cannot be presented in cartoon form. The plots didn't give information about the regulation of the two genes by chromatin loops. There are several problems: 1. The schematic diagram of the gene is too vague to see the gene structure information, such as direction, promoter, and exons; 2. The information on BRD9 and CTCF occupancy and histone modification are lacking; 3. The expression information is lacking; 4. The promoter-enhancer loop cannot be seen, not just on the Hi-C heat map. In addition, the authors seem to be wrong about the relationship between chromatin loops and gene expression. An increase in chromatin interactions does not affect gene expression, only if the interaction affects the proximity of enhancers and promoters. The authors should show their RNA-seq, ChIP-seq of BRD9, CTCF, H3K27ac, and promoter-enhancer interactions (not just Hi-C heatmap) to prove that chromatin loops may directly regulate the expression of these two genes, otherwise, it is very difficult to know whether changes in gene expression are directly or indirectly regulated by BRD9.

RESPONSE: Regarding the relationship between gene expression with the formation of chromatin loops bridging promoters and enhancers (EP loops), please note that the loop-anchored genes containing the genes with EP loops are associated with enhanced transcription on a genome-wide scale (revised Figure 6i and Supplementary Figure 11f-i). We also added the new data showing that the transcription of EP loop-anchored genes increased compared to that of non-loop-anchored genes (revised Figure S11i). These findings agree remarkably with the correlation between EP loops and the gene expression reported by Bonev et al. (Cell. 2017. PMID: 29053968).

According to your insightful suggestions, we presented the combined mapping data of HiC, RNA-seq, and ChIP-seq for BRD9, CTCF, and H3K27ac in two gene locus, *Igfbp7* (revised Figure 6h) and *Hgfac* (revised Supplementary Figure 11e). Our analysis demonstrates that the genome-wide knockout of *Brd9* significantly impacts the increment of CTCF and EP loops. For example, at the BRD9-binding region of the *Igfbp7* and *Hgfac* locus, mRNA transcription in the H3K27ac active region is enhanced, which was compatible with the upregulation of chromatin loops and CTCF caused by *Brd9* deletion.

Fig. 6J and S6G S6H. For the reduced expression of the Dntt gene in KO cells, the authors interpreted it as excessive recruitment of transcription factor complexes to the loop-anchored gene. But this is wrong. CTCF/Cohesin-mediated chromatin loops are generated by DNA extrusion, which is a dynamic process. The increased loop intensity in KO cells is due to enhanced CTCF binding, which blocked extrusion, but this process does not necessarily lead to an increase of transcription factor complex in loop anchors. Moreover, the two loop anchors in KO near Dntt in the figure also did not have an obvious increase in transcription factor complexes. It is not convincing that the loop anchors compete for transcription factor complexes to repress the Dntt gene. It is more likely that the reduction of Dntt expression in KO is an indirect consequence of BRD9 deletion. The authors' repeated emphasis on Dntt expression in KO cells seems to explain the reduced differentiation of HSCs into B cells. However, the authors do not seem to understand the function of the protein encoded by Dntt, and thus have a misunderstanding of the role of this gene expression in B cell commitment. Terminal deoxynucleotidyl transferase (TdT), also known as DNA nucleotidylexotransferase (DNNT) or terminal transferase, is a specialized DNA polymerase expressed in pro-B, pre-B, pro T, and pre-T lymphoid cells. TdT adds N-nucleotides to the V, D, and J exons of the TCR and BCR genes during antibody gene recombination, enabling the phenomenon of junctional diversity. So Dntt doesn't play any role in HSCs.

RESPONSE: We thank the Reviewer for these educational comments. We understand that the Dntt are involved in adding N-nucleotides to the V, D, and J exons of the TCR and BCR genes during antibody gene recombination, not in B cell lineage commitment at HSPC. Although the remarkably decreased Dntt expression in KO may prevent the normal rearrangement of antigen receptor genes in pro-B and pro-T cells, thereby impairing the development of B and T lymphocytes, we cannot explain that Dntt loss is the primary event in the observed phenotype of Brd9 KO model. Moreover, as the Reviewer suggested, we don't have clear evidence for the negative impact of two loop anchors in KO near Dntt on the transcription, although the transcription of genes outside loops were significantly lower than that of loop-anchored genes on a genome-wide scale (revised Figure S11h). Thus, we deleted the Dntt HiC data in the original Fig. 6J and modified the manuscript accordingly.

Fig. 7J. There is no need to put citation information on the y-axis.

RESPONSE: We revised the figures accordingly.

Fig. 7. The authors examined the role of BRD9 in myeloid leukemia and found that BRD9 is essential for both the initiation and maintenance of myeloid leukemia. The authors also performed ATAC-seq assays and observed that BRD9 loss resulted in increased chromatin accessibility, while motif analysis of up-regulated chromatin accessible regions also revealed enrichment of CTCF. But the results cannot explain the role of BRD9 in the initiation and maintenance of leukemia, except for the known BRD9 target myc gene. The author needs to explain this.

RESPONSE: We thank the Reviewer for this insightful question. We believe that these data are very informative for the future study of BRD9 inhibition or degradation for AML treatment. In addition to the role of BRD9 sustaining MYC expression via its interaction with the MYC super-enhancer, BRD9 inhibition/degradation consistently impaired ribosomal programs in transcriptome analysis. We newly performed RNA-seq analysis using dBRD9 (a protein degrader against human BRD9 protein, Remillard et al. Angew Chem Int Ed Engl 2017. PMID: 28418626)-treated MOLM-13 cells (50 nM for four days) and found that the ribosome biogenesis pathways were most significantly downregulated in dBRD9 group. It is well described that sustaining ribosomal functions are essential for AML maintenance (Pauli et al. Blood 2020. PMID: 32097467). We added these data in revised Figure S12d-e.

Altogether, the authors present an extensive study on the role of BRD9 in HSC differentiation, providing many valuable data. But the authors' interpretation of their experimental results is not plausible, especially regarding the role of BRD9 in B cell commitment and development. The authors observed that BRD9 loss resulted in increased CTCF occupancy in the promoter region. The authors performed HiC assays to detect changes in chromatin organization. But the authors' analysis of chromatin loops has something unreasonable. The authors' examples of chromatin loops affecting expression are either incomplete or misinterpreted. Overall, the conclusion presented by the authors is unconvincing: BRD9 loss promotes myeloid lineage skewing at the expense of B cell development.

RESPONSE: We thank the Reviewer for the careful evaluation of our original manuscript. Following the constructive feedback, we greatly improved the resolution of the study, especially at B cell commitment and chromatin loops.

Reviewer #3 (Remarks to the Author):

In their manuscript, Xiao, Kondo, Nomura et al. showed that BRD9 is important for human HSC differentiation and stemness. Importantly, they generated a novel conditional BRD9 KO mice to study in vivo effects of BRD9 loss on haematopoiesis. Through multi-omics studies including scRNA-Seq, they identified changes at the cell types and CTCF binding upon BRD9 KO. They showed an interplay between BRD9 depletion-mediated CTCF binding and chromatin looping enhancement that result in mainly gene activation. The text and data presentation are clear, and the manuscript is likely be of interest to the fields of gene regulation, leukemia and haematopoiesis. I have a few suggestions to help strengthen the main take-home messages.

RESPONSE: We are thankful to receive the positive comments from the Reviewer. Our point-by-point responses to the Reviewer's comments are noted below.

Major Points:

1. As Brd9 KO cells are analysed 8 week after plpC injection, this raises the concern of indirect/secondary effects of BRD9 depletion. Could authors validate their main conclusions in Figure 5F-H and 6E-J by using short term BRD9 perturbation for e.g. 12h or 24h either by BRD9 bromodomain inhibitors, BRD9 degrader or siRNA mediated BRD9 depletion in cultured cells? This would be of crucial importance to validate the authors' mechanistic explanations regarding CTCF enrichment and gene regulation.

RESPONSE: We thank the Reviewer for this critical point. As the Reviewer suggested, we evaluated chromatin and CTCF state shortly after BRD9 depletion using dBRD9, a well-studied specific degrader against human BRD9 protein (Remillard et al. Angew Chem Int Ed Engl 2017. PMID: 28418626). As for the timepoint, we treated K562 cells for four days (100 nM dBRD9) to obtain complete degradation and avoid secondary effects. Of interest, even with short-term depletion of BRD9, we clearly observed differential gene expression between DMSO and dBRD9 group (revised Figure S8a-b). Importantly, we confirmed the increment of CTCF peaks by ChIP-seq after the short-term depletion of BRD9 (revised Figure S8c). Moreover, when we performed the TF motif analysis at the opened chromatin sites by BRD9 depletion, CTCF was most highly enriched in MACS2-called peaks specific to the dBRD9 group (revised Figure S8d). These results indicate that the alteration of chromatin structure and the increment of CTCF peaks occur as a primary or direct event after BRD9 depletion. We added these data in revised Figure S8.

2. The following sentence in summary is not correct: "While cBAF and PBAF regulate fates of multiple cell types, roles for ncBAF in stem cells have not been investigated." Please see <https://doi.org/10.1038/s41467-018-07528-9> which identified a role for ncBAF in regulating mouse naïve pluripotent stem cells as well as <https://www.biorxiv.org/content/10.1101/2021.05.27.445940v1> for the description of a role of BRD9 in human pluripotent stem cells. Therefore, this sentence should be revised, and the relevant references should be added to the introduction or discussion.

RESPONSE: We agree with the Reviewer. We should mention HSCs, not entire "stem cells". We cited the indicated papers to show the literature where researchers' efforts on clarifying the roles of ncBAF in human/murine pluripotent stem cells in Discussion.

3. As BRD9 bromodomain was shown to be important in B-cell development and stemness in Figure1, it would be valuable to show if BRD9 bromodomain inhibitors exhibit similar effects.

RESPONSE: We agree with the Reviewer that these are important points. We have evaluated how BRD9 bromodomain inhibitors affect B-cell development and stemness. As expected, the well-studied inhibitor (BI-7273, Martin et al. J Med Chem 2016. PMID: 26914985) reduced the pre-B colonies and pre-B cell proportion in a dose-dependent manner, indicating the functional roles of the bromodomain in B-cell development. We added the result in revised Figure 2f and S2h.

4. Please revise the analysis in Figure 5B given that H3K4me1 is not a mark for poised sites if it co-occupies the regulatory element with H3K27ac. Additionally, it would be necessary to show genomic distribution of BRD9+BRG1- peaks as the authors claimed that BRD9 localized to the genome via ncBAF.

RESPONSE: We thank the Reviewer for this vital comment and suggestion. It was technically challenging to clearly distinguish primed sites from active enhancers using K-mean clustering analysis since H3K27Ac is still detected in the original “primed sites” enriched with H3K4me1 mark. As the Reviewer suggested, we combined the proportion of primed sites and active enhancers as “enhancers” and revised the pie chart (Revised Figure 5b) and modified revised Figure S7d-e. We also added information on the genomic distribution of limited number of BRD9+BRG1- peaks (ncBAF-independent BRD9 peaks, n=273) (revised Figure 5b and S7c). Considering that the number of BRD9+BRG1- peaks was small and that the distribution pattern was quite different from BRD9+BRG1+ or BRD9+ peaks (revised Figure 5b), we concluded that BRD9 localizes to the genome mainly via ncBAF.

5. Could authors show the average expression levels of genes whose promoters show CTCF up, neutral and

down peaks upon BRD9 loss as boxplots and perform relevant statistical tests to ensure CTCF enrichment has any correlation with gene expression (instead of Figure 5G)?

RESPONSE: We agree with the Reviewer that these are essential points. First, we utilized two-sided t-test to evaluate the average value of log2 fold changes compared to 0 in Figure 5g, where the altered peaks in the Up group located at promoter-TSS sites were associated with enhanced transcription (mean log₂FC=0.03648238, p=1.2838e-08), in contrast to Neutral/Down promoter peaks (mean log₂FC=0.01032778, p=0.06398 and mean log₂FC=0.01327493, p=0.7486, respectively). Next, according to the Reviewer's suggestion, we added the violin plot figure to show a distribution of log2 fold change for each category. When we performed a two-sided KS test, we observed a modest difference in distributions of log2 fold change of gene expressions between CTCF-up and CTCF-neutral (p=0.062). However, we could not see much difference between CTCF-up and CTCF-down groups (p=0.457). The reason for this is basically due to the limited sample size in CTCF-down (<100). We added the violin plot in revised Figure S7j.

6. Similar to what authors performed for CTCF enrichment, loop formation and gene regulation upon BRD9 KO as in Figure 6H-J, could the authors perform similar analysis on YY1 (since they showed a decrease in chromatin accessibility around YY1 binding motifs upon BRD9 KO)? If this is not feasible during revision stage, could authors provide YY1 enrichment for BRD9 bound genomic loci as heatmaps (by using a publicly available data)?

RESPONSE: We thank the Reviewer for this insightful suggestion. Although we tried to perform endogenous YY1 ChIP several times, we could not obtain enough DNA for sequencing. We therefore instead reanalyzed a publicly available dataset for YY1 binding genome in lineage negative wildtype bone marrow cells (GSE184776, Yang et al. Nat Commun 2022. PMID: 35915095) and evaluated the relevance of YY1 with BRD9 and CTCF. This analysis revealed the following findings: (1) YY1 peaks significantly overlap with BRD9 peaks, not with CTCF peaks when evaluated by hierarchical clustering performed on ChIP-seq read density over the merged set of peaks across all ChIPs, (2) CTCF cannot localize the regions with high YY1 enrichment both in BRD9 wildtype and KO conditions. Thanks to the Reviewer's suggestion, these results indicate that CTCF and YY1 are relatively exclusive on the genome and that CTCF-upregulated sites by BRD9 KO don't harbor YY1 binding. We added the information in revised Figure S8e-f and Discussion.

7. The following sentence is not supported by data "Altogether, these findings indicate that the CTCF enrichment after BRD9 depletion occurs selectively and enhanced CTCF peaks at gene promoters results in active transcription for the observed phenotype." Could authors calculate the significance of enhanced CTCF peaks upon BRD9 loss compared to weakened CTCF peaks? Additionally, could authors calculate the significance of observed differential CTCF peaks around promoters upon BRD9 loss compared to expected ones?

RESPONSE: We thank the Reviewer for the comments. Performing Fisher's exact test, we found that enhanced CTCF peaks are enriched in promoter regions compared to weakened CTCF peaks upon BRD9 loss (p-value<2.2e-16). We also calculated Log₂ Ratio (obs/exp) and found that enhanced CTCF peaks are enriched in

promoters ($\text{Log}_2 \text{Ratio} = 3.566$). Log ratio (obs/exp) represents the ratio of the observed proportion of the peaks against the expected ones in a given feature. We added this information in Revised Figure S7k.

Minor points:

1. Y-axis in Figure 3K should be equalized for both wild type and Brd9 KO.

RESPONSE: We revised the figure accordingly (Revised Figure S5e).

REVIEWERS' COMMENTS

Reviewer #1 (Remarks to the Author):

The authors have satisfactorily addressed all of my concerns with the exception of the following points:

1. Revised Figure S4H – please include statistical analysis, accounting for the replicates used
2. Revised Figure S4A – please include statistical analysis or note that none of these results are not significant?
3. The authors should include a paragraph in their discussion elaborating on their interpretation of the competitive transplant data (Rebuttal point 5)
4. Revised Figure S6 – I would still like to see a picture of the BRD9 excision PCR or western blot in the blood and spleen included in this analysis
5. Revised Figure S3B – did the authors evaluate bone marrow morphology for dysplastic cells? I would not feel comfortable stating this phenotype without it and based on the peripheral blood findings only, which are not as convincing in my mind.

Reviewer #2 (Remarks to the Author):

This manuscript provides comprehensive data on the role of BRD9 in hematopoietic stem cells, making a significant contribution to our understanding of the mechanisms involved in their maintenance and differentiation. The revised version has notably improved, especially in its detailed description of BRD9's impact on B cell determination and development, the functional role of Dnmt, and the presentation and elucidation of chromatin loops. These modifications have addressed my concerns effectively, rendering the manuscript suitable for publication.

Reviewer #3 (Remarks to the Author):

In the revised manuscript the authors have addressed all the major issues and strengthened the conclusions by newly generated data with acute degradation of BRD9. I commend the authors for this work and support its publication in its revised form.

REVIEWER COMMENTS

Reviewer #1 (Remarks to the Author):

The authors have satisfactorily addressed all of my concerns with the exception of the following points:

1. Revised Figure S4H – please include statistical analysis, accounting for the replicates used

RESPONSE: The data of Figure S4H were obtained from a single scRNA-seq result in each group.

2. Revised Figure S4A – please include statistical analysis or note that none of these results are not significant?

RESPONSE: We modified the Figure S4A accordingly.

3. The authors should include a paragraph in their discussion elaborating on their interpretation of the competitive transplant data (Rebuttal point 5)

RESPONSE: We agree with the Reviewer and we included a paragraph regarding the interpretation of the competitive transplant data in the Discussion section accordingly.

4. Revised Figure S6 – I would still like to see a picture of the BRD9 excision PCR or western blot in the blood and spleen included in this analysis

RESPONSE: We agree with the Reviewer and added the picture of the BRD9 excision PCR and western blot (revised Figure S6E).

5. Revised Figure S3B – did the authors evaluate bone marrow morphology for dysplastic cells? I would not feel comfortable stating this phenotype without it and based on the peripheral blood findings only, which are not as convincing in my mind.

RESPONSE: We agree with the Reviewer and modified the Figure S3B to clearly demonstrate the bone marrow morphology for dysplastic cells accordingly.

Reviewer #2 (Remarks to the Author):

This manuscript provides comprehensive data on the role of BRD9 in hematopoietic stem cells, making a significant contribution to our understanding of the mechanisms involved in their maintenance and differentiation. The revised version has notably improved, especially in its detailed description of BRD9's impact on B cell determination and development, the functional role of Dntt, and the presentation and elucidation of chromatin loops. These modifications have addressed my concerns effectively, rendering the manuscript suitable for publication.

RESPONSE: We thank the Reviewer for the careful reevaluation of our manuscript and positive feedback.

Reviewer #3 (Remarks to the Author):

In the revised manuscript the authors have addressed all the major issues and strengthened the conclusions by newly generated data with acute degradation of BRD9. I commend the authors for this work and support its publication in its revised form.

RESPONSE: We thank the Reviewer for the careful reevaluation of our manuscript and positive feedback.